# Key innovations and the diversification of Hymenoptera

**Bonnie B. Blaimer** [1,2] ✉, **Bernardo F. Santos** [1,2], **Astrid Cruaud** [3], **Michael W. Gates** [4], **Robert R. Kula** [4], **István Mikó** [5], **Jean-Yves Rasplus** [3], **David R. Smith** [4], **Elijah J. Talamas** [6], **Seán G. Brady** [2] & **Matthew L. Buffington** [4]

The order Hymenoptera (wasps, ants, sawflies, and bees) represents one of the most diverse animal lineages, but whether specific key innovations have contributed to its diversification is still unknown. We assembled the largest time-calibrated phylogeny of Hymenoptera to date and investigated the origin and possible correlation of particular morphological and behavioral innovations with diversification in the order: the wasp waist of Apocrita; the stinger of Aculeata; parasitoidism, a specialized form of carnivory; and secondary phytophagy, a reversal to plant-feeding. Here, we show that parasitoidism has been the dominant strategy since the Late Triassic in Hymenoptera, but was not an immediate driver of diversification. Instead, transitions to secondary phytophagy (from parasitoidism) had a major influence on diversification rate in Hymenoptera. Support for the stinger and the wasp waist as key innovations remains equivocal, but these traits may have laid the anatomical and behavioral foundations for adaptations more directly associated with diversification.

The question of why some groups of organisms have diversified more than others has fascinated biologists since the early days of phylogenetics. The success of particular clades has often been attributed to the evolution of novel traits or key innovations conferring an evolutionary advantage[1,2], and linking these traits to changes in diversification rates estimated from phylogenies has been a major goal in macroevolutionary research[3–6]. The concept of what defines a key innovation has been fluid[1,2,7]. Originally developed to describe traits that facilitate the radiation of a clade into new adaptive zones, it has more recently been generalized in macroevolutionary studies to describe an evolutionary change in a trait leading to increased species diversification in the clade that possesses this trait[1,6,8]. This broader definition has appeal due to its simplicity with regard to hypothesis testing; however, the idea of one trait single-handedly influencing survival and diversification has been criticized as overly simplistic[7]. More nuanced concepts and terminology have therefore been suggested for more complex evolutionary scenarios involving multiple traits[9–11].

Much speculation and numerous hypotheses have attempted to account for the diversity of insects, summarized by Mayhew[12]. In particular, several morphological, physiological, and behavioral traits have been suggested as key innovations promoting the early diversification of insects and greatly influencing their success. These traits include the evolution of insect flight, complete metamorphosis, and phytophagy[6,8,13–16]. The Hymenoptera, which comprise the ants, bees, wasps, and sawflies, is one of the most species-rich and abundant insect orders[17,18], possessing a remarkable diversity of life histories and

[1]Museum für Naturkunde, Leibniz Institute for Evolution and Biodiversity Science, Center for Integrative Biodiversity Discovery, Invalidenstraße 43, Berlin 10115, Germany. [2]National Museum of Natural History, Smithsonian Institution, 10th & Constitution Ave. NW, Washington, DC, USA. [3]CBGP, INRAe, CIRAD, IRD, Montpellier SupAgro, Université de Montpellier, Montpellier, France. [4]Systematic Entomology Laboratory, USDA-ARS, c/o NMNH, Smithsonian Institution, 10th & Constitution Ave. NW, Washington, DC, USA. [5]Department of Biological Sciences, University of New Hampshire, Durham, NH, USA. [6]Florida State Collection of Arthropods, Division of Plant Industry, Florida Department of Agriculture and Consumer Services, 1911 SW 34th St, Gainesville, FL 32608, USA. ✉e-mail: bonnie.blaimer@mfn.berlin

morphological adaptations, some of which set records among the insects. For example, the smallest insect on Earth is a fairy wasp of the genus *Dicopomorpha* (Mymaridae)[19], while the longest egg-laying organ (the ovipositor, measured in absolute size) occurs in Darwin wasps of the genus *Megarhyssa* (Ichneumonidae)[20]. These two extreme examples have a major life history strategy in common: they are both parasitoids, carnivores that complete their entire life cycle feeding on just one individual prey item, the host[21]. However, they are adapted to very different hosts: *Dicopomorpha* is an egg parasitoid of bark lice (Psocodea: Lepidopsocidae)[22], while *Megarhyssa* is a parasitoid of wood-boring horntail larvae (Hymenoptera: Siricidae)[20]. A suite of morphological, physiological, and genetic adaptations provides the means for Hymenoptera to exploit a myriad of host niches[23]. In fact, about 70% of all described hymenopterans are parasitoids, while the other 30% are phytophages, such as leaf-feeding and wood-boring sawflies, gall-inducing wasps, and pollen-collecting bees, or predators, such as many social wasps. Among the insects, Hymenoptera is unique in that they comprise 75–80% of all described parasitoid insect species[24,25]. Parasitoidism and its associated features may be key drivers that explain diversification in Hymenoptera, as the adaptation to different host species and, therefore, niche subdivision may have resulted in increased speciation rates in parasitoids[26].

Two other defining morphological features within the early evolution of Hymenoptera are putative key innovations. First, in the lineages of Hymenoptera comprising the grade of taxa commonly referred to as the "sawflies and wood wasps" ("Symphyta"), the abdomen is broadly attached to the thorax, which is the plesiomorphic ground plan for Hymenoptera[27,28]. In contrast, in apocritan wasps (suborder Apocrita), which comprise the Aculeata and numerous other primarily parasitoid lineages, the first abdominal tergum is fused with the metathorax forming the propodeum. The thorax and propodeum together form the mesosoma, which is connected to the remainder of the abdomen (the metasoma) via a constricted articulation[29]. It is this narrow wasp waist that gives female apocritan wasps the flexibility to bend their metasoma, enabling them to insert their ovipositor or stinger into a substrate, prey item, or enemy[29], and thus potentially facilitating the success of both parasitoidism and stinging in apocritan wasps. Second, in the Aculeata, the female ovipositor is modified from an egg-laying to a stinging apparatus (the stinger)[30]. This group of stinging wasps also includes all social and colony-forming lineages of Hymenoptera, such as ants and certain groups of bees and wasps. In social hymenopterans, the stinger enables the defense of a nest and its brood, and possibly was an important factor in the diversification of Aculeata, as the increased defense system may have substantially decreased the risk of extinction in this clade.

While carnivory and parasitoidism are pervasive across the apocritan wasps, extant members of the sawfly and woodwasp lineages are generally phytophagous. This feeding strategy is therefore considered ancestral in Hymenoptera[14]. However, phytophagy has secondarily evolved in several groups of apocritan wasps, for example, in pollen-collecting bees or gall-inducing cynipoid wasps. Arguably, these transitions to a secondarily phytophagous life strategy were additional major innovations in Hymenoptera as they represent the formation of new niches (e.g., use of plant galls and pollen) and not reversals to a symphytan-style phytophagy. All four innovations—parasitoidism, the wasp waist, the stinger, and secondary phytophagy—could thus potentially be catalytic in the diversification of Hymenoptera[26,27,29,31–33], yet they have never been analyzed across a phylogeny of the entire order in a macroevolutionary framework.

Phylogenetic relationships among major hymenopteran lineages have been the focus of much scrutiny over the last decade, based on either morphology or molecular data, or both[29,33–39], but even the most recent phylogenomic analyses[36,38,40] were not able to provide clarity on the placement of some superfamily-level lineages, mainly due to sampling bias toward aculeate wasps. In this study, we infer a robust

and balanced family-level phylogenetic framework from 765 taxa representing 94 families in Hymenoptera to study the early diversification and evolution of the four putative key innovations in Hymenoptera. Phylogenomic analyses are based on -1,100 loci of ultraconserved elements (UCE), a group of markers widely used across higher-level groups of insects[38,41,42], largely generated de novo by us. From these data, we estimate the most extensive phylogeny for Hymenoptera to date and establish support for hypotheses of superfamily-level relationships. Time-calibrated phylogenies infer the early evolution of parasitoidism in the order and allow us to investigate whether the four innovations outlined above are associated with the diversification dynamics of Hymenoptera by using macroevolutionary models. We discuss our results within a broader framework of insect evolution and diversification and give a perspective of future avenues for investigation.

## Results

### Phylogenetic inference and major relationships within Hymenoptera

Our phylogenetic results are covered in detail in the Supplementary Discussion. In brief, to infer a robust phylogeny for Hymenoptera, we assembled our data set in several arrangements with varying levels of missing data and phylogenetic information content. We first filtered all captured UCE loci for taxon completeness, retaining a 50%, 60%, and 70% complete matrix (nuc-50% = 1118 loci, nuc-60% = 765 loci and nuc-70% = 446 loci) for further analyses. UCE loci were also matched to a reference protein database, and a 50% taxon-complete set (=324 loci) of protein-coding loci was created, which was further analyzed as nucleotide (prot-NUC) and amino acid (prot-AA) data set. These five matrices (nuc-50%, nuc-60%, nuc-70%, prot-AA, prot-NUC) were then analyzed via partitioned and unpartitioned analyses in a maximum likelihood framework. Supplementary Fig. 1 provides a flowchart overview of analyses; descriptions of data characteristics and filtering can be found in Supplementary Data 1–5 and in the Supplementary Methods, together with additional analyses that explored the impact of GC content on phylogenetic inference.

We recovered full support for two subdivisions within Apocrita, consisting of (1) Proctotrupomorpha, a clade including the superfamilies Chalcidoidea, Mymarommatoidea, Diaprioidea, Proctotrupoidea, Cynipoidea and Platygastroidea (ultrafast bootstrap support (ufBS) = 95–100, across the main analyses summarized in Fig. 1 and Supplementary Fig. 2), and (2) a clade consisting of Aculeata and the former "Evaniomorpha" (itself a grade). All analyses recover Orussoidea as a sister to Apocrita with ufBS=100 (Fig. 1 and Supplementary Fig. 2), corroborating Vespina (sensu Rasnitsyn[43]). Our nucleotide data sets recovered Tenthredinoidea as the sister group to all other Hymenoptera (Fig. 1 and Supplementary Fig. 2), a unique result compared to previous studies. In the protein-coding, amino acid translated matrix, the sister group to all other Hymenoptera was Tenthredinoidea + (Xyeloidea + Pamphilioidea), similar to previous results using transcriptomes[36]. Full phylogenetic trees are shown in Supplementary Figs. 3–15. We recovered three competing hypotheses regarding the position of Ichneumonoidea, in combination with three alternative positions of Ceraphronoidea. We summarized these by establishing mutually exclusive topology groupings based on the superfamily-level relationships (summarized in Supplementary Table 1). We used the best supported (based on ufBS and Four-cluster Likelihood Mapping (FcLM)[44]) superfamily-level topology C-1 (=topC-1: Ichneumonoidea sister to the rest of Apocrita; Ceraphronoidea sister to Evanioidea + Stephanoidea) and the most frequently recovered topology A-0 (=topA-0: Ichneumonoidea sister to Ceraphronoidea, together sister to Proctotrupomorpha) as input for divergence dating and macroevolutionary analyses. However, the alternate positions of these two superfamilies had little impact on our results investigating

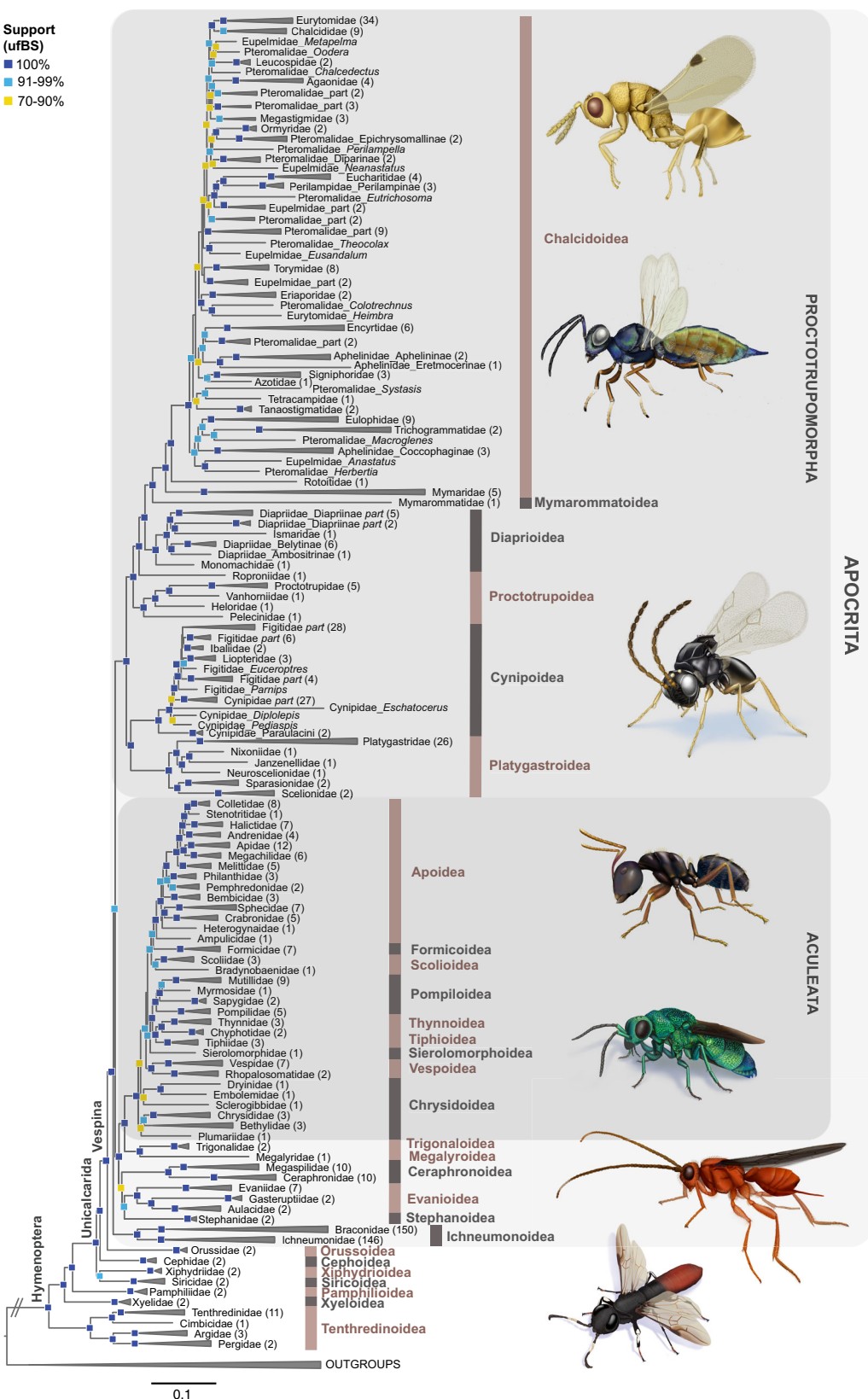

**Fig. 1 | Family-level phylogeny of Hymenoptera.** Phylogeny shows relationships between hymenopteran families as estimated from 446 UCE loci in the nuc-70%-SWSC analysis. This data set was partitioned using the Sliding-Window Site Characteristics Entropy (SWSC-EN) algorithm[106] and PartitionFinder2[107] in combination with the r cluster algorithm[108] and analyzed using Maximum Likelihood (ML) best-tree and ultrafast bootstrap searches in IQ-TREE v1.6.10[109]. This result is referred to as topology C-1 (topC-1) throughout the text and the remaining figures and tables.

Various nodes have been collapsed for clarity of display, with numbers of species subtended by the respective branches included in brackets. Ultrafast bootstrap (ufBS) support values are indicated by colored squares on respective nodes: dark blue = 100%, light blue = 91–99%, and yellow = 70–90%. Support values lower than ufBS = 70 are not shown. Scalebar represents substitutions/bp. Source data for this figure can be found in the Dryad repository at https://doi.org/10.5061/dryad.08kprr54m (folder 2.1.7).

the evolution of key innovations and the diversification history of Hymenoptera.

## Timescale of hymenopteran evolution and their life histories

We estimated divergence times within Hymenoptera, employing 12 fossil calibrations as minima plus a minimum-maximum bound on the root node. Time-calibrated phylogenies were then used to reconstruct the evolution of life history strategies (parasitoidism, phytophagy, predation, and secondary phytophagy) in Hymenoptera. A model with a rate matrix of equal transition frequencies (ER) provided the best fit for all ancestral state reconstructions. Analyses using topC-1 and topA-0 generated similar results (Fig. 2; Supplementary Figs 16–19; and Supplementary Data 6–8); thus, mainly results based on topC-1 are discussed unless specified.

Hymenoptera was estimated with a median crown-group origin of 280 million years ago (Ma) in the Permian (Fig. 2 and Table 1). The sawfly and woodwasp lineages diverged from the remaining Hymenoptera, respectively, over the course of the next 50 million years, throughout the Permian and Early to Mid-Triassic (Fig. 2). Unicalcarida and Vespina (Orussoidea + Apocrita) originated in the Early Triassic (node 4 and 6, Fig. 2) around 248 Ma and 234 Ma, respectively. Crown-group apocritan wasps with their constricted wasp waist appeared just a few million years later, around 226 Ma (node 7, Fig. 2). Our ancestral state reconstructions confirmed that the earliest hymenopterans were plant-feeding insects similar to the extant members of the sawfly and woodwasp grade. The first transition between phytophagy and parasitoidism is estimated in the Late Triassic in the most recent common ancestor (MRCA) of Vespina (node 6, Fig. 2; Supplementary Figs. 18–19). Parasitoidism thus evolved once and remained the dominant life strategy in Hymenoptera, with no subsequent major innovations in life history evolving until the Early Cretaceous around 140 Ma (Fig. 2). Proctotrupomorpha and the Aculeata + "Evaniomorpha" grade both originated in the Late Triassic, around 210 Ma and 206 Ma, respectively (node 9 and node 24, Fig. 2 and Table 1). The origin of crown-group Ichneumonoidea is also estimated within a similar timeframe in the Late Triassic (206 Ma, node 64; Fig. 2), while most of the other non-aculeate superfamilies originated throughout the Jurassic and Early Cretaceous (Table 1). Aculeata was estimated with an age of 142 Ma (node 26; Fig. 2), placing the origin of the modified ovipositor-stinger at the edge of the Jurassic-Cretaceous boundary. Within Aculeata, three transitions to a non-parasitoid strategy are reconstructed in the Late Cretaceous (starting ca. 80 Ma, Fig. 2), two to a mixed predatory and secondarily phytophagous lifestyle in vespid wasps and ants, and one to secondary phytophagy (pollen-collecting) in bees (node 36, Fig. 2). Another major transition to secondary phytophagy, in this case plant-galling, occurred within Cynipoidea in the Early Cretaceous (around 105 Ma, node 20, Fig. 2). Several additional adaptations to plant-galling in Chalcidoidea are not captured by these results, as all chalcidoid families were collapsed into one clade and coded as polymorphic in our analyses. Independent transitions from a parasitoid to a predatory lifestyle occurred in Evaniidae and Gasteruptiidae.

## Diversification history of Hymenoptera

Net diversification rates and potential rate shifts in Hymenoptera were estimated using Bayesian Analysis of Mixture Models (BAMM[45,46]) and stepwise AIC (MEDUSA[47]). In BAMM, we estimated and plotted the best shift configuration with maximum a posteriori probability (MAP), mean phylorate, and cumulative shift probabilities. Six (topC-1) and eight (topA-0) rate shifts were estimated in the best shift configurations (Fig. 3a, Table 2, and Supplementary Fig. 20a). To quantify all rate shifts, we calculated the mean net diversification rates (estimated with BAMM) for all clades for which rate shifts were indicated by either of the above analyses (Table 2; Supplementary Data 9). Shifts in net

diversification rate that are supported by BAMM across both topologies occur in Aculeata minus Chrysidoidea (non-chrysidoid aculeates hereafter), within the bees minus family Melittidae (non-melittid bees hereafter), on the branch leading to Cynipidae s.s. (sensu stricto, or in the narrow sense), on the branch leading to (or within) Eurytomidae (Eurytominae), and within Ichneumonidae (shifts 3–6, and 1, respectively; Fig. 3 and Supplementary Fig. 20). All of these shifts represent increases in net diversification compared to the background rate across the phylogeny (Table 2). Additionally, for topC-1 a rate increase (i.e. positive shift) is estimated on the branch leading to Apocrita minus Ichneumonoidea (shift 2, Fig. 3a, Table 2), while for topA-0 rate decreases (i.e. negative shifts) are estimated on the branches leading to Tenthredinoidea and Ichneumonoidea (shifts 7, 8), and a rate increase on the branch leading to Apocrita (shift 2) (Supplementary Fig. 20a, Table 2). These MAP shift configurations only had a probability of 0.019 (topC-1) and 0.009 (topA-0) among 9910 and 12,354 distinct shift configurations in the 95% credibility set, respectively; however, the next-best shift configurations were very similar (Supplementary Fig. 21).

Due to the low probability for MAP shift configurations, we summarized support for shifts also by plotting cumulative shift probability (CSP) trees, which show the cumulative probability for each branch that a shift occurred somewhere between the root of the tree and the branch under scrutiny. For topC-1, the CSP tree clearly shows that only the positive shifts leading to the non-melittid bees, Cynipidae s.s., Eurytomidae (Eurytominae), and Ichneumonidae (internal) are present across most of the credible shift set (0.97–1.0 probability) (Fig. 3b). Additionally, a shift is indicated leading up to Chalcidoidea that is not present in the best shift configuration. For topA-0, all shifts in the best shift configuration are present in the CSP tree with high probability, except the negative rate shifts on the branches leading to Tenthredinoidea and Apocrita (Supplementary Fig. 20b).

MEDUSA analyses estimated six (topC-1) and five (topA-0) distinct rate regimes within Hymenoptera, which are summarized in Table 3 (see also Supplementary Fig. 22). Both analyses agreed on a positive shift in diversification rate within non-chrysidoid aculeates and within non-melittid bees (shifts ii and iii, Table 3, respectively). The analysis using topC-1 additionally estimated shifts to a faster rate regime along the branch leading to Cynipoidea and further to a core cynipoid clade comprising Cynipidae s.s. and Figitidae s.l. (sensu lato, in the wider sense) (shifts vi and v, Supplementary Fig. 22a; Table 3). Both analyses estimated a negative shift in diversification rate ancestral to Xiphydriidae + Siricidae (shift iv, Table 3, and Supplementary Fig. 22), and the topA-0 analysis further estimated a decreased rate regime for Pamphiliidae + Xyelidae (shift v, Table 3, and Supplementary Fig. 22b).

Thus, two positive rate shifts are supported by the BAMM MAP configuration and MEDUSA: one on the branch to non-aculeate chrysidoids and one for non-melittid bees. Both analyses also place positive shifts within Cynipoidea but at varying locations. Rates for all clades that had a shift indicated in either analysis show a corresponding deviation from the background rate. Interestingly, net diversification rates for Vespina and Aculeata were elevated compared to the background rates, although no shifts were indicated in these clades. The highest net diversification rates are estimated within Ichneumonoidea, Cynipoidea (Cynipidae s.s.), Chalcidoidea (Eurytomidae: Eurytominae), and for non-melittid bees (Table 2); these rate patterns are also visible in mean phylorate plots (Supplementary Fig. 23).

## Key innovations and the diversification of Hymenoptera

The association of four key innovations (parasitoidism, wasp waist, stinger, secondary phytophagy) with diversification in Hymenoptera was assessed using 30 models in the HiSSE (Hidden State Speciation and Extinction) framework (Supplementary Data 10), which can incorporate unobserved traits (hidden states) that may influence diversification rate together with the observed traits[48]. We also

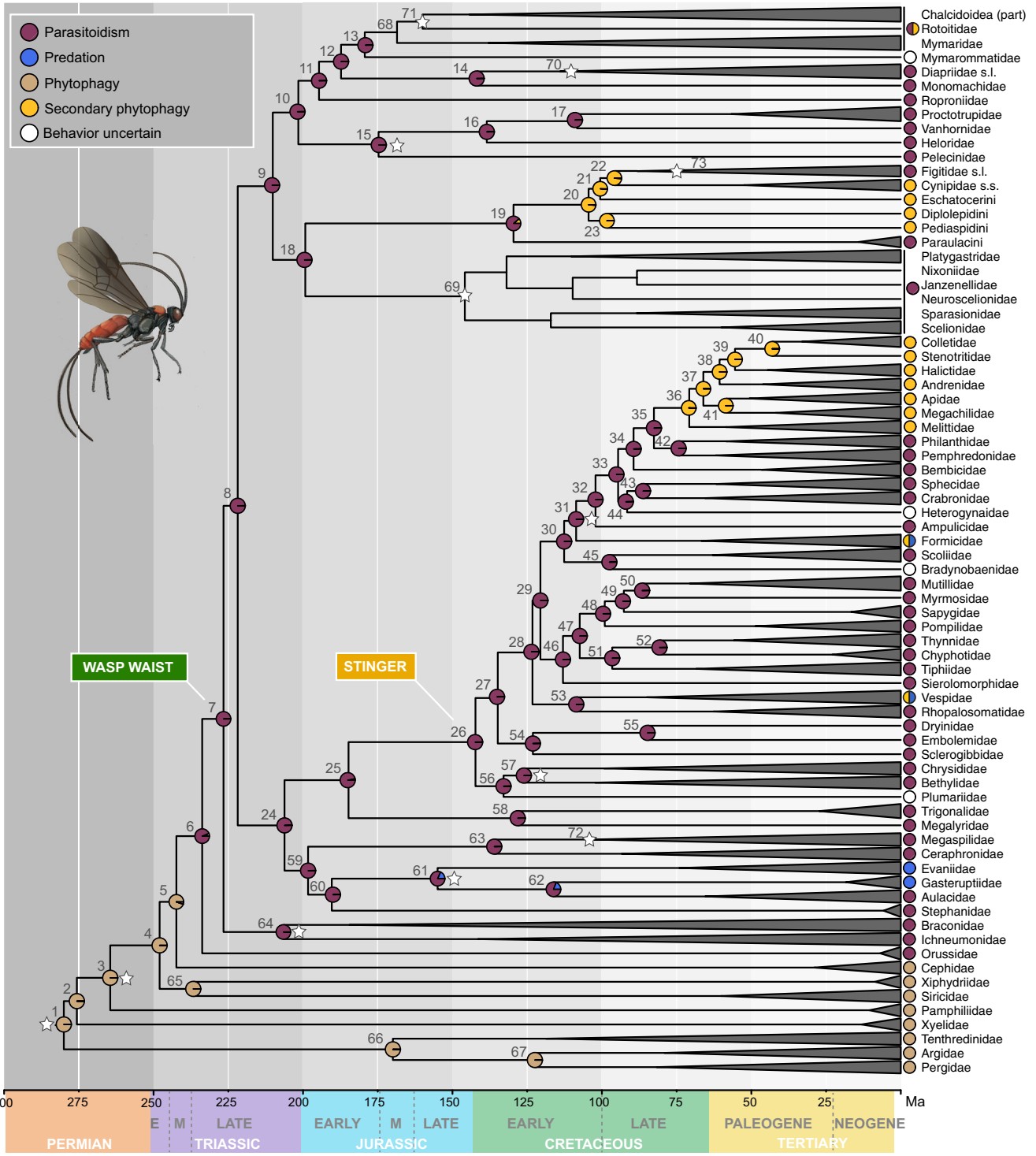

**Fig. 2 | Timeline and evolution of parasitoidism in Hymenoptera.** Chronogram estimated using approximate likelihood in MCMCTREE with the PAMLv4.9 package[113] from the nuc-70% matrix and topology C-1. All outgroups were pruned from the tree and alignment prior to divergence time estimation. Twelve calibration points were used, which are indicated by a white star. Terminals have been collapsed down to family or clade-level post analysis. Families of Platygastroidea are shown in the chronogram, but were lumped at the superfamily level for ancestral state reconstructions; therefore, pies are absent in this clade. Ancestral state reconstructions with corHMM were estimated from topology C-1 and mapped in pie format onto the chronogram; states for terminals are indicated beside terminal branches. Divergence estimates and ancestral state probabilities can be accessed in Supplementary Data 6 and 8, referring to numbers beside nodes. Pie states are red = parasitoid, blue = predatory, brown = phytophagous, yellow = secondarily phytophagous, and white = behavior unknown. For detailed information on methodology and results, also refer to the main text and the Supplementary Methods. Source data for this figure can be found in the Dryad repository at https://doi.org/10.5061/dryad.08kprr54m (folder 2.2.2).

**Table 1 | Divergence time estimates for selected clades within the Hymenoptera**

| | Node | Topology C-1 | | Topology A-O | |
|---|---|---|---|---|---|
| | | Median (Ma) | 95% HPD (Ma) | Median (Ma) | 95% HPD (Ma) |
| Hymenoptera | 1 | 280.0 | 267.1,287.9 | 278.5 | 262.5,287.3 |
| Tenthredinoidea | 66 | 169.9 | 132.9,208.6 | 167.1 | 134.5,208.2 |
| Unicalcarida | 4 | 248.0 | 233.7,261.6 | 256.3 | 239.8,270.7 |
| Vespina | 6 | 233.7 | 220.4,247.8 | 242.6 | 227.2,257.6 |
| Apocrita | 7 | 226.6 | 213.5,240.6 | 233.6 | 218.2,249.1 |
| Ichneumonoidea | 64 | 206.6 | 192.4,221.3 | 203.5 | 186.8,220.3 |
| Proctotrupomorpha | 9 | 210.1 | 1.963,224.6 | 217.0 | 201.6,233.3 |
| Chalcidoidea | 68 | 168.6 | 154.9,183.3 | 166.0 | 151.6,182.0 |
| Proctotrupoidea | 15 | 174.7 | 151.5,195.4 | 176.5 | 153.2,200.0 |
| Platygastroidea – Cynipoidea | 18 | 199.2 | 182.0,216.0 | 206.2 | 188.7,223.7 |
| Cynipoidea | 19 | 129.6 | 109.3,154.4 | 142.9 | 122.5,167.5 |
| Platygastroidea | 69 | 145.9 | 131.1,166.5 | 152.0 | 134.1,174.6 |
| Diaprioidea | 14 | 141.9 | 116.2,173.2 | 147.8 | 123.6,172.6 |
| Aculeata + "Evaniomorpha" grade | 24 | 206.2 | 186.3,224.5 | 220.2 | 197.0,239.3 |
| Aculeata + (Trigonaloidea + Megalyroidea) | 25 | 184.7 | 158.8,207.9 | 192.9 | 163.5,217.9 |
| Trigonaloidea + Megalyroidea | 58 | 128.2 | 85.1,172.9 | 131.9 | 90.6,175.8 |
| Evanioidea | 61 | 155.0 | 130.1,185.2 | 162.1 | 132.0,197.2 |
| Ceraphronoidea | 63 | 136.0 | 113.8,162.9 | 151.9 | 125.8,182.4 |
| Aculeata | 26 | 142.3 | 129.1,157.6 | 142.9 | 129.7,157.8 |
| Apoidea | 32 | 102.1 | 98.5,120.9 | 100.8 | 91.1,111.6 |
| Vespoidea | 53 | 108.6 | 92.7,123.9 | 106.7 | 93.8,120.1 |
| Tiphioidea + Thynnoidea | 51 | 96.6 | 81.8,110.8 | 98.9 | 85.1,112.5 |
| Pompiloidea | 48 | 99.1 | 87.0,112.0 | 101.4 | 89.6,114.0 |

Summarized are median nodes ages and 95% HPD intervals as estimated by MCMCTREE with the PAMLv4.9 package[113]. Node numbers refer to Fig. 2. –refers to a split between two lineages, whereas + indicates a combined clade of two or more lineages. For an extended version of this table, see Supplementary Data 6. Source data for this Table can be found in the Dryad repository at https://doi.org/10.5061/dryad.08kprr54m (folder 2.2).

analyzed carnivory and phytophagy (i.e., primary + secondary) as the fundamental feeding strategies, including parasitoidism and secondary phytophagy, respectively. HiSSE identifies the most probable combination of the observed trait as being absent (0) or present (1) with the estimated hidden state in state A or B, resulting in combined state inferences such as 0A, 0B, 1A, and 1B, which are reported below. We further estimated tip diversification rates across Hymenoptera using MiSSE (Missing State Speciation and Extinction), a trait-free extension of the HiSSE framework. Parameter estimates from best-fitting models are summarized in Supplementary Data 10–13, and visualized in Fig. 4 and Supplementary Figs. 24–28.

Trait-dependent diversification in association with a hidden state was supported for the wasp waist, as a full HiSSE model with irreversible states was found to be the best-scoring model (Table 4). Trait-diversification plots show a higher estimated diversification rate in association with the wasp waist (Fig. 4a; Supplementary Fig. 25a), and net diversification rate showed >100× increase after the transition to the wasp waist in combination with hidden state A and 8.2–11.9× in combination with hidden state B (Supplementary Data 11). Most tip states are estimated with higher probabilities in state 1A (Supplementary Data 12); however, state 1B has a higher net diversification rate than state 1A. Taxa whose marginal probabilities for state 1B are on the higher side (up to 0.58) overlap to some extent with clades that showed elevated rates in BAMM and MEDUSA analyses (Supplementary Data 12), but elevated probabilities for state 1B are also seen in taxa outside these clades.

An association of parasitoidism with diversification in Hymenoptera in combination with a hidden state was suggested, as the full HiSSE model received the most support (Table 4). However, a 0.83–0.84× decrease in net diversification rate from the non-parasitoid to the parasitoid state in combination with hidden state A (i.e., 0A to

1A) was indicated (Supplementary Data 11; Fig. 4c; Supplementary Fig. 25b), and this state was estimated as the likeliest state for most parasitoid taxa (Supplementary Data 12). By contrast, the net diversification rate of the parasitoid state in combination with hidden state B (i.e., 1B) increased by 17.7–29.2×. The highest net diversification rate was estimated for state 0A, yet all non-parasitoid taxa were reconstructed with state 0B as the most probable. Interestingly, some taxa in clades with rate shifts recovered (BAMM shifts 1, 3, 6, 8 and MEDUSA shifts ii, v, vi; Table 4) were estimated with a significant proportional probability in state 1B (up to 0.71), and taxa with probabilities on the higher side (up to 0.38) for state 0A clustered predominantly (but not exclusively) in clades with rate shifts (4, 5, iii; Table 4) as well.

For secondary phytophagy, trait-dependent diversification was also supported by a full HiSSE model with irreversible states (Fig. 4d; Supplementary Fig. 24b); net diversification rates showed a 1.6–21.1× increase in the observed trait in combination with hidden state A and a 0.33–0.95× increase in association with hidden state B, although state 1B has the highest net diversification rate (Supplementary Table 11). Most secondarily phytophagous clades (largely congruent with the rate shifts 4–6 and iii recovered by BAMM and MEDUSA; Table 4) are estimated with state 1A as the most probable, but also have a high proportion of probabilities for state 1B (up to 0.58; Supplementary Table 12). For non-secondarily phytophagous taxa the pattern is reversed, with state 0A being the most probable, but 0B having the faster net diversification rate. Trait-dependent diversification for the fundamental feeding strategy phytophagy was only supported by topC-1, whereas a character-independent model scored best for topA-0 (Table 4, Supplementary Fig. 27). Character-independent models had the highest support for the stinger and carnivory (Table 4; Fig. 4b; Supplementary Figs. 24a and 26), indicating that these traits were not associated with diversification rate.

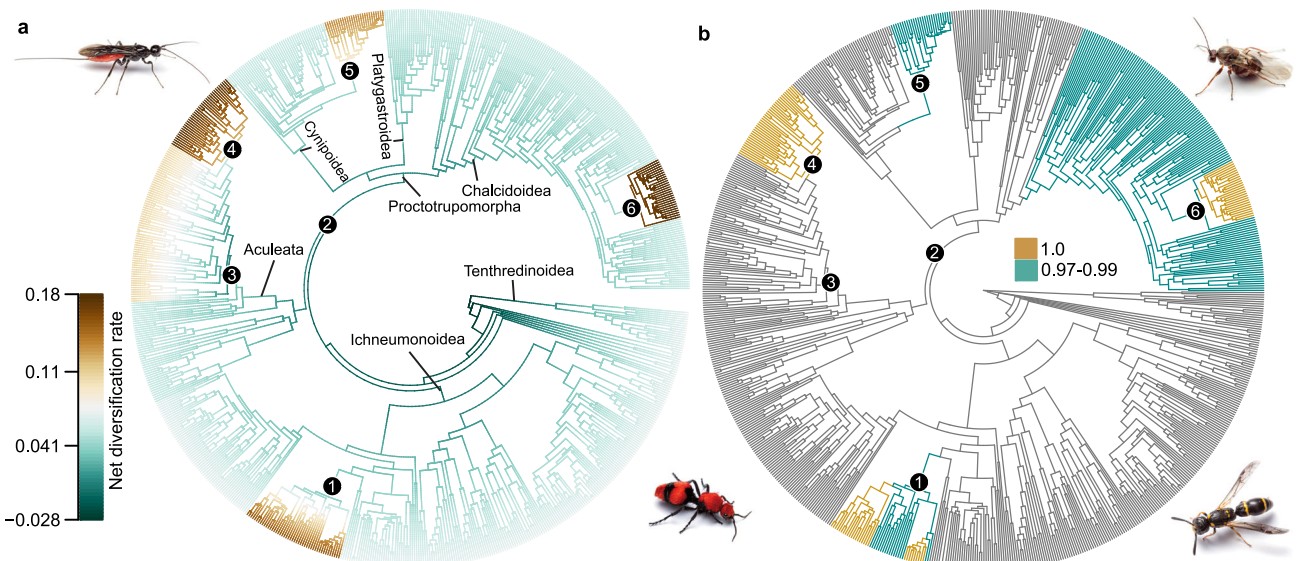

**Fig. 3 | Diversification history of Hymenoptera.** We assessed potential shifts in diversification rates over time in Hymenoptera using a sampling fraction approach and clade-specific sampling probabilities in BAMM v2.5[45,46] and the associated R package BAMMtools v2.1.7[115]. All analyses shown are based on topology C-1. **a** Plot of best shift configuration with maximum a posteriori probability (MAP), indicating rate shifts on respective branches. Rates are shown as net diversification rates. Since this rate shift configuration only has a 0.019 probability among 9910 distinct shift configurations in the 95% credibility set, we summarized the cumulative probabilities for each branch that a shift occurred somewhere between the focal branch and the root of the tree. **b** Cumulative shift probability tree, indicating in dark cyan the branches with a shift probability ≥0.97 ≤ 0.99 and in gold the branches with a cumulative shift probability = 1. Golden branches thus occur in every distinct shift configuration and cyan branches in 97–99% of all distinct shift configurations. Numbered clades/shifts are (1) Ichneumonidae (internal); (2) Apocrita minus Ichneumonoidea; (3) Aculeata minus Chrysidoidea; (4) Bees minus Melittidae; (5) Cynipidae s.s.; (6) Eurytomidae (Eurytominae). For detailed information on methodology and results, refer to the main text, Supplementary Methods, and Supplementary Data 9 and 16. Images of wasps courtesy of Matt Bertone. Source data for this figure can be found in the Dryad repository at https://doi.org/10.5061/dryad.08kprr54m (folder 3.2).

The best-scoring models tested with MiSSE incorporated five (topC-1) and six (topA-0) hidden states. Tip net diversification rates ranged from 0.048 to 0.072 (topC-1) and 0.051 to 0.079 (topA-0). Several clades are highlighted with higher diversification rates (Supplementary Data 13; clades 1–4, Supplementary Fig. 28). These are congruent with BAMM/MEDUSA rate shifts 1, 4/iii, 5, and 6, three of which are associated with secondary phytophagy. The same hidden states (topC-1: 0D; topA-0: 0A) are estimated as most probable in these four clades (but not across most other taxa), suggesting that these states may be associated with secondary phytophagy and a higher diversification rate.

## Discussion

Our work targeted an increased sampling of parasitoid lineages to create an improved phylogenomic framework to evaluate the association of key traits with the diversification of Hymenoptera. A detailed discussion of the phylogenetic insights of our work can be found in the supplementary information accompanying this article. Below we focus on the implications of our time-calibrated hymenopteran phylogeny on the evolution of putative key innovations and the diversification of the order.

The pervasiveness of parasitoidism across Vespina (Orussidae and Apocrita) suggests the ancestry and antiquity of this behavior in Hymenoptera. Several attempts to trace the evolution of the parasitoid lifestyle have been made[24,26,49], but our chronogram has enabled us to reconstruct the origin of parasitoidism with a maximum likelihood-based approach. We find a single evolution of parasitoidism most probable in the MRCA of Vespina, dating to the late Triassic (around 234 Ma). Branstetter et al.[38], Peters et al.[36], and Ronquist et al.[39] estimated the age of Vespina at ~200 Ma, ~247 Ma, and ~270 Ma, respectively. Likely reasons for incongruities are differences in taxon composition between ours and the previous analyses, and different

fossils and calibration strategies (i.e., node-dating in the present, and Branstetter et al.[38] and Peters et al.[36] analyses, whereas Ronquist et al.[39] also used a tip-dating and total-evidence approach). Naturally, the discovery of new fossil evidence may have unforeseeable consequences on our current estimates and the derived diversification patterns. Different sizes and types of sequence data (UCEs vs. transcriptomes in Peters et al.[36] and seven mitochondrial and nuclear markers in Ronquist et al.[39]) and methods of divergence dating (MCMCTREE used by us and Peters et al.[36], BEAST[50] in Branstetter et al.[38], and MrBayes 3.2[51] in Ronquist et al.[39]) most certainly also have had an impact on divergence estimates. We compare the above aspects in more detail in the supplementary discussion; overall, our results are closest to Peters et al.'s[36] estimates (Supplementary Data 6).

Our data suggest that parasitoidism has been the dominant life strategy in Hymenoptera since the late Triassic. Several subsequent switches from a parasitoid strategy to a secondarily phytophagous or predatory habit occurred, yet markedly only from the mid-Cretaceous onward. Diversification of angiosperms may have facilitated the evolution and diversification of gall-inducing and pollen-collecting hymenopterans, as evidenced by the known cases of codiversification between plants and their pollinating hymenopterans[52–55]. For angiosperm diversification to be a facilitator of secondary phytophagy, we would expect the onset of the former first, though not necessarily closely followed by the origins of the latter. The timing of the origin of the angiosperms is contentious, but most estimates (130–180 Ma[56]) indeed predate the evolution of secondarily phytophagous clades (from ca. 105 Ma onwards based on our estimations) in Hymenoptera. Interestingly, the ages of secondarily phytophagous clades correspond with the beginning of the "Angiosperm Terrestrial Revolution" ca. 100 Ma[57]. Angiosperm diversification has also been suggested as a major element in the diversification of tenthredinoid sawflies, which are

**Table 2 | Evolutionary rate dynamics estimated by BAMM and MEDUSA for selected clades**

| | | | Topology C-1 | | | | | | | | Topology A-O | | | | | |
| | | | Focal clade | | | Background rate | | | | | Focal clade | | | Background rate | | |
| | Δ net. div. | ± | net. div. | spec. | ext. | net. div. | spec. | ext. | Δ net. div. | ± | net. div. | spec. | ext. | net. div. | spec. | ext. |
|---|---|---|---|---|---|---|---|---|---|---|---|---|---|---|---|---|
| *Shifts identified by BAMM and MEDUSA* | | | | | | | | | | | | | | | | |
| (3, ii) Aculeata minus Chrysidoidea | 0.031 | + | 0.085 | 0.321 | 0.236 | 0.054 | 0.244 | 0.190 | 0.037 | + | 0.091 | 0.261 | 0.170 | 0.054 | 0.224 | 0.170 |
| (4, iii) Bees minus Melittidae | **0.099** | + | **0.154** | 0.298 | 0.145 | 0.055 | 0.252 | 0.197 | **0.101** | + | **0.156** | 0.291 | 0.134 | 0.055 | 0.226 | 0.171 |
| *Shifts identified by BAMM only* | | | | | | | | | | | | | | | | |
| (1) Ichneumonidae (internal) | 0.050 | + | 0.106 | 0.517 | 0.411 | 0.056 | 0.244 | 0.188 | 0.028 | + | 0.085 | 0.438 | 0.353 | 0.057 | 0.220 | 0.163 |
| (2) Apocrita minus Ichneumonoidea | 0.010 | + | 0.062 | 0.178 | 0.116 | 0.052 | 0.360 | 0.309 | — | | — | — | — | — | — | — |
| (2) Apocrita | — | | — | — | — | — | — | — | 0.033 | + | 0.060 | 0.225 | 0.165 | 0.027 | 0.275 | 0.249 |
| (5) Cynipidae s.s. | **0.058** | + | **0.115** | 0.302 | 0.188 | 0.057 | 0.252 | 0.196 | **0.055** | + | **0.112** | 0.353 | 0.241 | 0.057 | 0.226 | 0.169 |
| (6) Eurytomidae (Eurytominae) | **0.117** | + | **0.173** | 0.260 | 0.087 | 0.056 | 0.253 | 0.197 | — | | — | — | — | — | — | — |
| (6) Eurytomidae (Eurytominae, int.) | — | | — | — | — | — | — | — | **0.128** | + | **0.185** | 0.265 | 0.081 | 0.056 | 0.227 | 0.171 |
| (7) Tenthredinoidea | −0.017 | − | 0.041 | 0.368 | 0.326 | 0.058 | 0.249 | 0.191 | −0.017 | − | 0.042 | 0.307 | 0.265 | 0.059 | 0.225 | 0.166 |
| (8) Ichneumonoidea | −0.003 | − | 0.056 | 0.366 | 0.310 | 0.059 | 0.193 | 0.134 | −0.004 | − | 0.056 | 0.361 | 0.305 | 0.060 | 0.157 | 0.097 |
| *Shifts identified by MEDUSA only* | | | | | | | | | | | | | | | | |
| (v) core Cynipoidea | 0.015 | + | 0.072 | 0.183 | 0.111 | 0.057 | 0.257 | 0.201 | 0.012 | + | 0.069 | 0.173 | 0.104 | 0.057 | 0.231 | 0.174 |
| (vi) Cynipoidea | 0.009 | + | 0.066 | 0.171 | 0.104 | 0.057 | 0.259 | 0.202 | 0.007 | + | 0.065 | 0.160 | 0.095 | 0.058 | 0.233 | 0.175 |
| (iv) Siricidae + Xiphydriidae | −0.038 | − | 0.020 | 0.293 | 0.273 | 0.058 | 0.253 | 0.194 | −0.050 | − | 0.009 | 0.239 | 0.230 | 0.059 | 0.228 | 0.169 |
| (v) Pamphiliidae + Xyelidae | — | | — | — | — | — | — | — | −0.048 | − | 0.011 | 0.241 | 0.230 | 0.058 | 0.228 | 0.169 |
| Aculeata | 0.024 | + | 0.078 | 0.303 | 0.224 | 0.054 | 0.245 | 0.191 | 0.029 | + | 0.083 | 0.239 | 0.156 | 0.054 | 0.226 | 0.172 |
| Vespina | 0.035 | + | 0.060 | 0.225 | 0.164 | 0.026 | 0.272 | 0.247 | 0.033 | + | 0.060 | 0.225 | 0.165 | 0.027 | 0.275 | 0.249 |
| Hymenoptera | — | | 0.058 | 0.253 | 0.195 | — | — | — | — | | 0.058 | 0.228 | 0.170 | — | — | — |

Summary of mean net diversification, speciation and extinction rates estimated by BAMM[45,46] and MEDUSA[116]. + indicates a higher focal net diversification rate compared to the background net diversification rate; − indicates a higher background net diversification rate than in the focal clade. Note that this does not indicate a positive or negative rate shift per se, as the mean background rate may be different from the mean rate in a particular subtree. The three clades with the highest net diversification rates are highlighted in bold font. Rate estimates are given for both topology C-1 and A-O. net. div. = net diversification rate, spec. = speciation rate, ext. = extinction rate, Δ net. div. = difference between net diversification rate in the focal clade and the background rate, int. = internal. All rates are rounded to the third decimal. Latin and roman numbers refer to shifts as labeled in Fig. 3, Supplementary Figs. 20 and 22, and Supplementary Data 9. Source data for this Table can be found in the Dryad repository at https://doi.org/10.5061/dryad.08kprr54m (folder 3.2).

**Table 3 | Summary of diversification rates and shifts estimated with MEDUSA**

| Clade | Regime | Ancestral | Age (Ma) | Rate | Shift |
|---|---|---|---|---|---|
| *Topology C-1* | | | | | |
| Background rate | i | | | 0.028 | |
| Aculeata minus Chrysidoidea | ii | i | 123.2 | 0.081 | + |
| Bees minus Melittidae | iii | ii | 66.1 | 0.151 | + |
| Siricidae + Xiphydriidae | iv | i | 236.7 | 0.021 | – |
| core Cynipoidea | v | vi | 95.8 | 0.077 | + |
| Cynipoidea | vi | i | 129.6 | 0.031 | + |
| *Topology A-0* | | | | | |
| Background rate | i | | | 0.029 | |
| Aculeata minus Chrysidoidea | ii | i | 124.8 | 0.081 | + |
| Bees minus Melittidae | iii | ii | 64.4 | 0.155 | + |
| Siricidae + Xiphydriidae | iv | i | 239.9 | 0.020 | – |
| Pamphiliidae + Xyelidae | v | i | 239.0 | 0.022 | – |

Rate regimes, ancestral regimes, and ages of rate shifts inferred by MEDUSA[116] analyses based on topology C-1 and A-0. + indicates an increase in rate, – indicates a decrease in rate. Refer to Supplementary Fig. 22 for a graphical illustration of shifts. Source data for this Table can be found in the Dryad repository at https://doi.org/10.5061/dryad.08kprr54m (folder 3.4).

presently largely angiosperm-feeding (~85%[58]) and estimated in our analyses with a crown-group origin around 170 Ma.

Despite the more recent transitions to secondary phytophagy (and predation), the single appearance and long dominance of parasitoidism in Hymenoptera is striking. Parasitoidism does occur in other holometabolous insects, namely in the orders Diptera, Coleoptera, Lepidoptera, Trichoptera, and Neuroptera[24,59], but Hymenoptera harbors the majority of parasitoid diversity (75–80%[24]). In Diptera and Coleoptera, the other orders with substantial parasitoid members, parasitoidism has evolved repeatedly[24,60] but probably much more recently, at least in Diptera[61]. Our timeline of parasitoid diversification in Hymenoptera is generally older than scenarios suggested by the fossil record alone, which depict an expansion of insect parasitoid clades only starting in the late Early Jurassic and Early Cretaceous ("Mid-Mesozoic Parasitoid Revolution"[59]). Yet our data imply a parasitoid regime already in full swing in the late Triassic. According to our current understanding of insect diversification[62], parasitoidism was likely already dominant in Hymenoptera when many of their current primarily holometabolous host groups (e.g., Diptera and Lepidoptera) began to diversify. Thus, diversification in parasitoid Hymenoptera may have been mediated by both the long history and antiquity of the behavior, as a consequence of the longer timeframe for speciation events to take place and the opening of new niche space and resources by tracking host lineage diversification.

What roles have key innovations played in the diversification of Hymenoptera? Several clades are highlighted with a history of increased diversification rates across our analyses, related to some degree to the evolution of the four traits under investigation. An association of the wasp waist with diversification in Hymenoptera in combination with a hidden state was supported by HiSSE analyses, and both combinations of observed and hidden states show a significant increase in net diversification rates in apocritan waisted wasps. This confirms that the diversification patterns are largely driven by the hidden state(s), as there was no support from MEDUSA or BAMM analyses for a rate shift at the origin of Apocrita. Interestingly, BAMM and MEDUSA analyses recovered negative shifts or net diversification

rate decreases in non-apocritan clades (i.e. Tenthredinoidea, Siricidae + Xiphydriidae, Pamphiliidae + Xyelidae), suggesting that the disparity in species richness we see between apocritan and non-apocritan lineages today may be due at least in part to a slowdown of diversification and higher extinction in the latter. It is possible that the rate increase associated with hidden states in the presence of the wasp waist is driven by the fact that lineages with this trait contain all the inferred rate shifts, while the absence of the wasp waist is restricted to lineages with low present-day diversity. Despite the wasp waist being undeniably a major morphological innovation in Hymenoptera[29], there is thus no evidence that this character immediately accelerated the diversification of Apocrita.

Trait-dependent diversification was not supported for the stinger, although a positive diversification rate shift in non-chrysidoid Aculeata received support across BAMM and MEDUSA analyses, and net diversification in this clade is elevated compared to the background rate and the general rate in Aculeata. This observed rate shift is only imperfectly aligned with the origin of the stinger, excluding chrysidoids, which form a grade respective to all other aculeates. While the stinging apparatus evolved in the ancestor of Aculeata, this complex unit is composed of structures that have undergone several modifications. For example, all aculeates except chrysidoids have the third valvula of the sting shaft subdivided, a putative synapomorphy for this group, potentially leading to greater stinging precision[63]. Moreover, the loss of the egg delivery function in favor of exclusive delivery of venom by the ovipositor may not have been simultaneously fixed in the MRCA of aculeates, as there are several members of Chrysidoidea that still use the ovipositor to deliver eggs (e.g., Dryinidae, Embolemidae, Sclerogibbidae, and Chrysididae[64,65]). Some of the innovative functions of the stinger may, therefore, only have had an impact on accelerating diversification once a series of modifications were completed. In any case, support for trait-independent diversification does not necessarily indicate that the aculeate stinger had no effect on the diversification rate at all, as it could be that elevated rates in other non-aculeate clades dilute the signal recovered for the stinger in statistical analyses.

We found indication for an association of parasitoidism with net diversification rate in combination with hidden states, but HiSSE results show an initial decrease in net diversification rate in Vespina rather than an increase. Parasitoidism can be viewed as a highly specialized strategy of carnivory (including predation), for which a character-independent model was supported. This may suggest the evolutionary significance of parasitoidism per se (as opposed to any form of carnivory) for the diversification in Hymenoptera. However, support for a hidden state model for parasitoidism over a character-independent model was not overwhelming, and there was no support by MEDUSA or BAMM analyses for a rate shift coinciding with the evolution of parasitoidism in Vespina. Wiegmann et al.[66] investigated parasitism (this broader definition also includes groups that do not kill their host) as a driver of diversification across insects using sister-clade comparisons but found no consistent association with increased diversification. They suggested that the specialization necessary for parasites to adapt to their host may actually decrease their evolutionary potential and negatively influence diversification[66]. In the case of Hymenoptera, our results indicate that parasitoidism evolved circa 230 Ma. This may have been too early for the group to have immediately benefited from the dramatic diversification of other holometabolous orders, which seems to have taken place in the last 150 million years[61,67–69]. Hence, the evolution of parasitoidism did not necessarily coincide with an abundance of available niches (in the form of host species) that would spur rapid adaptive radiation and lead to an increase in diversification rate. Instead, there may have been a lag time between the origin of parasitoidism and the diversification of parasitoid hymenopterans until their primarily holometabolous hosts became abundant. Such a delay in response to diversification has been

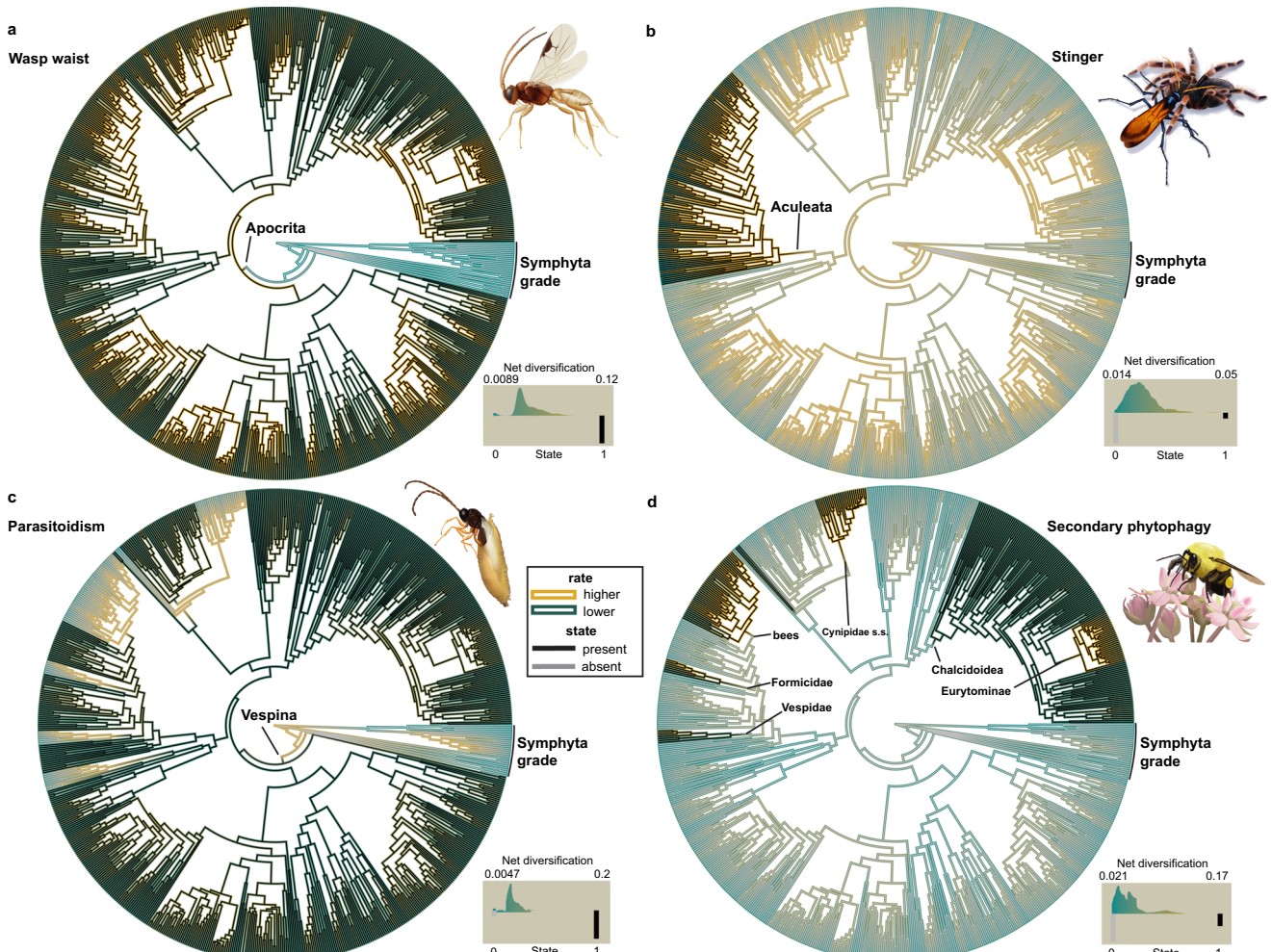

**Fig. 4 | Trait-dependent diversification analyses.** We tested 30 models of trait-dependent and trait-independent diversification in the HiSSE framework[48] for the four putative key innovations and plotted net diversification rates and state reconstructions from the best-fitting model onto the Hymenoptera phylogeny. All results shown were based on topology C-1. Inner branch colors represent the presence/absence of the respective states (black = presence; gray = absence); outer branch colors represent the net diversification rate (highest = gold, lowest = dark cyan). Clades discussed in the context of the results are labeled. The histograms in the lower right of each panel represent the distribution of net diversification rates associated with the observed states (0 = absent, 1 = present). **a** Wasp waist, model:

HiSSE−full, irreversible states; **b** Stinger, model: CID-4−9 distinct transition rates; **c** Parasitoidism, model: HiSSE−full, all free parameters; **d** Secondary phytophagy, model: HiSSE−full, irreversible states. Best-fitting models for the wasp waist (**a**), parasitoidism (**c**), and secondary phytophagy (**d**) are trait-dependent models that suggest the association of a hidden state with the analyzed traits influencing diversification in Hymenoptera. The best-fitting model for the stinger (**b**) is a trait-independent model. For detailed information on methodology, results, and model scores, refer to the main text, Supplementary Methods, and Supplementary Data 10−12 and 16−17. Source data for this figure can be found in the Dryad repository at https://doi.org/10.5061/dryad.08kprr54m (folder 3.3).

observed in other systems as well[10], for example, in the evolution of $C_4$ photosynthesis in grasses[70].

Several rate shifts were suggested ancestral to or closely related to secondarily phytophagous clades. For instance, one of the rate shifts supported by both BAMM and MEDUSA highlights non-melittid bees as a clade with an exceptionally high net diversification rate. This rate shift is loosely associated with what could be seen as a key behavioral transition, a shift from parasitoidism to pollen collecting in bees, and confirms a result from an earlier analysis focusing on bee diversification dynamics[31]. Melittid bees are oligolectic, narrowly adapted to pollen-feeding from a few host plants, while many other bees are adapted to a wider host breadth (polylecty) and, thus, a broader ecological niche. Given the sister-group relationship of Melittidae to the rest of the bees, oligolecty is suggested to be the ancestral condition[71]. Murray et al.[31] suggested that this expansion of the host plant niche led to increased diversification in non-melittid bees. Similar to a hypothesized association of an advanced stinger with diversification in non-chrysidoid aculeates, polylecty could be seen as an advanced form of

pollen-collecting in bees, the secondary innovation hereby driving diversification. Most analyses indicated some support for a positive diversification rate shift within Cynipoidea and particularly Cynipidae s.s., a group of gall-forming wasps, while another rate shift is suggested by BAMM within Eurytominae (Eurytomidae, Chalcidoidea), a group with multiple independent origins of phytophagy, commonly as gall associates (inducers or secondary feeders) or seed feeders. Both Cynipidae s.s. and Eurytominae were also estimated with very high net diversification rates (0.112−0.115 and 0.173−0.185, respectively). However, while our taxon sampling for cynipoids and bees was balanced and representative of species diversity, we cannot discount that a possible bias drives the accelerated diversification rates seen in Eurytominae within the more sparsely sampled Chalcidoidea, and the validity of the eurytomine rate shift, therefore, remains unclear.

The location of several diversification rates shifts ancestral to or within secondarily phytophagous clades, as well as their congruence with trait-dependent and tip diversification patterns (i.e. HiSSE and MiSSE), suggest that this life strategy played a major role in

## Table 4 | Trait-dependent diversification estimates within the HiSSE framework

| | Topology C-1 | | | Topology A-O | | |
|---|---|---|---|---|---|---|
| | Best-scoring model | AICc best model | Δ AICc 2nd best model | Best-scoring model | AICc best model | Δ AICc 2nd best model |
| *Innovations* | | | | | | |
| Stinger | CID-4–9 distinct transition rates | 7494.1 | 32.3 | CID-4–9 distinct transition rates | 7511.0 | 44.1 |
| Wasp waist | HiSSE—full parameters, irreversible | 7464.1 | 8.6 | HiSSE—full parameters, irreversible | 7499.8 | 0.9 |
| Parasitoidism | HiSSE—full parameters | 7549.3 | 6.6 | HiSSE - full parameters | 7583.4 | 2.7 |
| Secondary phytophagy | HiSSE—full parameters, irreversible | 7519.2 | 15.4 | HiSSE - full parameters, irreversible | 7524.6 | 39.3 |
| *Fundamental feeding strategies* | | | | | | |
| Phytophagy | HiSSE—full parameters | 7531.8 | 25.0 | CID-4–9 distinct transition rates | 7573.8 | 1.7 |
| Carnivory | CID-4–3 distinct transition rates | 7517.3 | 7.8 | CID-4–3 distinct transition rates | 7546.5 | 6.9 |

Summary of trait-dependent diversification estimates for the four putative key innovations (wasp waist, stinger, parasitoidism, secondary phytophagy) and the fundamental feeding strategies phytophagy and carnivory using 30 models within the Hidden State Speciation and Extinction (HiSSE) framework[48]. We list here only the best-scoring model and the difference in AICs scores to the second-best model. For a detailed list of results and model specifications, refer to Supplementary Data 10. Source data for this Table can be found in the Dryad repository at https://doi.org/10.5061/dryad.08kprr54m (folder 3.3).

diversification. Phytophagy has repeatedly been suggested to promote diversification in insects[12,15,16,68,72,73]. Yet, the comparatively lower diversity of phytophagous clades appeared to be a major argument against applying this hypothesis to Hymenoptera. The ancestrally phytophagous "symphytan" lineages have not diversified exceptionally (7882 vs. 144,809 described species in Apocrita at the time of writing[18]), and support for trait-dependent diversification for (fundamental) phytophagy also remained equivocal. Taken together, these results highlight the role of secondary phytophagy, but not phytophagy per se, in the more recent diversification history of Hymenoptera. This leads to the question: what adaptations do a parasitoid-converted-to-phytophage possess that may confer an evolutionary advantage over a primary phytophage?

We hypothesize that secondary phytophagy enabled further success because it is derived from the common parasitoid behavior of provisioning for their offspring by laying their eggs directly on or into a food resource[21]. This basic form of parental care increases the survival of offspring and could lead to a decreased extinction rate in the provisioning lineages compared to non-provisioning groups, although in our reconstructions, only one of the two hidden state combinations was associated with a lower extinction rate. While some forms of parental care, such as egg guarding, also occur in extant "symphytan" lineages (e.g., Argidae and Pergidae[74]), parasitoids are particular in that they generally provide all resources (i.e., food and shelter or a nest) needed for the development of their offspring. Secondarily phytophagous hymenopterans such as bees, cynipids, or eurytomines may have retained this strategy from their parasitoid ancestors. This may have allowed secondarily phytophagous groups to more efficiently explore new plant-based food resources (such as pollen or concealed feeding in plant galls), providing a level of larval provisioning not previously possible. Diversification of these clades may have accelerated via a combination of decreased extinction and increased speciation rate as a response to the adaptation to these new trophic niches and the escape from the competition with parasitoid groups. In this context, parasitoid behavior potentially could be considered a pre-innovation or precursor (sensu Donoghue and Sanderson[10]) to the evolution of specialized phytophagous strategies such as gall-inducing or pollen-collecting, and possibly represent the unobserved, hidden states influencing diversification rate. Of course, this scenario is not supported for the aforementioned melittid bees and other rather species-poor secondarily phytophagous clades, such as pollen wasps (Vespidae: Maserinae) or *Krombeinictus*, a monotypic genus of apoid wasps; conversely, it does not fit with the absence of rate shifts in the species-rich ants and the remaining vespid wasps.

Our study applied a broad definition of the key innovation concept widely used in the macroevolutionary literature[7,9,10]; however, this interpretation has been criticized recently as being too simplistic[7,9,10]. We argue that an operational, simple definition is still desirable as a first step for macroevolutionary tests of key innovations: if a trait shows a strong correlation with a diversification rate increase in a clade, this may highlight its relevance in the evolutionary history of that lineage. However, in case of an absence of trait-dependent diversification, the reverse conclusion may be less straightforward. For example, there may be a time lag between the innovation and the increase in diversification rate, in which case statistical tests will fail to detect a correlation[10]. Confirming key innovation hypotheses when the trait under scrutiny has evolved only once or early on in the evolution of a group, as the case for several of our innovations, may also be particularly challenging from a statistical and conceptual point of view[7,48,75]. Extinction may have a considerable influence on net diversification in some clades as well, despite high speciation rates connected to innovation. Extinction rates are notoriously difficult to estimate from phylogenies[76], and some authors have cautioned against inferring diversification dynamics from timetrees altogether[77]. Recent work further suggests that the location of diversification rate shifts

may be influenced by sampling completeness and the number of alignment sites analyzed[78]. We show that our results are largely congruent and robust across data sets of different sizes and types of analyses, and careful measures were taken to minimize the effect of taxon sampling. Our diversification analyses corrected for unsampled diversity on the clade level by using a backbone sampling frequency for accounting for the few lineages (i.e. families) missing from our analyses. As these were mainly lineages with low diversity, we believe their exclusion likely had little effect on the overall diversity estimates. On the species level, we employed clade-specific taxon and trait sampling frequencies, which represent the only possible strategy for such a diverse group as Hymenoptera, for which complete phylogenetic sampling is out of reach with the current methodology. Sampling frequencies, of course, can only be as accurate as the underlying estimates of described species and family-level diversity, and it is, therefore, possible that future improvements in these estimates will warrant updated analyses.

In conclusion, our results indicate that the evolution of secondary phytophagous strategies has played a prominent role in the diversification of Hymenoptera, while the impact of the wasp waist, the aculeate stinger, and parasitoidism as direct accelerators of diversification or key innovations in the traditional sense remains unclear. We suggest that modifications or specializations of these latter three prominent characters and behaviors, rather than their first appearance, may relate to the diversification of Hymenoptera. This may be a common scenario, as many traits perceived as important innovations seemingly only had major evolutionary impacts when combined with additional adaptations. For instance, while winged flight has been traditionally considered a key innovation in insects, it was only after the evolution of wing flexing (in Neoptera) that insect diversity rapidly expanded in terms of both species diversity and niche occupation[14]. Therefore, we propose parasitoidism, the wasp waist and the stinger may be part of more complex character synergies akin to "synnovations" (sensu Donoghue et al.[10]), i.e., characters that interact synergistically with other traits to open new evolutionary pathways. Future research should focus on dissecting these traits into their functional subcomponents to relate them with biological implications. A more nuanced analysis of these subcomponents may reveal innovations that can more directly be associated with diversification events. Searching for fossils that improve knowledge of divergence timing could also prove essential to better link innovations with diversification events. That parasitoidism was a superbly successful strategy in Hymenoptera cannot be disputed, given its long dominance in Vespina. Yet, among the characters assessed as potential key innovations by our study, only secondary phytophagy in the parasitoid lineages has left a discernible, direct imprint on diversification dynamics within the evolutionary history of the order.

## Methods

### Taxon sampling

We assembled a taxon set of 771 species across 94 out of 109 recognized extant families (sensu Huber[18], with modifications by Chen et al.[79], Pilgrim et al.[80], and Sann et al.[81]), belonging to all 22 recognized superfamilies within the Hymenoptera[18,80], and six non-hymenopteran outgroups. Our taxon sampling aimed for the representation of major lineages within families while sampling across the respective root nodes on the family level, covering between 0.06–50% (=1–150 representatives) of the described species diversity. While we generated UCE sequence data de novo for most taxa, some sequences have already been published in other studies by some of us: 126 aculeate wasps[38,82,83], 25 chalcidoids[84–86], 76 cynipoids[87], 26 Ichneumonidae[88–90] and 142 Braconidae[91]. We further included six representatives of other insect orders as outgroups by mining UCEs in silico from published genomes: Coleoptera (*Agrilus planipennis*), Diptera (*Aedes albopictus*), Lepidoptera (*Papilio glaucus*), Hemiptera (*Homalodisca vitripennis*),

Psocodea (*Pediculus humanus corporis*), and Blattodea (*Blattella germanica*). Supplementary Data 1 list voucher information and NCBI accession numbers for all sequences, while more detailed specimen data is provided for sequences newly released in this article. All specimens were collected with the required permits and in accordance with local regulations at the time of their collection, and vouchers have been deposited in major collections.

### UCE data collection and processing

We collected UCE data for this and allied studies using well-established library preparation and target enrichment protocols[92–94], which we summarize in the following. Most UCE laboratory work was conducted in and with the support of the Laboratories of Analytical Biology (L.A.B.) facilities of the National Museum of Natural History, Smithsonian Institution, Washington, DC, USA. Genomic DNA was extracted destructively or non-destructively (specimen retained after extraction) from whole specimens using the DNeasy Blood and Tissue Kit (Qiagen, Valencia, CA, USA), and quantified for each sample using a Qubit fluorometer (High sensitivity kit, Life Technologies, Inc., Carlsbad, CA). Between <5 ng and 1364 ng DNA was sheared for 0–60 s (amp = 25, pulse = 10) to a target size of approximately 250–600 bp by sonication (Q800, Qsonica Inc., Newtown, CT), depending on prior DNA degradation. A modified genomic DNA library preparation protocol (Kapa Hyper Prep Library Kit, Kapa Biosystems, Wilmington, MA) was applied to incorporate bead-based cleanup steps[95] and a generic SPRI substitute[96] as described by Faircloth et al.[82], as well as TruSeq-style adapters during adapter ligation[97]. Libraries had post-PCR concentrations from 0.1 to 102 ng/μL. Library input statistics are provided in Supplementary Data 1. Groups of eight to ten libraries were combined at equimolar ratios, and each pool was enriched using a set of custom-designed probes (MYcroarray, Inc., now ArborBiosciences, Ann Arbor, MI) targeting 2590 UCE loci in Hymenoptera[98] (now sold as predesigned panel myBaits UCE Hymenoptera 2.5Kv2P). The pooled libraries were sequenced using several lanes of 125 bp paired-end sequencing on an Illumina HiSeq 2500 instrument. For 36 chalcidoid taxa, UCE data were generated at Center de Biologie et de Gestion des Populations (CBGP), Montpellier, France using the myBaits UCE Hymenoptera 1.5Kv1 panel[82] with similar protocols[84], and sequenced on an Illumina MiSeq instrument. UCE sequences for the six non-Hymenoptera outgroup taxa were captured from genome assemblies published on NCBI (www.ncbi.nlm.nih.gov, see Supplementary Data 1 for accession numbers), using scripts provided within the PHYLUCE package v.1.5.0[99]. We followed the tutorial "harvesting UCEs from genomes" (https://phyluce.readthedocs.io/en/latest/tutorials/tutorial-3.html), except we reduced the stringency of the minimum coverage parameter of the phyluce_probe_run_multiple_lastzs_sqlite script (-minCov = 50) and used the myBaits UCE Hymenoptera 2.5Kv2P panel as input. We captured between 235 and 634 UCE loci for outgroup taxa.

All UCE data were processed using scripts within the PHYLUCE package[99]. We first trimmed the demultiplexed FASTQ data output for adapter contamination and low-quality bases using Illumiprocessor v2.0.7[100], based on the package Trimmomatic v0.32-1[101]. We assembled the cleaned reads using the program Trinity (version trinityrnaseq_r20140717)[102] and a wrapper script (phyluce_assembly_assemblo_trinity.py). At this step, we combined assemblies from previously published sequences (including outgroups) with these newly generated assemblies and aligned these to enrichment baits using phyluce_assembly_match_contigs_to_probes.py (min_coverage=50, min_identity=80), thereby creating a relational sqlite database containing the matched probes. Sequence quality statistics were calculated for Trinity contigs and UCE contigs using phyluce_assembly_get_fastq_lengths and are summarized in Supplementary Data 1. We aligned the sequence data for individual UCE loci using MAFFT v7.130b[103] through phyluce_assembly_seqcap_align.py (settings: max_divergence=0.2, min-length=100, -no-trim). We performed

internal trimming using Gblocks v0.91b[104] and a phyluce wrapper script (phyluce_assembly_get_gblocks_trimmed _alignment_from_untrimmed.py), with the relaxed trimming settings b1 = 0.5, b2 = 0.5, b3 = 12, b4 = 7. From 2590 trimmed UCE alignments, we prepared three different matrices using sets of loci recovered for at least 50%, 60%, and 70% of taxa for further analyses using the script phyluce_align_get_only_loci_with_min_taxa.py. 1118, 767 and 447 UCE loci were retained for analysis in a 50%, 60%, and 70% matrix, respectively. We have outlined the above deviations from default parameters for UCE processing only; for detailed documentation of this bioinformatics pipeline refer to https://phyluce.readthedocs.io/en/latest/tutorials/. Supplementary Fig. 1 provides a flowchart-style overview of our data treatments and data sets.

To extract protein-coding loci from our captured loci, we followed a published pipeline[105] and used the required script available at https://github.com/marekborowiec/uce-to-protein, which uses BLASTX (https://blast.ncbi.nlm.nih.gov/Blast.cgi) to match unaligned UCE sequences to a reference protein database. Filtering of the protein-coding UCE loci retained 324 loci present in at least 50% of taxa, which were used for subsequent phylogenetic analyses.

## Phylogenetic inference

We partitioned the nucleotide data matrices using the Sliding-Window Site Characteristics Entropy (SWSC-EN) algorithm[106] and PartitionFinder2 v2.1.1[107] employing the *r cluster* algorithm[108]. The resulting partitioned nucleotide data matrices (nuc-50%-SWSC, nuc-60%-SWSC, and nuc-70%-SWSC), as well as unpartitioned versions (nuc-50%-unpart, nuc-60%-unpart, and nuc-70%-unpart) were analyzed with Maximum Likelihood (ML) best-tree and ultrafast bootstrap searches in IQ-TREE v1.6.10[109] employing model selection for unpartitioned matrices while implementing a GTR + G model for data subsets in partitioned matrices. The 324 protein-coding loci were analyzed both as nucleotide matrix (prot-nuc-unpart) and translated to amino acids (prot-AA-unpart), employing model selection in IQ-TREE but no data partitioning. We kept third codon positions in the nucleotide data set to ensure comparability with the full data sets. All analyses were rooted using the outer, non-holometabolous outgroup (Blattodea: *Blattella germanica*). We calculated several alignment statistics (e.g., alignment length, amount of missing data, number of parsimony-informative sites) with AMAS[110], summarized in Supplementary Data 2.

We used the Four-cluster Likelihood Mapping (FcLM) approach[44] to test four topological hypotheses regarding the position of Ichneumonoidea and Ceraphronoidea on each of our four main data sets (nuc-50%, nuc-60%, nuc-70% and prot-AA). The four hypotheses were investigated by defining four taxon groups (specified in Supplementary Data 14 and Supplementary Methods), and FcLM analyses were performed in IQ-TREE v1.6.12 using 100,000 randomly drawn quartets. Further information on additional phylogenetic sensitivity tests and exploratory analyses, such as GC content analyses, can be found in the Supplementary Methods. Coalescence-based phylogenetic inference was not pursued after the preliminary analysis stage because some characteristics of our data set (short alignment size and high levels of missing data for individual loci) suggested a high propensity for gene estimation errors.

## Divergence dating

Divergence times were estimated using the information on twelve fossils within Hymenoptera, chosen following best practices for fossil calibrations[111], and representing the oldest and most reliable available calibration points for superfamily and family-level nodes, except for one fossil calibrating a subfamily-level node. We restricted our calibrations to these 12 fossils as they covered all major lineages and deep divergences within Hymenoptera for which confidently placed fossils were available. Such deep calibrations have been shown to increase the accuracy of divergence estimates[112]; calibrations at more shallow nodes

were unlikely to improve the age estimates, yet increase the number of parameters in the computationally challenging analyses. Supplementary Data 15 detail the implemented calibrations, the characters used for their placement, and corresponding references. In addition to the fossil calibrations, we set a soft maximum bound of 283.7 Ma for crown Hymenoptera, which represents the upper 95% CI for the age of the order estimated by Misof et al.[62] (see Table S25 in that paper). We employed approximate likelihood to estimate divergence times in mcmctree and codeml as included in PAMLv4.9[113], using the two data sets and trees that were best-supported (by ufBS and FcLM) and most frequently recovered across our phylogenetic analyses (see Supplementary Methods and Discussion for further details): 1) the nuc-50% matrix and the best ML tree resulting from SWSC-EN partitioning of this matrix (topA-0), and 2) the nuc-70% matrix and the best ML tree resulting from SWSC-EN partitioning of this matrix (topC-1). Outgroups were pruned from the tree and alignment prior to divergence time estimation. All fossil calibrations were implemented as soft minima, except for the calibration on the root node on which we placed soft minimum and maximum bounds, and using default settings (heavy-tailed density based on a truncated Cauchy distribution with an offset $p = 0.1$, a scale parameter $c = 1$, and a left tail probability of $α = 0.025$ creating the soft bound). We set samplefreq=10 and $n$ samples = 2,000,000, resulting in a potential chain length of 20,000,000 states (sample freq × $n$ samples). We set up four separate runs for each data set and periodically checked progress and convergence parameters by visualizing mcmc convergence and effective sample sizes (ESS) using TracerV1.7.1[114]. Runs were stopped at 1,732,010–4,632,960 states once most parameters reached ESS values above 200 (excluding burnin). Most parameters well exceeded the ESS threshold in individual runs, but due to the large number of parameters to estimate (>750), a few only reached the threshold after combining results from the four runs. 731,251 samples for topC-1 and 1,180,127 for topA-0 were summarized across four runs each, after discarding 25–50% of samples as burnin. To evaluate the impact of our calibrations, we also performed analyses without sequence data using only the prior.

## Diversification and comparative analyses

To assess potential shifts in diversification rates over time in Hymenoptera we used a sampling fraction approach in BAMM v2.5[45,46] and the associated R package BAMMtools v2.1.7[115], as well as a taxonomic approach implemented in the MEDUSA function in the R package Geiger v2.0.7[116] in R v4.0.3. For BAMM, we created clade-specific sampling probabilities by assembling a richness matrix with the number of described species for all families of Hymenoptera included in our analyses. We predominantly used species estimates published in Huber[18], except for the following groups for which the classification in that volume was outdated or did not correspond with natural monophyletic groupings. Within Aculeata we used the family-level classifications established by Pilgrim et al.[80] and Sann et al.[81] and species diversity estimates from Branstetter et al.[117] and Pulawski's catalog[118]. Within Cynipoidea, we distinguished six monophyletic clades as identified in Blaimer et al.[87] and assigned species richness based on Buffington et al.[119]. Due to several non-monophyletic families in the Chalcidoidea (e.g., Pteromalidae) and the associated uncertainty about lineage-specific species richness, we treated this entire superfamily as one clade for the purpose of this analysis. Platygastroidea were treated similarly; the recently updated taxonomy of this group[79] could not yet be incorporated at the time of analysis. This merging and splitting of families into recognized monophyletic lineages resulted in 68 clades defined for analyses (with 13 missing). Clade-specific sampling probabilities were then calculated as proportions of sampled diversity divided by the total described species diversity in these clades and are listed in Supplementary Data 16. We also applied a backbone sampling fraction (68 sampled/81 recognized clades = 0.8395) to account for the

unsampled clades in the analyses. Sampling and parameter choice are further discussed in the Supplementary Discussion. We used the two chronograms (topA-0 and topC-1) generated in the dating analyses for two sets of BAMM analyses. Our analyses were configured using the function "setBAMMpriors" within BAMMtools to obtain appropriate priors for speciation-extinction analyses as outlined in the guidelines in the BAMM documentation (http://bamm-project.org/); see Supplementary Methods for details on priors used. Our runs included four mcmc chains with a length of 200 million generations, sampling every 10,000 generations, and discarding a burnin of 10%. We confirmed that ESS values were appropriate (>200) and used "compute-BayesFactors" to identify the best-supported model of rate shifts in our data. Results were analyzed and plotted with various functions in BAMMtools to infer mean phylorate plots (plot.bammdata), best shift configurations (getBestShiftConfiguration), credible shift sets (credibleShiftSet), and cumulative shift probability trees (cumulative-ShiftProbsTree). Cumulative shift probabilities were displayed within a range of 0.97–1.00, as less stringent cutoff values support shifts leading to almost every major clade. We calculated mean speciation and extinction rates for specific clades for which rate shifts were indicated (both by BAMM and MEDUSA, see below).

A taxonomic method, in which clades are simply collapsed to terminal lineages of equal rank, may be more appropriate for incomplete sampling in diversification analyses[120] and circumvent some of the problems raised in the debates about BAMM[121,122]. The MEDUSA algorithm first fits a single diversification model (the background rate) to the entire data set, and then adds single breakpoints (i.e. shifts) in the diversification process in a step-wise fashion, so that different parts of the tree are allowed to evolve with different parameter values and have different rate regimes[47]. We implemented this analysis using clade-level chronograms for both topA-0 and topC-1, which we created by dropping all tips except one representative for each of the 68 clades also designated in BAMM analyses. The species richness matrix assembled for BAMM, composed of the described species diversity for each clade (Supplementary Data 16), was further designated to assign diversity estimates to the clade-level tree for this analysis. We used an AICc threshold of 3.760758 (computed automatically by MEDUSA) as a stopping criterion for the algorithm, at which further breakpoints in the diversification process are not added.

To investigate the evolution of parasitoidism in Hymenoptera, we used the same clade-level approach to integrate the entire character diversity within each group, thus also accounting for taxa missing from our phylogeny and avoiding bias by over- or underrepresentation of particular states. We inferred (1) the presence or absence of parasitoidism as a binary trait, and (2) the evolution of hymenopteran life strategies on a more detailed level, assigning the four categories parasitoidism, primary phytophagy (including xylophagy and mycophagy), secondary phytophagy (i.e. gall-inducing, pollen collecting), and predation. We also (3) contrasted carnivory (parasitoidism and predation combined) with phytophagy and secondary phytophagy in an analysis comprising three trait categories, though the results shown here focus on parasitoidism-centered analyses. Clades were assigned to one of these categories based on the life strategy exhibited by most (>80–95%) members. We allowed polymorphism for Chalcidoidea, Formicidae, and Vespidae, for which such a fully binary choice would not be representative of the group. Our rationale is further described in the Supplementary Discussion; see also Supplementary Data 16 for trait coding and the references used to score the biology of each group. We used the rayDISC function in the R package corHMM v2.1 in R v4.0.3 to reconstruct ancestral states using the reduced clade-level phylogenies created for MEDUSA analyses. We performed reconstructions under the "equal rates" model (ER) and the "all rates different" model (ARD) and compared the fit of these models with a likelihood ratio test.

We tested for state-dependent diversification associated with the four putative key innovations (wasp waist, stinger, parasitoidism, secondary phytophagy) and related traits (carnivory, phytophagy) in Hymenoptera using the HiSSE (Hidden State Speciation and Extinction) framework and associated R package[48] in R v4.0.3. The HiSSE framework has been developed to overcome some of the shortcomings of the SSE models (e.g., BiSSE[123] or MuSSE[124]), which is a benefit of the more complex null models applied in this approach[48,125,126]. HiSSE models incorporate hidden states, which are unobserved traits that influence the diversification rate together with the observed traits[48]. In a HiSSE model where the focal states are, for example, 1 = parasitoidism present and 0 = parasitoidism absent, the diversification parameters of the hidden character will be modeled as a second character with states A and B. We decided on this approach rather than employing the Multistate Hidden Speciation and Extinction model (MuHiSSE[127]), for example, as we specifically wanted to evaluate support for each trait as a key innovation independently. This framework addresses concerns about the false positive inference of state-dependent diversification when applying less complex models such as BiSSE[126,128]. We compared 30 models in the HiSSE framework, using the full set of models tested by Beaulieu and O'Meara[48] plus six models suggested in the HiSSE documentation and a study assessing the diversification of Squamates[129]. We tested the fit of our data to a full HiSSE model with unconstrained parameters and 17 subsets with various constraints on transition and diversification rates. Four models excluded hidden states and modeled trait-dependent diversification in a BiSSE-like fashion. Eight character-independent null models, or CID-2 and CID-4 models (sensu Beaulieu & O'Meara[48]), were further tested, also including an extension of the currently implemented CID-4 models in the HiSSE package allowing for nine transitions rates[129]. We performed all analyses using the full phylogenies for topA-0 and topC-1 (same species-level trees as used for BAMM) but assigned character states based on the clade level. Calculating the occurrence of these traits across our sampling of Hymenoptera, we employed sampling fractions for character states as proportions of sampled vs. unsampled trait occurrences; these are specified in Supplementary Data 17 and the Supplementary Methods. Specifications of models and parameters are given in Supplementary Data 10. Since the HiSSE function does not allow for polymorphic states or missing data, we resolved all polymorphisms to a present state, and taxa with uncertain states were coded with the state present if that was the case for their closest relatives. This follows the logic that a state can generally be counted as present in a clade even if not displayed in every taxon, while closely related taxa have a higher probability of sharing the same traits. The diversification rate and trait reconstruction results were plotted and summarized for the best-scoring models.

We also employed a recent direct extension of the HiSSE framework in the same R package (but using v2.1.9 in R v4.2.2), the MiSSE (Missing State Speciation and Extinction) model. MiSSE executes a trait-free version of the HiSSE model and thereby focuses only on the impact of unobserved, hidden states on the diversification dynamics of a clade[130]. We automated the process of fitting MiSSE models by using the "MiSSEGreedy" function and possible.combos=generateMiSSEGreedyCombinations(), using the default for stop.deltaAICc=10 and a sampling fraction of 0.005 based on our sampling of Hymenoptera (765 of 152,691 described species). We performed analyses using both topA-0 and C-1 topologies and summarized and plotted tip rates for the best-scoring model across both phylogenies.

## Reporting summary

Further information on research design is available in the Nature Portfolio Reporting Summary linked to this article.

## Data availability

The raw sequence reads newly generated in this study have been deposited in the NCBI Sequence Read Archive under BioProject accession code PRJNA811764 and PRJNA632862. The UCE sequence data from prior publications used in this study are available under BioProject accession codes PRJNA379583, PRJNA248919, PRJNA495844, PRJNA814466, PRJNA606284, PRJNA647791, PRJNA625490, and PRJNA473845; the genome assemblies used in this study are available under accession codes GCA_001444175, GCA_000699045, GCA_000762945, GCA_000696855, GCA_000006295, and GCA_000931545. The raw sequence data can also be accessed using individual accession numbers given in Supplementary Data 1. Source data for this study (assembled contig files, data matrices, tree and log files for phylogenetic analyses; input and output files for FcLM; R code and input and results files for comparative analyses) are available in the Dryad repository at https://doi.org/10.5061/dryad.08kprr54m[131]. Information on the location of voucher specimens is provided in Supplementary Data 1.

## Code availability

No new custom code is published with this article. We direct the reader to the Dryad repository accompanying this article (https://doi.org/10.5061/dryad.08kprr54m) which includes details for all code implemented in our analyses.

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

## Acknowledgements

Laboratory work was conducted in and with the support of the L.A.B. facilities of the National Museum of Natural History. We acknowledge general support for molecular work from L.A.B. staff and thank Michael Lloyd for assistance in laboratory methods. Many thanks to Taina Litwak for the illustrations and Matt Bertone for the images of wasps. Phylogenomic analyses utilized the Smithsonian Institution High-Performance Cluster (SI/HPC). Mention of trade names or commercial products in this publication is solely for the purpose of providing specific information and does not imply recommendation or endorsement by the USDA. USDA is an equal opportunity provider and employer. This study was supported by the National Science Foundation (DEB-1555905, S.G.B.) and a Smithsonian Institute for Biodiversity Genomics and Global Genome Initiative grant (S.G.B., M.L.B., and B.B.B.), as well as a grant from the GGI Peer-Review Awards Program (B.F.S.). R.R.K., M.W.G., and M.L.B., as well as part of the sequencing effort, were funded by SEL-ARS. B.B.B. was supported by the Museum für Naturkunde, Berlin, during the analyses and writing stages of this project. J.Y.R. and AC were supported by the INRAe SPE department. B.F.S. was funded by a GGI Peter Buck Postdoctoral Fellowship (Smithsonian Institution) during much of this work. E.J.T. was supported by the Florida Department of Agriculture and Consumer Services, Division of Plant Industry.

## Author contributions

B.B.B., S.G.B., and M.L.B. conceived the study, with significant input from all other authors. R.R.K., B.F.S., M.W.G., M.L.B., E.J.T., I.M., D.R.S., J.Y.R., and A.C. provided samples. S.G.B., M.L.B., and M.W.G. provided funding or resources. B.B.B., B.F.S., J.Y.R., and A.C. generated ultraconserved element data. B.B.B., B.F.S., and A.C. processed data, and B.B.B. performed phylogenomic and macroevolutionary analyses. B.B.B. wrote the initial paper draft, and all other authors contributed to revising subsequent drafts of the paper and approved the final version.

## Funding

## Competing interests

The authors declare no competing interests.
