## [Peer Review File · Nature Communications]

Key innovations and the diversification of HymenopteraReviewers' Comments:

Reviewer #1:

Remarks to the Author:

The manuscript by Blaimer and coauthors presents a) a more comprehensive phylogeny of Hymenoptera than preceding studies, b) new divergence time estimates, and c) an analysis of whether or no specific traits (i.e., parasitoidism, wasp waist, stringer, and secondary phytophagy) possibly impacted the diversification rate in this mega-diverse insect order.

The backbone phylogeny and the divergence times of Hymenoptera have been the subject of various phylogenetic studies published during the past five years. What has been missing is a credible analysis of where in the phylogeny of Hymenoptera shift in diversification rates occurred and what factors (key innovations) could have fostered these shifts. I consider such an analysis of major interest, and the results possibly be included in text books.

While the manuscript aims to provide the above analysis of key innovations and diversification rates, it has various severe shortcomings.

One major problem that I see concerns the experimental design. The authors state that a reliable phylogeny and credibly dated phylogeny are paramount for assessing diversification rate shifts. This is true to some extent, but an at least equally important factor is proper species sampling. A recent (2022) study published by Craig et al. in *Mol. Bio. Evol.* re-examined previously published studies that addressed speciation rate shifts and found species sampling and sequence variation to be the main driver of rate shifts, rather than the biological explanation put forth by the authors of the respective studies. While the present study analyzed significantly more species than previous phylogenetic studies on Hymenoptera, it does not outline based on what criteria species were selected given the aims of the study. There is some information given in the supplementary text ("Our goal was to create a well-sampled, balanced taxon sampling across the order; however, our sampling has an emphasis on the hyperdiverse lineages within the superfamilies Ceraphronoidea, Chalcidoidea, Cynipoidea, Ichneumonoidea and Platygastridae."), but it does not provide details, and the second part of the sentence suggests that imbalance in taxon sampling may just have shifted. Looking at the phylogenetic tree, phylogenetic lineages that differ in one to two orders of magnitude in their species richness, are represented by one and two species, respectively, which does not seem to be representative at all. For me, the taxonomic sampling does not seem to be specifically tailored for a species diversification analysis of Hymenoptera in general, but was focused on parasitoid non-Aprocrita (which is fine and welcome, but which is not sufficient for the aims of the study).

A major part of the results and discussion is dedicated to the backbone phylogeny and divergence times of the major lineages. However, the presented results regarding the backbone phylogeny are virtually identical to the ones published in previous studies. On the other hand, results that differ significantly, such as the early divergence of sawfly lineages, are not addressed in the main text. The presentation of the ancestral state reconstruction is also for the most part repetitive (e.g., that parasitoidism evolved in the most recent common ancestor of Vespina). The extended phylogenetic sampling holds a lot of information, but the authors decided to present almost exclusively confirming results. The divergence time estimates differ from those given in previous studies. The question is why: it is very well possible that the estimates presented in the present study are more accurate. However, the authors do not assess the credibility of different estimates. All they state is that the taxonomic sampling and the calibration strategies differed from those in previous studies (which is true, but the different studies also used different fossils to calibrate the trees and they used different types of data; none of which is or discussed mentioned). If the divergence times are important and a major result of this study, I think the authors should put significantly more effort into working out what causes the differences and what estimates are more credible. The paper otherwise presents just another set of estimates and the reader has to decide which one to trust more.

Overall, I found the phylogenetic part of the study not being well presented, as it focusses on verifying hypotheses that have been widely accepted by now. The part on divergence time estimates is not sufficiently worked out. And the divergence rate estimate part suffers from the problem that the taxonomic sampling does not seem to be specifically tailored for such an analysis and it thus bears the risk of being possibly misleading. I found the last sentence in the supplementary discussion quite illuminating in this regard "In the light of these caveats, we regard our estimates as the best achievable for hymenopteran diversification at the moment, but improved models for estimating diversification rate or an expanded phylogeny for the group may result in different conclusions."

I therefore cannot recommend publication of the manuscript in its current form in the journal Nature Communications.

Specific comments

Line 25: Why are sawflies not mentioned?

Line 25: The statement "one of the most diverse animal lineages" is unscientific, unless the basis for the comparison (geological age of the lineage or the taxonomic rank) is specified. Every superordinated lineage is more diverse, and this results in many more lineages being more species rich than Hymenoptera.

Line 28: I do not think that it is possible to test whether the presence/absence of a trait historically influenced (= causal relationship) the diversification rate. All that can be done is to assess whether there is a correlation that suggests that there could have been a causal connection. Please rephrase.

Line 29: "apocritan wasps", better write Apocrita, because Apocrita includes bees.

Line 29: "aculeate Hymenoptera", better write Aculeata, as Aculeata are per definition Hymenoptera and hence Hymenoptera is redundant.

Line 31: I find the expression "parasitoid regime" awkward. Why not parasitoid lifestyle?

Line 34: "hymenopteran diversification rate", better "diversification rate of Hymenoptera".

Lines 39–40: I wonder whether this bold statement is justified. Comparing diversification patterns makes in my eyes only sense when accepting common descent of organisms and when correcting for the available time for diversification. The basis for comparing diversification was set 166 years ago: less than two centuries ago. If diversification patterns were legitimately compared and discussed before, please provide the corresponding reference(s).

Line 47: "earth", better "Earth", more explicitly referring to the planet

Line 53: "(Psocodea13)", better "(Psocodea)13", as the reference refers to the entire statement, not only on the information in parentheses

Line 54: "(Hymenoptera: Siricidae11)", better "(Hymenoptera: Siricidae)11", as the reference refers to the entire statement, not only on the information in parentheses.

Line 58: "generalists, such as many ants and social wasps". This is too unspecific and is even misleading. Aculeata includes various lineages that are kleptoparasitic and that cannot be considered generalists by any means. Most ants and social wasps are considered predators.

Lines 65–81: The order of the two key innovations should be reversed, as the wasp waist evolved before the stringer.

Line 73: Sawflies do not represent a natural group and hence the use of the old taxonomic name for the group should be put in parentheses: "Symphyta".

Lines 80–81: "This character [the wasp waist] was potentially foundational for enabling both parasitoidism and stinging.". I find this statement misleading. The parasitoid lifestyle is ancestral in Apocrita and in Orussoidea. Given the widely accepted sister group relationship of the two lineages, the most parsimonious explanation is that parasitoidism had already evolved in the ancestor of the two groups (Vespina) and hence BEFORE the evolution of the wasp waist. Insinuating that the wasp waist represents an evolutionary innovation that was necessary for parasitoidism to evolve is incompatible with all evidence gathered during the last decades. Whether the wasp waist represents a character that was necessary for a stinger to evolve is certainly possible, but in my opinion fruitless speculation. The stinger evolved only once, in a subordinated lineage of Apocrita. Given these data, I do not see how it could possibly be tested whether a string could have only evolved in wasps with a

wasp waist. And if a hypothesis cannot be tested, it is scientifically worthless.

Lines 84–86: “, phytophagy has secondarily evolved in several groups of apocritan wasps, for example in pollen-collecting bees and in gall-inducing cynipoid wasps”. These are two prominent examples, and these examples are well covered in the sampling. However, other examples, such as in Vespidae (pollen wasps) and in apooid wasps (Krombeinictus; Krombein & Norden 1997) are neglected in the sampling and completely ignored in the discussion. Their consideration is important, though, when discussing whether a switch from carnivory to phytophagy generally results in an increase in the diversification rate. Chances are good that it did not result in such an increase in these two groups and it points at the fundamental problem that the success of a particular life style depended on multiple factors (e.g., additional traits, food sources, competition).

Lines 88: “colonization of new niches”, better “formation of new niches” as some niche concepts consider the species being part of the niche.

Lines 90–91: “All four innovations—the wasp waist, the stinger, parasitoidism and secondary phytophagy”. What is the logic of the order (neither synchronically nor alphabetically)? The in my eyes more logical order would be parasitoidism, wasp waist, stinger and secondary phytophagy (the order of the last two could be changed, but as bees serve as one major example of secondary phytophagy, listing stinger first is more plausible).

Lines 96–98: “Yet, a complete and robustly supported phylogeny for the order is still elusive, and the even most recent phylogenomic analyses^{27,29,30} were not able to provide clarity on the placement of some lineages, mainly due to sampling bias toward aculeate wasps. A reliable phylogeny of supra-familial relationships and their evolutionary timescale, particularly of non-aculeate Apocrita, is paramount to any study of the diversification and evolution of key innovations in Hymenoptera”. I find these statements problematic, as they do not seem informative or justified. First, a complete and robust phylogeny of Hymenoptera is impossible achieve any time soon, as there will always be species missing given the size of the taxon. So, this statement is more or uninformative. It is true that the phylogenetic positions of some major lineages are still considered unreliable. But why is this important in this context? The second sentence states that a reliable phylogeny is paramount for studying diversification pattern. However, the phylogenetic results presented in the main text are basically identical to the ones published during the past five years. And a previously reported phylogenetic uncertainty in the relationship of Ichneumonoidea and Ceraphronoidea is not resolved in the present study either. Interestingly, this uncertainty did not impact the diversification rate estimates, indicating that robust phylogenetic relationships are not per se a prerequisite for studying diversification rates. I think the problem with the entire paragraph is that does not explicitly state what the goals of the present study are and what is required for achieving these goals. Studying more genes and more species is pointless unless it serves a specific purpose. If the purpose is studying diversification pattern, then a representative sampling and a reliably dated phylogeny for this sampling is required. Unfortunately, the manuscript does not mention this and what was done to compile such a dataset. The text on the Supplement states “Our goal was to create a well-sampled, balanced taxon sampling across the order”. But what measures were taken to realize such a balanced sampling? What percentage of the species of each lineage is included and what was done to not oversample early splits (most diverged lineages)? The sentence immediately following the above one in the supplement suggests that the current sampling may be biased in a different direction: “our sampling has an emphasis on the hyperdiverse lineages”.

Line 103: “for 765 taxa”, better “from 765 taxa”

Line 129: “two important subdivisions”: what makes these two subdivisions important? Are Ichneumonoidea and Ceraphronoidea unimportant?

Lines 129–147: The paragraph presents the backbone of the main tree, which is identical to the ones

inferred in previous studies. Even the uncertainty in the phylogenetic position of Ichneumonoidea and Ceraphronoidea to each other has previously been reported. Thus, paragraph does not provide new information on the evolution of Hymenoptera. The paragraph can be condensed into a single sentence stating that the inferred backbone of the phylogeny is consistent with our current knowledge of the phylogeny of the major lineages of Hymenoptera.

Line 152: "other dominant strategies in Hymenoptera". Can you please outline what these strategies are specifically?

Lines 158–159: "The sawfly and woodwasp lineages branched off from other hymenopterans". Please rephrase, as it remains unclear what "other hymenopterans" refers to specifically.

Line 163: "Our ancestral reconstructions", better ancestral state reconstruction, as the reconstruction is not ancestral, but the states that are reconstructed.

Line 169–170: "The two main divisions within Apocrita, Proctotrupomorpha and the Aculeata + "Evaniomorpha" grade". Why are these two the main subdivisions and why are Ichneumonoidea and Ceraphronoidea not considered main divisions? What is needed to make a lineage be "main"?

Line 192: "s. s.". Most readers will not know this abbreviation, so better write *sensu stricto*. However, I expect most readers to not know the meaning of this Latin phrase either, so I suggest to better write "in the narrow sense". This applies to all subsequent instances of "s. s.", and also to instances of "s. l."

Lines 267–268: "The origin of species richness in Hymenoptera has been of acute interest^{8,39}, but phylogenetic uncertainties in the early evolution of the order have persistently prevented strong conclusions." What uncertainties and what conclusions specifically? Is this really true? The taxonomic sampling of sawflies in the here presented phylogeny is still very limited (but their diversification rates extensively discussed) and the uncertainty about how some lineages are related to each other (e.g., Tenthredinoidea relative to Pamphiloidea and Xyeloidea) has not been solved either.

Lines 269–271: "Our work targeted an increased sampling of parasitoid lineages to create an improved phylogenomic framework to evaluate the association of putative key traits with the diversification of Hymenoptera." The major novel aspect in this study is the analysis of diversification rates. Such analyses critically depend on a representative sampling of the diversity in a group. I acknowledge that previous studies had a very imbalanced taxonomic sampling. It is therefore important to outline what was done in the current study to achieve a balanced taxonomic sampling. Increasing the taxonomic sampling of parasitoid lineages could have helped reducing the imbalance, but the above statement does not state this. In fact, it could have resulted in a new imbalance. Note that a recent study (Craig et al. 2022. *Mol. Biol. Evol.*) found insufficient species sampling and paucity of sequence variation to be main drivers of speciation rate shifts in many previously published studies (rather than the proposed biological explanations).

Lines 271–273: "Many of our phylogenetic results corroborate previous findings with support that is substantially enhanced or overwhelming, while other results underline where uncertainties continue to persist". Can you be more specific? I am having trouble matching this statement with the presented results. The phylogenetic backbone data presented in the results text are identical to previously reported ones and the support for these was in at least one of these studies maximal. Only in case of the suggested sister group relationship between Ceraphronoidea and Ichneumonoidea was support previously low, and it still is in the present study. I acknowledge that the phylogeny shown in Figure 1 provides support for clades previously not well supported, but these clades are not addressed in the results at all. So why no present the new results? Of particular interest would be the earliest splits of Hymenoptera. The phylogeny shown in Figure 1 is incompatible with some of the recently published phylogenies, but this result seems to depend on what data were analyzed. This information is buried

in the Supplementary Text. It states that Pamphilioidea + (Tenthredinoidea + Xyeloidea) is found when having analyzed amino acid data only. So, what does this mean? Is the amino acid data less or more trustworthy than the nucleotide data? I find it problematic that Figure 1 suggests relationships that contradict those of other studies and that these differences are not discussed. Many readers in an interdisciplinary journal will erroneously assume that a high statistical support value indicates credibility.

Line 279: What is meant with "regime" of parasitoidism?

Line 286–289: "Likely reasons of these incongruities are differences in taxon composition between ours and the previous analyses, as well as different calibration strategies (i.e., node-dating in the present and Peters et al.27 analyses, and tip-dating in Ronquist et al.41)." These studies also used different fossils, and some of the studies analyzed different types of data (nucleotide incl. 3rd codon position vs amino acid).

Line 291–292: "Our data suggest the entire Jurassic hymenopteran fauna was dominated by a parasitoid regime possibly lasting for about 100 million years." Given that even today there are (to my knowledge) more Hymenoptera with parasitoid lifestyle than with any other life style, has the parasitoid life style not be dominant ever since?

Line 336: "diversification rate shift" In what direction?

Lines 369–371: "This may have been too early for the group to have immediately benefited from the dramatic diversification of other holometabolous orders, which seems to have taken place in the last 150 Ma49,55–57." Note that a study recently published by Oeyen et al. (2020) in Genome Bio. discusses molecular adaptation that evolved in Apocrita, but not in Orussoidea that could have fostered the diversification of parasitoid wasps.

Lines 375–376: "in the vicinity of secondarily phytophagous clades" sounds awkward. Perhaps better "closely related to clades with phytophagous species"

Lines 385–387: "Similar to a hypothesized association of an advanced stinger with diversification in non-chrysidoid aculeates, polylecty could be seen as an advanced form of pollen-collecting in bees, the secondary innovation hereby driving diversification." Is there evidence that oligolecty is indeed ancestral? If so, please provide reference. Otherwise, this is quite some speculation. Only because melitid bees are oligolectic does not imply that the ancestor of all bees was also oligolectic.

Lines 397–399: "However, while our taxon sampling for cynipoids and bees was balanced and representative of species diversity". This is important information. It should be presented earlier. However, it remains still extremely vague. How was balance and representativeness accomplished? What measures (e.g., percentage of species) were taken? Looking at the phylogeny, I do not see a balance. For example, within Chrysididae, bethylids and chrysidids are quite species rich, yet they are represented by only two species each and the species poor taxa in this superfamily with one species. This does not appear proportional and representative. Could this explain why a rate shift was only detected in non-chrysidoid Aculeata? Incidentally, why does Figure 1 list three bethylids, but the phylogenies in the supplement contain only two? A discrepancy in species counts exists in other families, too.

Line 401: Please rephrase, as "vicinity" of rate shifts sounds awkward.

Lines 405–408: "The ancestrally phytophagous "symphytan" lineages have not diversified exceptionally (7,882 vs 144,809 described species in Apocrita at the time of writing); in fact, net diversification rates estimated for Tenthredinoidea, Siricidae + Xiphydriidae and Pamphiliidae + Xyelidae estimated with BAMM were slightly negative (Table 3)". A statement on a shift in the

diversification rate at the base of the Hymenoptera phylogeny depends to my knowledge critically on the ancestral condition and hence on the speciation rate in the outgroup. However, the outgroup sampling comprises only five species, each one representing a different insect order. I therefore do not see that the outgroup sampling allows to inferring ancestral rates and hence deducing that the sawfly lineages had a lower speciation rate than expected.

Lines 410–411: “Taken together, these results highlight the role of secondary phytophagy, but not phytophagy per se, in the more recent diversification history of Hymenoptera.” Because the dataset does not allow to infer the speciation rate outside Hymenoptera, it cannot allow assessing whether the speciation rate in phytophagous sawflies is elevated or reduced. The dataset therefore cannot make a statement on whether or not phytophagy per se has an impact. The dataset is only (if at all, because of a possible imbalanced sampling of the ingroup) suitable to assess the impact on secondary phytophagy. However, the dataset focused on two lineages (bees, gall wasps) that were likely to show accelerated speciation rates based on prior knowledge. What is missing is a corresponding analysis on other lineages, such as pollen wasps, that are not known to be particularly species rich.

Line 411–413: “This leads to the question: why was phytophagy more prone to induce diversification in Hymenoptera when arising from parasitoid lineages?” The presented data do not allow to deduce this question, because of the lack of a benchmark. As outlined above, the lack of a proper outgroup sampling does not allow to qualify (increased/decreased) the speciation rate in the phytophagous sawfly lineages. The presented data at most allow to state that in those lineages that secondarily switched to phytophagy the speciation rate increased.

Line 413–414: “What adaptations does a “parasitoid converted-to-phytophage” possess that may confer an evolutionary advantage over a primary phytophage?” Same concern as in the previous comment: we lack a proper benchmark to conclude this!

Lines 415–426: I can follow the logic. What I am missing are additional hypotheses. Assuming that the speciation rate of pollen wasps is not elevated, laying eggs directly at the food source seemingly was not sufficient to increase the group's speciation rate. Life is about competition. Escaping from the competition within a guild can result in at least temporarily more available resources. What works for one group at a given time (e.g., bees) may not work for another group (e.g., pollen wasps) at a later time, because the resources are already competitively used (e.g., by bees).

Lines 429: “phytophagy ... can arise via different genetic and ecological pathways”. Please be more specific and provide references.

Lines 457–459: “That parasitoidism was a superbly successful strategy in Hymenoptera cannot be disputed given its long dominance in Apocrita and the much higher species diversity of parasitoids in the order.” The dominance of parasitoidism includes also Orussoidea, and hence Apocrita should be replaced by Vespina. I do not understand the difference in what the first and what the second part of the sentence is meant to state. Dominance means that one thing is more present than another. To my knowledge, once there were more parasitoid Hymenoptera species than non-parasitoid Hymenoptera (=> parasitoids became dominant), this dominance was retained. Hence, the long dominance has lasted till today (or am I wrong here?). It follows from pure logic that if the life style has been and is dominant that this life style is represented in most of the species. Thus, the second part of the sentence bears no additional information.

Lines 465–466: Based on what criteria were species selected to allow for a species diversification analysis. Please make transparent that the species sampling representatively depicts the species diversity in the different families (e.g., 0.5–1.0 % of the known species in a family were randomly sampled).

Lines 465–476: Please provide data on where samples were collected when and by whom.

Supplementary Table 1 only mentions voucher numbers. Transparency in sample information is important given the legal aspects related to the analysis of DNA of samples from various countries (Nagoya agreement).

Lines 494: The third codon position in nucleotide sequence data is known to bias phylogenetic results because it frequently violates the assumption of homogeneity. Why has the third codon position not been excluded, given that it also likely contributes little phylogenetic information?

Lines 509–511: Please provide details based on what criteria the fossils were selected? The table with the fossils does not outline why a fossil was assumed to be part of a clade (e.g., by providing autapomorphies).

Line 569: I noticed that the manuscript thought that only parasitoidism and predation were considered. However, these categories capture only part of the hymenopterans' life styles. A substantial number of Aculeata are kleptoparasites, either carnivorous or phytophagous ones. What was done with them?

Figure 1 and all other phylogenetic figures: I noticed that Crabronidae s. str. is polyphyletic. This is because of the placement of *Ectemnius_sonorensis_BND831* in the family Sphecidae. The same relationship was already presented by Branstetter et al. (2017) and it was heavily criticized in the community, because virtually all evidence suggests that *Ectemnius* is very closely related to *Crabro* — the type genus of the family Crabronidae. The most likely explanation is that the identity of the analyzed DNA is not correct (e.g., by mixing up of samples). The identity must thus be verified. If the placement of *Ectemnius* in the family Sphecidae is confirmed, it should be discussed in detail.

Reviewer #2:

Remarks to the Author:

In their paper "Key innovations and the diversification of Hymenoptera", the authors address different hypotheses on how certain traits could have acted as key innovations to give rise to the diversity of different hymenopteran clades.

The manuscript is well written and structured logically. It appears that much care was taken at each analysis step to run the required tests and checks to ensure all was done according to best practices and that the obtained results are trustworthy. I commend the authors for their effort there, and also for the effort taken towards transparency and replicability by depositing raw data and analysis scripts for readers to review.

Overall, I think this paper addresses a very relevant and interesting question concerning the much debated reasons behind the high diversity of some hymenopteran clades. The improved phylogeny and application of established methods in the field to test these exciting hypotheses represent meaningful advances in this field.

Despite this, I have a few major points of criticism towards this paper:

1. There is a general issue with the concept of key innovations. While they are a very appealing idea and since the rise of SSE models have stimulated a lot of exciting research into the topic, I think it has been repeatedly stated that the concept is probably overly simplistic and should only be employed with care. The authors do cite Rabosky 2017 regarding this towards the end of their manuscript, but I believe that is not sufficient to address this issue properly. The implications of this should be stated upfront (i.e. in the introduction), and their impact on the used approach and the interpretation of the results should be mentioned again wherever needed. There exists more literature on problems with the concept, as well as improved ways of thinking about it, e.g. in Donoghue 2005, Donoghue & Sanderson 2015, or maybe also Bouchenak-Khelladi et al. 2015 (for first steps towards addressing this

methodologically), and these or similar studies should be considered to address this issue.

2. Similarly, there are issues with some of the diversification methods employed, namely the SSE models. While they do hint at this somewhere, I believe the authors should cite Maddison and FitzJohn's 2015 caveat about unreplicated trait- and rate-shifts, and how one can't necessarily be distinguish whether they are causal or merely coincidental (HiSSE helps to address this, as the authors rightly point out, but not entirely). This caveat should influence the interpretation of the results too. Beyond this, Louca & Pennell (2020) have pointed out an identifiability issue with diversification models, and while they did not address those in SSE models per se, there is no reason to assume those issues wouldn't extend to those at least in some way. Different other authors have attempted solutions for this, but there is not definitive solution for those yet, so this caveat should be addressed somewhere.

3. I would generally welcome if the authors were to elaborate the suspected mechanisms/scenarios for diversification more. I am pointing this out at various points in my line comments too. It is in general very tempting (and has probably been practiced in the field for too long), to remain rather vague when explaining why a chosen trait should convey higher diversification rates to the organisms that carry it (or why not). This is understandable, as there is still a gap in understanding between microevolutionary processes of speciation and extinction, and the study of their macroevolutionary dynamics. However, I believe it is important to think more deeply about those and make sure those thoughts enter the hypotheses posed. Such arguments could e.g. go along the lines of 'trait X allows the species that carry it to explore new niches, which could facilitate speciation as they become (reproductively) isolated from their conspecifics as they specialise into those different niches', or 'trait Y conveys higher survivability, thus potentially lowering extinction rates in comparison to their sister clades'. The authors are at times getting pretty close to making such arguments, so in those cases they may only need to be spelled out a bit more explicitly. Since the authors seem to possess great knowledge of hymenopteran biology, I believe this puts them in a prime position of devising such scenarios/mechanisms to enhance their hypotheses.

4. Vaguely related to the previous point: Interpreting the HiSSE plots (e.g. see line 241ff) is great, though I think some of these would become clearer if actual rate estimates were looked at. Indeed, I couldn't find parameter estimates for these models anywhere, only the numbers on model fit. I think those numbers should be shown somewhere and interpreted, as it would make it much more transparent what the actual rate differences are between the different rate categories (i.e. combinations of hidden and observed rates). It could even make the results a lot more impactful by interpreting the rate estimates too. In theory, it would allow to determine whether a trait leads to rate increases in all cases, but maybe of different strengths, or whether rate increases only happen in a subset of clades, etc. It also allows to spot whether a rate difference that is supported by model-testing only relies on a very small actual difference, which may be considered biologically negligible. Besides that, Beaulieu & O'Meara emphasize in their original paper introducing HiSSE, that considering the parameter estimates may be more important than merely focusing on which model fits best. Looking at inferred rates on the plot can help intuitively interpreting those values, but shouldn't be relied on (notably, I think it's technically possible to reconstruct the actual inferred hidden states, though I'm not sure how straightforward that is, and it may not be necessary).

5. Having read the rationale for the majority rule approach for ancestral trait reconstruction in the supplementary discussion (and on line 566ff), I have to say that while I can acknowledge the problem described, I am not fully convinced by the chosen solution.

Firstly, if the species level values are not representative of the missing diversity in the tree, then a clade-level assignment would be just as unrepresentative, assuming that you based this on information on the same taxa. If this was the result of surveying more species than are in the tree, to get a more accurate view on the character diversity in the group, then that would make this choice appear more justified, however, I could not find an indication anywhere that this is what you did, so this should be clarified. Indeed, I couldn't really tell what the source of the trait data was to begin

with, i.e. did you score it yourself from samples, or from the literature, or both? This also should be clarified.

Secondly, assigning the trait state by those clades may bias the result towards finding coinciding shifts in diversification rates and trait state at the base of those clades, while in reality, the trait change may have happened much later in a subset of the clade, depending on how the trait is distributed within. I would also presume that this simplified coding homogenised the rates of character evolution across your tree, which may be what is behind your result that the ER model fits best. Possibly, an analysis on species-level (or any other level) trait data would result in ARD to be a better fit, or at least would have affected the inferred rates and thereby the result. It is well possible that the result wouldn't change enough to topple your current interpretation, but this can't be said confidently as it stands. Furthermore, I was wondering whether you made any assumptions regarding the root state when reconstructing the trait states? The root state of many of your reconstructions seems (unsurprisingly) strongly affected by the states of the groups that are sister to the rest of the tree, but since you excluded the outgroups, you cannot tell if this is accurate. The outgroup sampling may not make it appropriate to include them in the reconstruction as is, but maybe with some knowledge on the trait states of the closest relatives to hymenopterans may allow you to make an informed decision on setting any prior probabilities on the root states.

All in all, it is clear that some simplifications were necessary given the data you are working with, but if those simplifications are potentially biasing your results in some way that matters for the hypotheses you are trying to test, the potential implications of this have to be addressed by tests (if possible) and in the discussion.

So at least, I would suggest you reconstruct ancestral trait states on a species-level tree as well, and compare/discuss the resulting patterns to your current approach. The actual species level data seems to exist, after all you are using it in the HiSSE analyses.

Apologies if some of those points are indeed mentioned somewhere, although I searched both the manuscript and supplementary material for it and couldn't find it, which may suggest that it has to be stated somewhere more obvious or had to be referred to more explicitly.

A few of those main points are echoed in my specific line comments below again too.

On the side, I would like to point out that there are multi-state SSE models, such as MuSSE (in diversitree for regular state-dependent diversification) or MuHiSSE (Nakov et al. 2018 for a hidden-states version) - or any custom variations of such one could devise of in RevBayes -, which might be preferable over multiple binary-trait analyses of what is really a multi-state trait (e.g. parasitoidism vs. carnivory vs. secondary herbivory etc.). While I wouldn't want to insist on the use of these for the paper at hand, I feel the authors may want to consider them, even if for future work (and maybe still address somewhere in this manuscript why they decided against them).

Similarly, the authors may want to consider work by Braga et al 2020 and 2021 on a new Bayesian approach to analyse ancestral host-parasite interactions. While the available data might not quite allow for this yet, this might be an avenue to consider in the future. As they mention themselves e.g. on line 305, the connection of parasitoidism and diversification hinges on the idea of the parasitoids tracking separate particular host lineages, which could be tested/established using such an approach.

Overall, I very much enjoyed this manuscript, though I strongly feel that my points above should be addressed in a satisfactory manner before publishing. This may include some of my minor comments below too. I hope the authors perceive my comments as constructive and potentially enhancing this exciting work further, and that they forgive any criticism that was merely the result of me misunderstanding their writing. I am looking forward to seeing an improved version of this work published in the near future!

Specific comments by line number: [Some of these may be shifted by down by a few lines due to some edits I made, though I hope this won't be a problem in finding what they refer to]

59: calling them 'singular' struck me as odd. I would have called them 'unique' or something the like, but this may just be me, it is definitely clear what is meant.

59ff: It is unclear to me what those reasons are - the fact that a very diverse clade is containing a lot of parasitoids? The hypothesis seems to be that they diversify because they adapt to different hosts, maybe this needs to be stated more explicitly and elaborated a bit?

64: not sure why 'the stinger' is in quotation marks?

66ff: Also here, is the suspected connection between diversity and stingers that the defense possibilities of it decrease extinction risk? Or something else? This should be clarified and referred to when discussing the results.

68ff: I appreciate you not calling them 'basal', though I believe technically 'early branching' may also not be entirely correct. If you can find a way to reword that along the lines of "sister to the rest of the clade" or something the like, that might be good.

71: I've not encountered the term groundplan before, only bauplan or body plan, though that might just be ignorance on my side.

86ff: Maybe I'm nitpicking here, but I don't think they can be considered catalytic if that hasn't been formally tested yet. They 'could' be seen as such, or be suspected or hypothesised as such, of course.

88: Maybe nitpicking too, but you are technically analysing them separately (albeit in the same study), and not jointly (i.e. in the same analysis).

141ff: I suppose the last sentence here is a foreshadowing on how results using topology C-1 and A-0 gave qualitatively similar results?

150: You don't seem to be using the acronym ACR anywhere else in the manuscript, so since it afaik isn't standard usage in the field, I'd suggest dropping it for simplicity.

160ff: As stated as a major point in the methods section, it should be discussed whether the location and frequency of these shifts could be affected by how the traits were coded by family.

162: Maybe write out 'node' for clarity.

242ff: along those lines, it should also be looked at and discussed, how the reconstructions and transition rate differences compare to those resulting from the ancestral trait reconstructions. Some differences can be expected - also because SSE models were initially thought up primarily as a way to account for diversification when reconstructing traits, rather than as a way to test for trait dependent diversification - but nevertheless inconsistencies should be addressed at least briefly.

250: I believe it should say *Vespina*

272: Oxford comma after 'regime', if you subscribe to that

289: Might be worth spelling out that the evolution of secondary phytophagy does not necessarily have to coincide with the onset of angiosperm diversification in order to be caused/affected by it.

305ff: Again, I think the reasons/mechanisms why the long history and tracking host lineages would have led to higher diversity (though it's more intuitively clear for the latter).

318ff: Inspecting the rate estimates more closely could help elucidate that idea.

321ff: Might be worth exploring whether any rate increase there might have been interpreted as a rate decrease in the sister-taxon of Apocritans (or generally members of the grade before)?

324ff: Maybe it could be discussed as a preadaptation/background trait (sensu Bouchenak-Khelladi et al. 2015 and therein)? Especially since it seems that having that kind of waist is critical to being a parasitoid (unless I have misunderstood that)? However, it should also be noted that SSE models are likely experiencing issues if a trait is too rare, which may be the case for wasp waist too.

325ff: I very much like the careful wording here!

329: the colours of the BAMM plots should be interpreted with care. The initial decrease could also be the result of high turnover, etc.

341ff: I like the idea that the stinger only had an impact once other modifications were in place too (making the stinger a preadaptation/background trait again, maybe). However, one could argue that a HiSSE model should potentially capture that, by finding elevated rates for a combination of the stinger and a hidden state nested within this clade. Support for character independence also must not mean that the stinger had no effect, but e.g. if there's elevated rates in clades outside the stinger-bearing

hymenopterans (or nested within, etc.), the rates within them would conflict with the impact of the stinger - and after all, you are looking at other candidate traits which could cause this, so potentially other traits affecting diversification might cancel out the signal of this one. This is purely speculative of course, but again looking at rate estimates (and possibly the location of the hidden states on the tree) could help interpret this.

347ff: A few things I pointed out for traits above apply here too.

352ff: This is a very interesting result/interpretation, especially since coding by carnivory only adds a few species (23 I think) to the set, when compared to parasitoidism. I would encourage to investigate this more (also looking at inferred rates and location of hidden states, I may add - with the hazard of sounding like a broken record). It's technically possible that this only reveals that the positive result for parasitoids was just driven by some odd bias, of course, which one might discover that way as well.

361ff: I find this point about the scenario for delayed effect of parasitoidism very interesting and indeed in line with the initial theory of adaptive radiations *sensu* Simpson, i.e. requiring both the key innovation as well as entering the adaptive landscape - which would here be the presence of an abundance of suitable hosts (this is sort of what I meant above regarding angiosperms too).

414ff: Again, the scenario needs to be fleshed out more to justify why increased larval provisioning would accelerate diversification (i.e. whether and how this leads to more speciation and/or less extinction)

417ff: The idea of a 'pre-innovation' would probably connect to related concepts in Donoghue et al. 2015 or Bouchenak-Khelladi et al. 2015?

440ff: I like the idea of the modification of 'key innovations' as an actual driver of diversification (like a modifier *sensu* Bouchenak-Khelladi et al. 2015), and would be looking forward to seeing this addressed more in future work! Same for the analysis of the subcomponents mentioned at 447!

459ff: I think it would be good to explicitly state what coverage these taxa represent at different taxonomic levels (I'm aware this information can be found in table S13, but maybe a coverage-range might be helpful for readers to get a rough idea of the sampling heterogeneity).

467ff: The apparent availability of the raw data is great!

477: "50%, 60%, and 70%"

490/491: Oxford commas before the 'and' in both cases here (and throughout), if you are partial to that

508: Not sure, but think 'Additional' should instead either say 'Additionally', or better 'In addition'?

508/515/516: It's not immediately clear to me here how those maximum and minimum bounds are made to be soft. I believe this is usually meant to say that the prior distribution has some kind of tail where the probability decreases gradually. Is the truncated Cauchy distribution referenced later on meant to describe these soft bounds (and all of them)? It would seem a truncated distribution would have a hard bound rather (unless the truncated end is on the side of the present time), so maybe this should be clarified a bit. Also, if possible add a short comment on why the default settings are justified here.

511ff: Maybe I missed this, but what is the rationale for using these two data sets specifically here?

519: For the effective sample size, what was your cutoff to signify sufficient convergence (and did all parameters reach/exceed it)? I presume the ESS were post-burnin too?

520: I got a bit confused with the post burnin samples here. You said you ran 2mio samples, subsampling every 10th, but then each run has more than 2mio states even after discarding burnin? I'm sure I'm just overlooking something that's maybe implied here, but that might suggest this needs to be clarified a bit better.

528ff: I believe assigning missing sampling by clade - while being an important correction - has the potential to bias the analysis towards finding shifts at the base of the clades to which the sampling fractions were assigned. I am not aware of a formal way to test for that (or whether the taxonomic level at which you chose to assign missing sampling has an impact on shift location), but maybe this can be addressed somewhere as a possible caveat? Also, in table S13, some taxa inexplicably have a slash instead of the sampling fraction, including one of the clades that have a shift, it seems unclear whether this is an error or whether it means something.

543: You do mention the low frequency of the MAP shift configuration is, and I agree that the

cumulative shift probability is a good way to deal with this problem. However, I would have liked to see a plot of at least the few most common configurations and their frequencies, e.g. somewhere in the suppmat, to get a better intuition for just how representative the MAP configuration is.

546ff: As you point out correctly here, Chang et al. suggest that the taxonomic method is preferred when sampling fraction is below 1%, which seems to be the case for your data. However, when comparing the total number of taxa in the whole group vs represented in your tree between what you describe on line 398 (7882+144809 = 152691 total) and line 99 (765 taxa sampled) vs. the numbers shown in table S13 (181358 total, 930 sampled), it might strike some as confusing why those numbers differ. The overall sampling fraction of both number pairs comes out to be pretty much the same (0.0051 vs 0.005), so I'm sure the difference relates to something reasonable that doesn't reveal itself to me here, and probably doesn't affect the result, but you might note somewhere why the numbers differ, to clarify that.

581: I am very pleased to see the thoroughness in testing alternative models here!

590ff: It technically becomes apparent from the plots and supplementary tables (and esp the caption of figure 4), but I think it would make things clearer if you would state somewhere around here which topology you used for these analyses, and that it was on the full tree and not the one collapsed to family level again.

Figure captions:

- I thought citing references in figure captions was unusual, but I'll trust the journal has guidelines for that.

- I overall find the figures very clear and aesthetically pleasing!

- Figure 3: the word darkcyan should probably be separated into two; the signs for the shift probability should probably be switched to 'smaller or equal to 0.99'

- Figure 4: dito for darkcyan; I like the colour scheme overall, but was wondering whether the choice of grey for absence of the trait makes it blend in a bit too much with the outer colour (though HiSSE plots are notoriously difficult to make clear)

Reviewer #3:

Remarks to the Author:

Authors are presenting a strong, very impressive study on the evolution of Hymenoptera. This study is outstanding by the quality and quality of the dataset (sampling of taxa and development of molecular dataset) but also by the comprehensive analyses of this dataset. The evolutionary questions are very interesting and they will interest a wide community of biologists. I have a very few comments to improve the manuscript.

Abstract: The description of the goal is a bit confusing as you insist on the impact of innovation. So we are expecting line 32 to have information on the impact of parasitoid regime on the diversification.

SO, I would write line 27 "... to describe the evolution of the Hymenoptera, the timing of apparition of their key innovation: ...; and their impact on diversification.

Introduction: I would add at least a sentence on the abundance of Hymenoptera. Their diversification is an important point, but the function they have in the ecosystem is also linked to their high abundance (bees for pollination, ants for biomass in tropics, ...).

Line 58: the term generalist is a bit ambiguous here while you describe after the wasp as predators. So, in this 30%, I would say, phytophages like bees, predators like wasp and generalists like ants.

Line 62: the characters are here as key innovation. So, I would not present this as an hypothesis and not as a common fact.

Line 65-66: bees and ants are ecosystem engineer because they are diverse and because they are abundance (partly due to their social behavior).

Results: very well summarized, and it was not an easy task with such a huge dataset set and important number of analyses.

Discussion: Line 286, I would not cite the study Genisse as they don't describe body fossils but trace fossil. Moreover, this study postulates the sweat bee as 100MA old. The present study itself does not support this age as all bee clades are younger than 75MA (including the Halictidae, sweat bees). Following references are more accurate:

Cardinal, S., & Danforth, B. N. (2013). Bees diversified in the age of eudicots. *Proceedings of the Royal Society B: Biological Sciences*. <https://doi.org/10.1098/rspb.2012.2686>

Michez D., M. Vanderplanck & M. S. Engel 2011. Chapter 5. Fossil bees and their plant associates. Pp 103-164, in *Evolution of Plant-Pollinator Relationships* (Ed. Patiny S.).Cambridge University press, ISBN-13: 9780521198929.

Line 326: like to say in the introduction, the sting could be one of the trait supporting the apparition of sociality, which is a key evolution, more in term of abundance than species diversity probably.

Line 375: generalism is quite rare in Hymenoptera or in Insect. So generalism (wide ecological niche), more than polylectism (many host plants) could be the key here. Moreover it's a generalism on an expending clade (Angiosperm). The secondary phytophagous shifted on an diversifying clade, not on Gymnosperm or Pteridophyta. There are key element showing co-diversification in these clades.

Masaridae are also secondary phytophagous and do nest, and they did not diversify well. Same for Melittidae (Michez et al. 2009: *Phylogeny of the bee family Melittidae* (Hymenoptera: Anthophila) based on combined molecular and morphological data. *Systematic Entomology*,34: 574-597).

So, I'm not fully convinced that the combo "phytophagy" + "nest" is the key to explain diversification .

Methodology: I don't understand why authors selected only 12 fossils and why they did not select fossils from more derived groups with well characterized taxonomic association (e.g. bees).

Overall, congratulations again for this amazing, inspiring article. I fully support its publication.

NCOMMS-22-11260 - Response to reviewers

We thank all reviewers for taking the time to review our manuscript and provide their detailed comments which greatly improved our manuscript.

We have provided below a point-by-point response to each comment, either describing how we addressed the respective issue in our revised submission, or explaining why it was not feasible or advisable to follow the specific recommendation. The original reviewer comments are copied in black font and our responses are inserted below in blue font. Line numbers refer to the word document of the manuscript with track changes highlighted (not the pdf). Changes in the supplementary text have been indicated with blue font.

In addition to the changes outlined below, we have also made minor edits to the text to improve clarity and conciseness, and to adhere to the journal formatting guidelines.

REVIEWER COMMENTS

Reviewer #1 (Remarks to the Author):

The manuscript by Blaimer and coauthors presents a) a more comprehensive phylogeny of Hymenoptera than preceding studies, b) new divergence time estimates, and c) an analysis of whether or no specific traits (i.e., parasitoidism, wasp waist, stringer, and secondary phytophagy) possibly impacted the diversification rate in this mega-diverse insect order.

The backbone phylogeny and the divergence times of Hymenoptera have been the subject of various phylogenetic studies published during the past five years. What has been missing is a credible analysis of where in the phylogeny of Hymenoptera shift in diversification rates occurred and what factors (key innovations) could have fostered these shifts. I consider such an analysis of major interest, and the results possibly be included in text books.

While the manuscript aims to provide the above analysis of key innovations and diversification rates, it has various severe shortcomings.

[#1] One major problem that I see concerns the experimental design. The authors state that a reliable phylogeny and credibly dated phylogeny are paramount for assessing diversification rate shifts. This is true to some extent, but an at least equally important factor is proper species sampling. A recent (2022) study published by Craig et al. in Mol. Bio. Evol. re-examined previously published studies that addressed speciation rate shifts and found species sampling and sequence variation to be the main driver of rate shifts, rather than the biological explanation put forth by the authors of the respective studies.

We thank the reviewer for referring us to the Craig et al. (2022) study, which investigates a correlation between the completeness of taxon sampling and sequence length and the location of rate shifts estimated within diversification rate analyses. The cited study inferred shifts at progressively more recent ages in the phylogeny with increasing size of data set and completeness of taxon sampling. While these results are indeed concerning, we think the criticism in this study cannot be directly applied to our work, for the following reasons. First, the software used in the Craig et. al. (2022) study (TreePar and TESS) to identify these trends is based on different algorithms than we employed in our study. The results may therefore not be directly comparable, as it is unclear if and how diversification rate estimations with BAMM, HiSSE and Medusa would be affected by the same variables in the same way.

Second, with regard to taxon sampling, all our analyses employed clade-specific and backbone sampling probabilities to correct for incomplete sampling and bias. It appears that the Craig et al. (2022) analyses producing the described correlative patterns did not correct for incomplete sampling and bias. Thus, we assume our analyses to be more robust against these criticisms. Third, with regard to sequence length, since we performed our dating and diversification analyses on two different subsets of our data (the nuc-50% data set with resulting tree topA-0 and the nuc-70% data set with the resulting tree topC-1), our paper compares results from data sets with different alignment sizes (132,042 bp vs 40,672 bp). As summarized in supplementary data 6, our divergence estimates from these two data sets are highly congruent and so is the placement of the inferred diversification rate shifts (supplementary data 9) based on the two different chronograms (except in areas where the underlying topology differs). Another aspect which we already highlighted in the manuscript is that one of our main recovered diversification rate shifts (in non-melittid bees) had been inferred in an earlier study as well, based on completely different taxon sampling and analysis methods (Murray et al. 2018). We think that these cross-validated results speak for the robustness of our analyses. Finally, we would like to politely point out that the level of sampling completeness suggested by Craig et al. (2022) appears to have vertebrates in mind and is unrealistic to be achieved for higher-level analyses of insects. At a recommended minimum of 75 aligned sites for every described species in the clade of study and 75% taxonomic sampling of all known species, such a data set for Hymenoptera would have to comprise around 11.5 million bp and 114,500 species – clearly not a feasible data set to compile and analyze at this point in time. In any case, we have acknowledged the concerns described in Craig et al. (2022) in the manuscript as follows in lines 654-659: “Recent work further suggests that the location of diversification rate shifts may be influenced by sampling completeness and the number of alignment sites analyzed⁶⁵, but our results are largely congruent and robust across data sets of different size and types of analyses. Incomplete sampling remains a limitation, yet one that is unlikely to be addressed for insects in the near future.”

[#2] While the present study analyzed significantly more species than previous phylogenetic studies on Hymenoptera, it does not outline based on what criteria species were selected given the aims of the study. There is some information given in the supplementary text (“Our goal was to create a well-sampled, balanced taxon sampling across the order; however, our sampling has an emphasis on the hyperdiverse lineages within the superfamilies Ceraphronoidea, Chalcidoidea, Cynipoidea, Ichneumonoidea and Platygastroidea.”), but it does not provide details, and the second part of the sentence suggests that imbalance in taxon sampling may just have shifted. Looking at the phylogenetic tree, phylogenetic lineages that differ in one to two orders of magnitude in their species richness, are represented by one and two species, respectively, which does not seem to be representative at all. For me, the taxonomic sampling does not seem to be specifically tailored for a species diversification analysis of Hymenoptera in general, but was focused on parasitoid non-Aprocrita (which is fine and welcome, but which is not sufficient for the aims of the study).

By placing an emphasis on parasitoid lineages, the bias has not shifted towards them; rather, we have achieved a more balanced sampling between parasitoid and non-parasitoid Aprocrita (see Table S14). Collectively, the author list of the study includes specialists on all major groups of Hymenoptera and each expert carefully selected the taxa for the study aiming at both (1) representation of all major lineages (i.e. subfamilies), sampling across the root node for each family; and (2) species-level representation approximately proportional to the extant diversity. The level of completeness in terms of family representation is very good: we have 95 (out of 101 currently recognized) families represented and for the most part these are sampled relative to their diversity.

To some extent, however, it is logistically impossible to achieve perfect taxonomic representation because as the reviewer points out, some major lineages differ in orders of magnitude in their extant diversity. If one family has four extant species and another one has 20.000, if we sample a single species of the former we would need 5.000 species of the latter to have perfectly unbiased sampling, which is unfeasible. If we sought such a randomly unbiased sampling on our 700-taxa tree, this would mean refraining from sampling several small families, many of which represent unique branches in the hymenopteran tree, which would also have deleterious effects on the diversification rate estimations.

We have clarified our sampling strategy in the main text of the manuscript (lines 956-961). Adding more tips to the tree to a level that would be financially and logistically feasible is unlikely to improve date and rate estimates drastically - for a significant advancement of estimates one would probably have to increase sampling 20x to include all generic diversity (and still only have 10% of the diversity of Hymenoptera sampled).

[#3] A major part of the results and discussion is dedicated to the backbone phylogeny and divergence times of the major lineages. However, the presented results regarding the backbone phylogeny are virtually identical to the ones published in previous studies. On the other hand, results that differ significantly, such as the early divergence of sawfly lineages, are not addressed in the main text. The presentation of the ancestral state reconstruction is also for the most part repetitive (e.g., that parasitoidism evolved in the most recent common ancestor of Vespina). The extended phylogenetic sampling holds a lot of information, but the authors decided to present almost exclusively confirming results.

We politely disagree that a major part of the results and discussion is dedicated to phylogenetic results. Most of the phylogenetic results are discussed in the supplementary material and only roughly 150 out of 1950 words in the Results section deal directly with the backbone phylogeny. Currently there is one paragraph outlining superfamily-level results and no mention of these results in the discussion in the main text.

Even if many of our results do agree in one way or the other with previously published results, it is important to provide a brief description of these results in the main text, since our study represents a substantial increase in the taxon sampling compared to previous phylogenomic studies and thus a more robust test of the relationships of many groups. There is still not a widely accepted consensus for the entire backbone phylogeny of Hymenoptera due to conflicting results. In particular, our results for some relationships within Proctotrupomorpha are worth highlighting as they have not been recovered elsewhere with strong support and comprehensive taxon sampling. At the same time, we have restricted our reporting of results in the main text to the “bigger-picture” level. Our tree does include results that are novel compared to previous study, in part due to the much-increased taxon sampling; these are discussed in detail in the Supplementary Material and also in recent and upcoming papers that expand on the dataset of this study (e.g. Blaimer et al. 2020, DOI: 10.1186/s12862-020-01716-2; Cruaud et al. 2020, DOI:10.1111/cia.12416; Jasso-Martínez et al. 2022, DOI: 10.1016/j.ympev.2022.107452). We do agree with the reviewer though that results regarding the early divergence of sawfly lineages are among the most interesting, so we have expanded a little on those results in the main text (lines 283-287).

On the other hand, we did recover some conflict in our data, and removing the short summary of results from the main text would be misleading. Minimum information on phylogenetic analyses, results and topological conflict is needed to understand why macroevolutionary analyses were performed on the two contending topologies, for example.

Regarding the dating analyses, our results are noteworthy for highlighting the considerable lag time between the evolution of parasitoidism and the advent of other derived strategies such as secondary phytophagy – the phenomenon we described as a “parasitoid regime”. Finally, while our ancestral state reconstruction results for parasitoidism may seem

obvious, no study has ever provided an actual model-based reconstruction for this trait, thus we strongly feel that we need to report such results in the main text, even if briefly.

[#4] The divergence time estimates differ from those given in previous studies. The question is why: it is very well possible that the estimates presented in the present study are more accurate. However, the authors do not assess the credibility of different estimates. All they state is that the taxonomic sampling and the calibration strategies differed from those in previous studies (which is true, but the different studies also used different fossils to calibrate the trees and they used different types of data; none of which is or discussed mentioned). If the divergence times are important and a major result of this study, it think the authors should put significantly more effort into working out what causes the differences and what estimates are more credible. The paper otherwise presents just another set of estimates and the reader has to decide which one to trust more.

We had kept the comparison of our divergence estimates to those from previous studies in the discussion relatively brief and general since the previous estimates were established using different taxon sampling, fossil calibrations, data types and dating methods. Getting to the bottom of why these estimates differ would require a comparative investigation that includes extensive re-analyses of the previous data sets, and potentially also some data simulations. This goes well beyond the scope of our study, but would indeed be an interesting separate paper. We do believe that our expanded taxon sampling presents a major advancement for estimating divergence ages for Apocrita, but one would need to control for all the other above-mentioned variables to conduct a fair and equal comparison.

In response to the reviewer's comment, we have expanded the relevant parts of the discussion in the main text, but still kept this comparison relatively succinct given the space limitations (lines 581-589). However, we have also added a section to the supplementary discussion ("Comparison of divergence dating results", p.45) including a more substantial comparison and discussion, and we have also added a side by side comparison of selected divergence ages from Branstetter et al., Peters et al. and Ronquist et al. to supplementary table 6.

Overall, I found the phylogenetic part of the study not being well presented, as it focusses on verifying hypotheses that have been widely accepted by now. The part on divergence time estimates is not sufficiently worked out. And the divergence rate estimate part suffers from the problem that the taxonomic sampling does not seem to be specifically tailored for such an analysis and it thus bears the risk of being possibly misleading. I found the last sentence in the supplementary discussion quite illuminating in this regard "In the light of these caveats, we regard our estimates as the best achievable for hymenopteran diversification at the moment, but improved models for estimating diversification rate or an expanded phylogeny for the group may result in different conclusions."

The criticisms reiterated in this summary paragraph have already been addressed above.

I therefore cannot recommend publication of the manuscript in its current form in the journal Nature Communications.

Specific comments

Line 25: Why are sawflies not mentioned?

We now mention sawflies (and woodwasps, too).

Line 25: The statement "one of the most diverse animal lineages" is unscientific, unless the basis for the comparison (geological age of the lineage or the taxonomic rank) is specified. Every superordinated lineage is more diverse, and this results in many more lineages being more species rich than Hymenoptera.

We changed this to read “The order Hymenoptera (wasps, ants, sawflies, and bees) represents one of the most diverse animal lineages”.

Line 28: I do not think that it is possible to test whether the presence/absence of a trait historically influenced (= causal relationship) the diversification rate. All that can be done is to assess whether there is a correlation that suggests that there could have been a causal connection. Please rephrase.

We rephrased (lines 27-28).

Line 29: “apocritan wasps”, better write Apocrita, because Apocrita includes bees.

Modified as suggested.

Line 29: “aculeate Hymenoptera”, better write Aculeata, as Aculeata are per definition Hymenoptera and hence Hymenoptera is redundant.

Modified as suggested.

Line 31: I find the expression “parasitoid regime” awkward. Why not parasitoid lifestyle?

We modified to “parasitoidism.”

Line 34: “hymenopteran diversification rate”, better “diversification rate of Hymenoptera”.

Modified as suggested.

Lines 39–40: I wonder whether this bold statement is justified. Comparing diversification patterns makes in my eyes only sense when accepting common decent of organisms and when correcting for the available time for diversification. The basis for comparing diversification was set 166 years ago: less than two centuries ago. If diversification patterns were legitimately compared and discussed before, please provide the corresponding reference(s).

We modified this statement to “since the early days of phylogenetics”.

Line 47: “earth”, better “Earth”, more explicitly referring to the planet

Modified as suggested.

Line 53: “(Psocodea¹³)”, better „(Psocodea)¹³“, as the reference refers to the entire statement, not only on the information in parentheses

Modified as suggested.

Line 54: “(Hymenoptera: Siricidae¹¹)”, better „(Hymenoptera: Siricidae)¹¹“, as the reference refers to the entire statement, not only on the information in parentheses.

Modified as suggested.

Line 58: “generalists, such as many ants and social wasps”. This is too unspecific and is even misleading. Aculeata includes various lineages that are kleptoparasitic and that cannot be considered generalists by any means. Most ants and social wasps are considered predators.

We have modified this statement to read “predators, such as many social wasps.” (line 124)

Lines 65–81: The order of the two key innovations should be reversed, as the wasp waist evolved before the stringer.

We have reversed the order of appearance in the text.

Line 73: Sawflies do not represent a natural group and hence the use of the old taxonomic name for the group should be put in parentheses: “Symphyta”.

Modified as suggested.

Lines 80–81: “This character [the wasp waist] was potentially foundational for enabling both parasitoidism and stinging.”. I find this statement misleading. The parasitoid lifestyle is ancestral in Apocrita and in Orussoidea. Given the widely accepted sister group relationship of the two lineages, the most parsimonious explanation is that parasitoidism had already evolved in the ancestor of the two groups (Vespina) and hence BEFORE the evolution of the wasp waist. Insinuating that the wasp waist represents an evolutionary innovation that was necessary for parasitoidism to evolve is incompatible with all evidence gathered during the last decades. Whether the wasp waist represents a character that was necessary for a stinger to evolve is certainly possible, but in my opinion fruitless speculation. The stinger evolved only once, in a subordinated lineage of Apocrita. Given these data, I do not see how it could possibly be tested

whether a string could have only evolved in wasps with a wasp waist. And if a hypothesis cannot be tested, it is scientifically worthless.

We agree that this statement may have been worded too strongly and the connection between these characters is difficult to test. What we meant was that the wasp waist may have been an important functional innovation leading to the success of parasitoidism and the stinger. We modified the statement (lines 138-139): “, and thus potentially facilitated the success of both parasitoidism and stinging in apocritan wasps.”

Lines 84–86: “, phytophagy has secondarily evolved in several groups of apocritan wasps, for example in pollen-collecting bees and in gall-inducing cynipoid wasps“. These are two prominent examples, and these examples are well covered in the sampling. However, other examples, such as in Vespidae (pollen wasps) and in apooid wasps (*Krombeinictus*; Krombein & Norden 1997) are neglected in the sampling and completely ignored in the discussion. Their consideration is important, though, when discussing whether a switch from carnivory to phytophagy generally results in an increase in the diversification rate. Chances are good that it did not result in such an increase in these two groups and it points at the fundamental problem that the success of a particular life style depended on multiple factors (e.g., additional traits, food sources, competition).

We thank the reviewer for pointing out the omission of these two examples. We have added a reference to pollen wasps and *Krombeinictus* in the discussion (lines 837-838). It is certainly true that these represent species-poor clades with secondarily phytophagous behavior that do not necessarily fit with the hypothesis of secondary phytophagy driving diversification. Our family-level analysis, however, aimed at identifying drivers of diversification in the earlier evolution of Hymenoptera and was not designed to identify rate shifts in isolated lineages at lower taxonomic levels.

Lines 88: “colonization of new niches”, better “formation of new niches” as some niche concepts consider the species being part of the niche.

Modified as suggested.

Lines 90–91: “All four innovations—the wasp waist, the stinger, parasitoidism and secondary phytophagy“. What is the logic of the order (neither synchronically nor alphabetically)? The in my eyes more logical order would be parasitoidism, wasp waist, stinger and secondary phytophagy (the order of the last two could be changed, but as bees serve as one major example of secondary phytophagy, listing stinger first is more plausible).

Modified as suggested.

Lines 96–98: “Yet, a complete and robustly supported phylogeny for the order is still elusive, and the even most recent phylogenomic analyses^{27,29,30} were not able to provide clarity on the placement of some lineages, mainly due to sampling bias toward aculeate wasps. A reliable phylogeny of supra-familial relationships and their evolutionary timescale, particularly of non-aculeate Apocrita, is paramount to any study of the diversification and evolution of key innovations in Hymenoptera”.

I find these statements problematic, as they do not seem informative or justified. First, a complete and robust phylogeny of Hymenoptera is impossible achieve any time soon, as there will always be species missing given the size of the taxon. So, this statement is more or uninformative.

We have modified the paragraph by removing the first sentence – we did not mean the word “complete” in the literal sense of including every single species of Hymenoptera, but this was apparently not clear.

It is true that the phylogenetic positions of some major lineages are still considered unreliable. But why is this important in this context? The second sentence states that a reliable phylogeny is paramount for studying diversification pattern. However, the phylogenetic results presented in the main text are basically identical to the ones published during the past five years. And a previously reported phylogenetic uncertainty in the relationship of Ichneumonoidea and

Ceraphronoidea is not resolved in the present study either. Interestingly, this uncertainty did not impact the diversification rate estimates, indicating that robust phylogenetic relationships are not per se a prerequisite for studying diversification rates. I think the problem with the entire paragraph is that does not explicitly state what the goals of the present study are and what is required for achieving these goals.

We have modified the paragraph to exclude the mention to the importance of a robust topology, while stating more clearly that the goal of the study was to investigate the early diversification of Hymenoptera and the evolution of the four putative key innovations (lines 219-220).

Studying more genes and more species is pointless unless it serves a specific purpose. If the purpose is studying diversification pattern, then a representative sampling and a reliably dated phylogeny for this sampling is required. Unfortunately, the manuscript does not mention this and what was done to compile such a dataset. The text on the Supplement states “Our goal was to create a well-sampled, balanced taxon sampling across the order”. But what measures were taken to realize such a balanced sampling? What percentage of the species of each lineage is included and what was done to not oversample early splits (most diverged lineages)? The sentence immediately following the above one in the supplement suggests that the current sampling may be biased in a different direction: “our sampling has an emphasis on the hyperdiverse lineages”.

We have addressed the criticism about the "balance" in taxonomic sampling under major comment #2 above.

Line 103: “for 765 taxa”, better “from 765 taxa”

Modified as suggested.

Line 129: “two important subdivisions”: what makes these two subdivisions important? Are Ichneumonoidea and Ceraphronoidea unimportant?

We changed the phrasing to “two subdivisions.”

Lines 129–147: The paragraph presents the backbone of the main tree, which is identical to the ones inferred in previous studies. Even the uncertainty in the phylogenetic position of Ichneumonoidea and Ceraphronoidea to each other has previously been reported. Thus, paragraph does not provide new information on the evolution of Hymenoptera. The paragraph can be condensed into a single sentence stating that the inferred backbone of the phylogeny is consistent with our current knowledge of the phylogeny of the major lineages of Hymenoptera.

The main criticism in this comment has already been addressed above (major comment #3). We emphasize again that hymenopteran phylogeny is still contentious and the community is far from agreeing on a backbone phylogeny. We would also like to highlight in addition that minimal information on the backbone phylogenetic results, as we provide them in the cited paragraph, is needed to understand why our diversification analyses were performed on two different topologies. The main text needs to be comprehensible without reading the supplementary material, and shortening this paragraph to one sentence as suggested will prevent this. No modifications have therefore been made.

Line 152: “other dominant strategies in Hymenoptera”. Can you please outline what these strategies are specifically?

We now list all strategies (lines 303-304).

Lines 158–159: “The sawfly and woodwasp lineages branched off from other hymenopterans”.

Please rephrase, as it remains unclear what “other hymenopterans” refers to specifically.

We rephrased this sentence (lines 316-317).

Line 163: “Our ancestral reconstructions”, better ancestral state reconstruction, as the reconstruction is not ancestral, but the states that are reconstructed.

Modified as suggested.

Line 169–170: “The two main divisions within Apocrita, Proctotrupomorpha and the Aculeata + “Evaniomorpha” grade”. Why are these two the main subdivisions and why are Ichneumonoidea

and Ceraphronoidea not considered main divisions? What is needed to make a lineage be “main”?

We modified this sentence (line 327).

Line 192: “s. s.”. Most readers will not know this abbreviation, so better write *sensu stricto*. However, I expect most readers to not know the meaning of this Latin phrase either, so I suggest to better write “in the narrow sense”. This applies to all subsequent instances of “s. s.”, and also to instances of “s. l.”

We agree that many readers may not be familiar with these abbreviations. However, spelling them out in English throughout the paper will be awkward, especially since the abbreviations appear in figures as well. We have therefore defined the two abbreviations on first mention (line 372 and 418), but retained their use throughout the manuscript.

Lines 267–268: “The origin of species richness in Hymenoptera has been of acute interest^{8,39}, but phylogenetic uncertainties in the early evolution of the order have persistently prevented strong conclusions.” What uncertainties and what conclusions specifically? Is this really true? The taxonomic sampling of sawflies in the here presented phylogeny is still very limited (but their diversification rates extensively discussed) and the uncertainty about how some lineages are related to each other (e.g., Tenthredinoidea relative to Pamphilioidea and Xyeloidea) has not been solved either.

We have removed this statement.

Lines 269–271: “Our work targeted an increased sampling of parasitoid lineages to create an improved phylogenomic framework to evaluate the association of putative key traits with the diversification of Hymenoptera.” The major novel aspect in this study is the analysis of diversification rates. Such analyses critically depend on a representative sampling of the diversity in a group. I acknowledge that previous studies had a very imbalanced taxonomic sampling. It is therefore important to outline what was done in the current study to achieve a balanced taxonomic sampling. Increasing the taxonomic sampling of parasitoid lineages could have helped reducing the imbalance, but the above statement does not state this. In fact, it could have resulted in a new imbalance. Note that a recent study (Craig et al. 2022. *Mol. Biol. Evol.*) found insufficient species sampling and paucity of sequence variation to be main drivers of speciation rate shifts in many previously published studies (rather than the proposed biological explanations).

These criticisms are reiterated from major comments #1 and #2 and have already been addressed above.

Lines 271–273: “Many of our phylogenetic results corroborate previous findings with support that is substantially enhanced or overwhelming, while other results underline where uncertainties continue to persist”. Can you be more specific? I am having trouble matching this statement with the presented results. The phylogenetic backbone data presented in the results text are identical to previously reported ones and the support for these was in at least one of these studies maximal. Only in case of the suggested sister group relationship between Ceraphronoidea and Ichneumonoidea was support previously low, and it still is in the present study.

As we point out in the sentence following the quoted statement, we discuss phylogenetic results in detail in the supplementary material. We agree though that the statement was rather vague and uninformative, and have therefore removed it.

I acknowledge that the phylogeny shown in Figure 1 provides support for clades previously not well supported, but these clades are not addressed in the results at all. So why no present the new results? Of particular interest would be the earliest splits of Hymenoptera. The phylogeny shown in Figure 1 is incompatible with some of the recently published phylogenies, but this result seems to depend on what data were analyzed. This information is buried in the Supplementary Text. It states that Pamphilioidea + (Tenthredinoidea + Xyeloidea) is found when having analyzed amino acid data only. So, what does this mean? Is the amino acid data less or

more trustworthy than the nucleotide data? I find it problematic that Figure 1 suggests relationships that contradict those of other studies and that these differences are not discussed. Many readers in an interdisciplinary journal will erroneously assume that a high statistical support value indicates credibility.

This criticism is repeated from above, please see our reply under major comment #2. We have inserted a very brief discussion of sawfly relationships in the main text (lines 283-287).

Line 279: What is meant with “regime” of parasitoidism?

Modified to “persistence.”

Line 286–289: “Likely reasons of these incongruities are differences in taxon composition between ours and the previous analyses, as well as different calibration strategies (i.e., node-dating in the present and Peters et al.²⁷ analyses, and tip-dating in Ronquist et al.⁴¹).” These studies also used different fossils, and some of the studies analyzed different types of data (nucleotide incl. 3rd codon position vs amino acid).

We now mention additional reasons in lines 581-589.

Line 291–292: “Our data suggest the entire Jurassic hymenopteran fauna was dominated by a parasitoid regime possibly lasting for about 100 million years.” Given that even today there are (to my knowledge) more Hymenoptera with parasitoid lifestyle than with any other life style, has the parasitoid life style not be dominant ever since?

We modified this statement to “Our data suggest that parasitoidism has been the dominant life strategy in Hymenoptera since the late Triassic.” (lines 590-591)

Line 336: “diversification rate shift” In what direction?

Modified to indicate a positive diversification rate shift.

Lines 369–371: “This may have been too early for the group to have immediately benefited from the dramatic diversification of other holometabolous orders, which seems to have taken place in the last 150 Ma^{49,55–57}.” Note that a study recently published by Oeyen et al. (2020) in Genome Bio. discusses molecular adaptation that evolved in Apocrita, but not in Orussoidea that could have fostered the diversification of parasitoid wasps.

We thank the reviewer for reminding us of this paper, but we do not think a reference to this study fits in the context of the “lag-time hypothesis” we postulated in this paragraph.

Lines 375–376: “in the vicinity of secondarily phytophagous clades” sounds awkward. Perhaps better “closely related to clades with phytophagous species”

Modified as suggested.

Lines 385–387: “Similar to a hypothesized association of an advanced stinger with diversification in non-chrysidoid aculeates, polylecty could be seen as an advanced form of pollen-collecting in bees, the secondary innovation hereby driving diversification.” Is there evidence that oligolecty is indeed ancestral? If so, please provide reference. Otherwise, this is quite some speculation. Only because melitid bees are oligolectic does not imply that the ancestor of all bees was also oligolectic.

Indeed, there is some evidence that oligolecty is ancestral in bees, as summarized by Dötterl & Vereecken (2010). We have added this reference to support this statement.

Lines 397–399: “However, while our taxon sampling for cynipoids and bees was balanced and representative of species diversity”. This is important information. It should be presented earlier. However, it remains still extremely vague. How was balance and representativeness accomplished? What measures (e.g., percentage of species) were taken? Looking at the phylogeny, I do not see a balance. For example, within Chrysoidea, bethylids and chrysidids are quite species rich, yet they are represented by only two species each and the species poor taxa in this superfamily with one species. This does not appear proportional and representative. Could this explain why a rate shift was only detected in non-chrysidoid Aculeata?

We have clarified the sampling strategy and percentages in the Methods section (lines 956-961), but found mentioning details that clearly belong in the methodology earlier in the text to be awkward.

With regard to the specific question if the rate shift in non-chrysidoid Aculeata could be an artifact of imbalanced sampling: As we have pointed out above, we have corrected for unsampled diversity in diversification analyses by using clade-specific sampling frequencies for BAMM. The MEDUSA analysis, for which the input was a family-level only phylogeny + species richness matrix, also recovered the rate shift in non-chrysidoid Aculeata, which speaks further against any artifacts due to imbalanced sampling.

General comments by the reviewer with regard to taxon sampling have been addressed above under major comment #2.

Incidentally, why does Figure 1 list three bethylids, but the phylogenies in the supplement contain only two? A discrepancy in species counts exists in other families, too.

Three bethylids are represented in Fig. 1 and all supplementary trees (*Epyris*, *Pristocera*, *Goniozus*). However, *Pristocera* was apparently misspelled as *Pristocerus*, a mistake that was perpetuated by using data from Branstetter et al. (2017). Maybe this caused some confusion; we have corrected the spelling throughout our supplementary data and figures.

We have further carefully cross-checked all species numbers listed in the collapsed tree of Fig. 1 with our supplements, and found and corrected the following discrepancies: 1) Azotidae was erroneously collapsed together with Signiphoridae in Fig.1; 2) Chalcididae should list 9 species instead of 10; and 3) Platygastriidae should list 26 species instead of 27. Thank you for alerting us to these errors.

Line 401: Please rephrase, as “vicinity” of rate shifts sounds awkward.

We replaced “vicinity” with “location”.

Lines 405–408: “The ancestrally phytophagous “symphytan” lineages have not diversified exceptionally (7,882 vs 144,809 described species in Apocrita at the time of writing); in fact, net diversification rates estimated for Tenthredinoidea, Siricidae + Xiphydriidae and Pamphiliidae + Xyelidae estimated with BAMM were slightly negative (Table 3)”. A statement on a shift in the diversification rate at the base of the Hymenoptera phylogeny depends to my knowledge critically on the ancestral condition and hence on the speciation rate in the outgroup. However, the outgroup sampling comprises only five species, each one representing a different insect order. I therefore do not see that the outgroup sampling allows to inferring ancestral rates and hence deducing that the sawfly lineages had a lower speciation rate than expected.

It is correct that identifying a diversification rate shift in the most recent common ancestor (MRCA) at the base of the Hymenoptera tree would require extensive outgroup sampling. However, we believe the reviewer has misunderstood the results we describe here, and equates the MRCA of Hymenoptera with the MRCA of the “symphytan” lineages (which does not exist, as they are a paraphyletic assemblage). Our analyses indicated rate decreases on the *individual* branches leading to Tenthredinoidea, Siricidae + Xiphydriidae and Pamphiliidae + Xyelidae compared to the overall background rate within Hymenoptera, respectively. These lineages also show lower net diversification rates compared to the rate calculated across Vespina. We do not need outgroup sampling to infer this imbalance of net diversification rates within Hymenoptera (in fact all outgroups were excluded from dating and diversification analyses to prevent artifacts). We have tried to clarify in the text to prevent further misunderstanding of the nature of this comparison (lines 803-805).

Lines 410–411: “Taken together, these results highlight the role of secondary phytophagy, but not phytophagy per se, in the more recent diversification history of Hymenoptera.” Because the dataset does not allow to infer the speciation rate outside Hymenoptera, it cannot allow assessing whether the speciation rate in phytophagous sawflies is elevated or reduced. The dataset therefore cannot make a statement on whether or not phytophagy per se has an impact. The dataset is only (if at all, because of a possible imbalanced sampling of the ingroup) suitable to assess the impact on secondary phytophagy. However, the dataset focused on two lineages (bees, gall wasps) that were likely to show accelerated speciation rates based on prior

knowledge. What is missing is a corresponding analysis on other lineages, such as pollen wasps, that are not known to be particular species rich.

As outlined immediately above, the reviewer possibly misunderstood our reasoning and support for this statement. We have therefore left this statement unchanged.

With regard to our sampling of secondary phytophagous clades, we have already explained above that our family-level diversification analyses necessarily lack these details at lower taxonomic levels. We mention the species-poor secondary phytophagous clades now in the discussion as counter-examples.

Line 411–413: “This leads to the question: why was phytophagy more prone to induce diversification in Hymenoptera when arising from parasitoid lineages?” The presented data do not allow to deduce this question, because of the lack of a benchmark. As outlined above, the lack of a proper outgroup sampling does not allow to qualify (increased/decreased) the speciation rate in the phytophagous sawfly lineages. The presented data at most allow to state that in those lineages that secondarily switched to phytophagy the speciation rate increased. Please see the above explanation. The benchmark for comparison is not the outgroup rate, but the background rate from which the rates estimated for the respective sawfly and woodwasp lineages originated. No changes to the text were made.

Line 413–414: “What adaptations does a “parasitoid converted-to-phytophage” possess that may confer an evolutionary advantage over a primary phytophage?” Same concern as in the previous comment: we lack a proper benchmark to conclude this!

Same explanation as above. No changes to the text were made.

Lines 415–426: I can follow the logic. What I am missing are additional hypotheses. Assuming that the speciation rate of pollen wasps is not elevated, laying eggs directly at the food source seemingly was not sufficient to increase the groups speciation rate. Life is about competition. Escaping from the competition within a guild can result in at least temporarily more available resources. What works for one group at a given time (e.g., bees) may not work for another group (e.g., pollen wasps) at a later time, because the resources are already competitively used (e.g., by bees).

We have fleshed out this hypothesis further (lines 821-823, 830-832), also based on reviewer 2’s comments.

Lines 429: “phytophagy ... can arise via different genetic and ecological pathways”. Please be more specific and provide references.

This paragraph has been revised for conciseness and this statement was deleted.

Lines 457–459: “That parasitoidism was a superbly successful strategy in Hymenoptera cannot be disputed given its long dominance in Apocrita and the much higher species diversity of parasitoids in the order.” The dominance of parasitoidism includes also Orussoidea, and hence Apocrita should be replaced by Vespina. I do not understand the difference in what the first and what the second part of the sentence is meant to state. Dominance means that one thing is more present than another. To my knowledge, once there were more parasitoid Hymenoptera species than non-parasitoid Hymenoptera (=> parasitoids became dominant), this dominance was retained. Hence, the long dominance has lasted till today (or am I wrong here?). It follows from pure logic that if the life style has been and is dominant that this life style is represented in most of the species. Thus, the second part of the sentence bears no additional information.

We have modified this statement by replacing “Apocrita” with “Vespina”, and deleted the second half of the sentence.

Lines 465–466: Based on what criteria were species selected to allow for a species diversification analysis. Please make transparent that the species sampling representatively depicts the species diversity in the different families (e.g., 0.5–1.0 % of the known species in a family were randomly sampled).

We have added a statement on the sampling criteria in lines 959-961: “Our taxon sampling aimed for representation of major lineages within families while sampling across the respective

root nodes on the family-level, covering between 0.06–50% (=1–150 representatives) of the described species diversity”.

Lines 465–476: Please provide data on where samples were collected when and by whom. Supplementary Table 1 only mentions voucher numbers. Transparency in sample information is important given the legal aspects related to the analysis of DNA of samples from various countries (Nagoya agreement).

We included the requested information in Supplementary Data 1.

Lines 494: The third codon position in nucleotide sequence data is known to bias phylogenetic results because it frequently violates the assumption of homogeneity. Why has the third codon position not excluded, given that it also likely contributes little phylogenetic information?

We kept the 3rd codon position to ensure comparability with the full data set, where no distinction was made between coding and non-coding data. We have clarified this in the manuscript (lines 998-999).

Lines 509–511: Please provide details based on what criteria the fossils were selected? The table with the fossils does not outline why a fossil was assumed to be part of a clade (e.g., by providing autapomorphies).

As stated in the main text (lines 1009-1012), fossils were chosen to represent the oldest and most reliable available calibration points for superfamily and family-level nodes. We have inserted a new column in Supplementary Data 13 (“Justification for nodal assignment”) providing more details on each fossil assignment.

Line 569: I noticed thought the manuscript that only parasitoidism and predation were considered. However, these categories capture only part of the hymenopterans’ life styles. A substantial number of Aculeata are kleptoparasites, either carnivorous or phytophagous ones. What was done with them?

Our approach was to code higher-level taxa (mostly families) based on the lifestyle expressed by the majority of their members. On the family-level, there are no taxa that consist entirely or to a majority of kleptoparasites. Kleptoparasitism was thus not considered as an independent strategy, but lumped depending the strategy of the host and their food resource. We have inserted a clarifying statement to this regard in the supplemental material (p. 9, section on character evolution).

Figure 1 and all other phylogenetic figures: I noticed that Crabronidae s. str. is polyphyletic. This is because of the placement of *Ectemnius_sonorensis_BND831* in the family Sphecidae. The same relationship was already presented by Branstetter et al. (2017) and it was heavily criticized in the community, because virtually all evidence suggests that *Ectemnius* is very closely related to *Crabro* — the type genus of the family Crabronidae. The most likely explanation is that the identity of the analyzed DNA is not correct (e.g., by mixing up of samples). The identity must thus be verified. If the placement of *Ectemnius* in the family Sphecidae is confirmed, it should be discussed in detail.

We thank the reviewer for alerting us to this issue. We were not aware of the doubtful identity of this sample, otherwise we would have tried to clarify before including this sample in our analyses. We were able to extract the full 658bp cytochrome oxidase I barcode region from the raw assembled contigs for the “*Ectemnius_sonorensis_BND831*” sample. A BLAST search identified a 100% match in the NCBI database to *Prionyx canadensis* (a Sphecidae). Personal communication with Bryan Danforth (second author of Branstetter et al. 2017) who extracted the DNA for this specimen revealed that a *Prionyx* spp. sample was processed in the same extraction series. Since the voucher specimen of *Ectemnius_sonorensis_BND831* is definitely *Ectemnius* (Danforth pers. comm.), there must have been a mix up of samples during extraction. We see all this as sufficient evidence to change the identification of this sample to *Prionyx canadensis* for the context of our study and have made updates accordingly throughout our manuscript files. Fortunately, we had already counted this terminal as a sphecid in the context of our diversification analyses, therefore no updates to analyses or results were

necessary.

Reviewer #2 (Remarks to the Author):

In their paper "Key innovations and the diversification of Hymenoptera", the authors address different hypotheses on how certain traits could have acted as key innovations to give rise to the diversity of different hymenopteran clades.

The manuscript is well written and structured logically. It appears that much care was taken at each analysis step to run the required tests and checks to ensure all was done according to best practices and that the obtained results are trustworthy. I commend the authors for their effort there, and also for the effort taken towards transparency and replicability by depositing raw data and analysis scripts for readers to review.

Overall, I think this paper addresses a very relevant and interesting question concerning the much debated reasons behind the high diversity of some hymenopteran clades. The improved phylogeny and application of established methods in the field to test these exciting hypotheses represent meaningful advances in this field.

We thank the reviewer for this positive feedback (and also the constructive comments below).

Despite this, I have a few major points of criticism towards this paper:

1. There is a general issue with the concept of key innovations. While they are a very appealing idea and since the rise of SSE models have stimulated a lot of exciting research into the topic, I think it has been repeatedly stated that the concept is probably overly simplistic and should only be employed with care. The authors do cite Rabosky 2017 regarding this towards the end of their manuscript, but I believe that is not sufficient to address this issue properly. The implications of this should be stated upfront (i.e. in the introduction), and their impact on the used approach and the interpretation of the results should be mentioned again wherever needed. There exists more literature on problems with the concept, as well as improved ways of thinking about it, e.g. in Donoghue 2005, Donogue & Sanderson 2015, or maybe also Bouchenak-Khelladi et al. 2015 (for first steps towards addressing this methodologically), and these or similar studies should be considered to address this issue.

We now mention the criticism on the traditional concept of key innovations in the introduction (lines 92-96) and have also expanded on this issue in the discussion (lines 633-648; we moved and expanded some relevant text from the supplemental discussion into the main text). Thank you for pointing us to the very relevant references which we cite now as appropriate, and we also consider some of the herein proposed terminology as it applies to our results.

2. Similarly, there are issues with some of the diversification methods employed, namely the SSE models. While they do hint at this somewhere, I believe the authors should cite Maddison and FitzJohn's 2015 caveat about unreplicated trait- and rate-shifts, and how one can't necessarily be distinguish whether they are causal or merely coincidental (HiSSE helps to address this, as the authors rightly point out, but not entirely). This caveat should influence the interpretation of the results too.

Beyond this, Louca & Pennell (2020) have pointed out an identifiability issue with diversification models, and while they did not address those in SSE models per se, there is no reason to assume those issues wouldn't extend to those at least in some way. Different other authors have attempted solutions for this, but there is not definitive solution for those yet, so this caveat should be addressed somewhere.

We have elaborated on the issues of unreplicated traits and difficulties in estimating net diversification rate from timetrees in the discussion and cite Maddison and FitzJohn (2015) and Louca and Pennell (2020) in this context (lines 649-656).

3. I would generally welcome if the authors were to elaborate the suspected mechanisms/scenarios for diversification more. I am pointing this out at various points in my line comments too. It is in general very tempting (and has probably been practiced in the field for too long), to remain rather vague when explaining why a chosen trait should convey higher diversification rates to the organisms that carry it (or why not). This is understandable, as there is still a gap in understanding between microevolutionary processes of speciation and extinction, and the study of their macroevolutionary dynamics. However, I believe it is important to think more deeply about those and make sure those thoughts enter the hypotheses posed. Such arguments could e.g. go along the lines of 'trait X allows the species that carry it to explore new niches, which could facilitate speciation as they become (reproductively) isolated from their conspecifics as they specialise into those different niches', or 'trait Y conveys higher survivability, thus potentially lowering extinction rates in comparison to their sister clades'. The authors are at times getting pretty close to making such arguments, so in those cases they may only need to be spelled out a bit more explicitly. Since the authors seem to possess great knowledge of hymenopteran biology, I believe this puts them in a prime position of devising such scenarios/mechanisms to enhance their hypotheses.

Thank you for this suggestion. We have fleshed out the suspected mechanisms and scenarios following the reviewer's suggestions in the line by line comments. Please see our responses below in the line comments regarding the specific changes made.

4. Vaguely related to the previous point: Interpreting the HiSSE plots (e.g. see line 241ff) is great, though I think some of these would become clearer if actual rate estimates were looked at. Indeed, I couldn't find parameter estimates for these models anywhere, only the numbers on model fit. I think those numbers should be shown somewhere and interpreted, as it would make it much more transparent what the actual rate differences are between the different rate categories (i.e. combinations of hidden and observed rates). It could even make the results a lot more impactful by interpreting the rate estimates too. In theory, it would allow to determine whether a trait leads to rate increases in all cases, but maybe of different strengths, or whether rate increases only happen in a subset of clades, etc. It also allows to spot whether a rate difference that is supported by model-testing only relies on a very small actual difference, which may be considered biologically negligible. Besides that, Beaulieu & O'Meara emphasize in their original paper introducing HiSSE, that considering the parameter estimates may be more important than merely focusing on which model fits best. Looking at inferred rates on the plot can help intuitively interpreting those values, but shouldn't be relied on (notably, I think it's technically possible to reconstruct the actual inferred hidden states, though I'm not sure how straightforward that is, and it may not be necessary).

Thank you for the constructive suggestion. We summarized the parameters (i.e. transition and diversification rates) estimated from best-fitting models in a new supplementary table (now Supplementary Data 11) and refer to the diversification parameters in the results where appropriate (lines 465, 469, 534, 536). Indeed, the estimated rate parameters confirm the rate increases/decreases in the observed traits as visualized in the plots and confirm our conclusions. In most cases, we see >100% increases in net diversification rate (min. 33%). While retrieving the parameter estimates from the results for the state reconstructions, we noticed that for one analysis (the wasp waist character estimated from topC-1) we had incorrectly reported the best-supported model as the full HiSSE model instead of the full HiSSE model with irreversible states. We corrected this error in Table 4, Supplementary Data 10, Figure 4 and throughout the results section of the main text. None of the conclusions change; in

fact, this result is more logical given that the irreversible model was also estimated for topA-0 for the wasp waist.

5. Having read the rationale for the majority rule approach for ancestral trait reconstruction in the supplementary discussion (and on line 566ff), I have to say that while I can acknowledge the problem described, I am not fully convinced by the chosen solution.

Firstly, if the species level values are not representative of the missing diversity in the tree, then a clade-level assignment would be just as unrepresentative, assuming that you based this on information on the same taxa. If this was the result of surveying more species than are in the tree, to get a more accurate view on the character diversity in the group, then that would make this choice appear more justified, however, I could not find an indication anywhere that this is what you did, so this should be clarified.

Indeed, this approach was chosen to most accurately represent the entire character state diversity in each clade, thereby also incorporating information from unsampled taxa and avoiding any bias by over- or underrepresentation of states (mainly important in polymorphic groups). We have clarified this in the methods section of the main text (lines 1091-1093), and in the supplementary discussion (p.46).

Indeed, I couldn't really tell what the source of the trait data was to begin with, i.e. did you score it yourself from samples, or from the literature, or both? This also should be clarified.

Mostly we used literature records, but also incorporated unpublished expert knowledge. This information is somewhat hard to find in Supplementary Data 14, last right-hand column, where we list the references used to score each clade. We have added a note in the methods pointing to this table for the references (line 1103).

Secondly, assigning the trait state by those clades may bias the result towards finding coinciding shifts in diversification rates and trait state at the base of those clades, while in reality, the trait change may have happened much later in a subset of the clade, depending on how the trait is distributed within.

We understand the reviewer's rationale in suggesting that our clade-level approach may have resulted in a bias of shifts occurring prior to or at the base of clades. In fact, the Medusa estimates are necessarily limited to shifts occurring prior to, or at the base of clades since they were performed on a clade-level tree only. However, the BAMM diversification rate analyses were performed on the species-level tree and completely independent from traits. These analyses did recover shifts within clades for Ichneumonidae and Eurytomidae, and the rate increase in Eurytomidae was also recovered by HiSSE where trait information was explicitly incorporated in the model. Because of the agreement of the different methods in these cases regardless of whether trait information was included or not, we think that a bias due this procedure is unlikely. However, we have inserted a short discussion about the possibility of this issue in the supplementary discussion (p.49): *"It is further possible that our clade-level approach to character coding could have resulted in a bias of shifts occurring prior to or at the base of clades. This is certainly true for the Medusa estimates as these were performed on a clade-level tree only and necessarily limited to shifts prior to the base of these clades. However, HiSSE recovered a rate increase in Eurytomidae within the larger Chalcidoidea clade, a result that was confirmed by BAMM analyses performed on a species-level tree and excluding all trait information. BAMM analyses also recovered a shift within Ichneumonidae, giving further reason to believe that the placement of shifts in these analyses was unbiased and not influenced by our character coding."*

I would also presume that this simplified coding homogenised the rates of character evolution across your tree, which may be what is behind your result that the ER model fits best. Possibly, an analysis on species-level (or any other level) trait data would result in ARD to be a better fit, or at least would have affected the inferred rates and thereby the result. It is well possible that

the result wouldn't change enough to topple your current interpretation, but this can't be said confidently as it stands.

Certainly, changes in character coding with regard to the level of detail of the analysis could result in different estimated rates of character evolution. This necessarily always has to be a concern if not every single species is represented in the phylogeny. Since we cannot perform a species-level analysis (likely the natural history and phylogenetic data necessary to do this in Hymenoptera will not exist for many years to come), the clade-level coding was chosen as the best possible approach to most accurately represent the entire character state diversity in each clade, and to avoid any bias by over- or underrepresentation of states.

To address the reviewer's concern, we have compared the reconstructions under the ARD model to those from the ER model and present the ARD results now as Fig. S19 in the supplementary material as well. This shows that the reconstructions between the two different models are essentially the same and would not change our conclusions even if the ARD model had been favored. Of course, in an analysis on the species-level as the reviewer suggests, the differences in transition estimates between the two models could be more pronounced. But at the same time, we would argue that the evolutionary history of the considered traits is relatively straightforward, with few independent origins or state reversals, which leads us to believe it unlikely that the reconstructions would turn out majorly different. Most importantly, the goal of our study was to highlight broad evolutionary patterns in Hymenoptera, focusing on shifts in the early diversification of the order, and we think our approach is appropriate for this objective given that it is impossible to perform this analysis on a complete phylogeny.

We have inserted the following text in the supplemental discussion (p.47) to address these concerns: *"A general concern with ancestral reconstructions performed on incomplete phylogenies is that changes in the level of detail of the analyses and associated character coding could result in different estimated rates of character evolution and conclusions. In our case, coding on a clade-level basis may have homogenized the rates of character evolution estimated across our phylogeny, potentially leading to the ER model being recovered as best-fitting. An analysis on species-level trait data or coding on a different taxonomic level may result in the ARD model to perform better, affecting the inferred rates of character evolution and possibly the reconstructed states. We therefore compared reconstructions under the ARD model with those from the ER model, and confirmed that the reconstructions under the two different models are essentially the same, indicating that our conclusions on trait evolution are robust to changes in estimated transition rates. As outlined above, the clade-level coding was chosen as the current best approach to represent the entire character state diversity in each clade and to avoid bias by over- or under-representation of states. We argue that this approach is appropriate for the goal to highlight broad evolutionary patterns in Hymenoptera by focusing on shifts in the early diversification of the order, although it presents a necessary approximation to a species-level approach."*

Furthermore, I was wondering whether you made any assumptions regarding the root state when reconstructing the trait states? The root state of many of your reconstructions seems (unsurprisingly) strongly affected by the states of the groups that are sister to the rest of the tree, but since you excluded the outgroups, you cannot tell if this is accurate. The outgroup sampling may not make it appropriate to include them in the reconstruction as is, but maybe with some knowledge on the trait states of the closest relatives to hymenopterans may allow you to make an informed decision on setting any prior probabilities on the root states.

We did not incorporate any assumptions regarding the root state (hymenopteran crown node) in our ancestral reconstructions or trait-dependent diversification analyses. Our current understanding of insect evolution (cf. Misof et al. 2014) posits that Hymenoptera is the sister group to all other ten holometabolous insect orders (e.g., beetles, flies, butterflies & moths, etc.). Thus, there is no one sister group from which a prior could be derived. However, most hymenopterists would agree that the traits exhibited by the sister groups to the rest of the tree,

the sawfly and woodwasp lineages (no wasp waist, no stinger, primary phytophagy) are likely plesiomorphic for the order. Since these are the root states estimated by our analyses from the ingroup data (with root states unconstrained), we think that implementing a prior on the root node would not provide an improvement to the ancestral reconstructions. We therefore politely decline to follow this advice but thank the reviewer for the comment as this would certainly be an important point in the context of many other data sets.

All in all, it is clear that some simplifications were necessary given the data you are working with, but if those simplifications are potentially biasing your results in some way that matters for the hypotheses you are trying to test, the potential implications of this have to be addressed by tests (if possible) and in the discussion. So at least, I would suggest you reconstruct ancestral trait states on a species-level tree as well, and compare/discuss the resulting patterns to your current approach. The actual species level data seems to exist, after all you are using it in the HiSSE analyses.

We have already explained above that the species-level data do not exist, but reiterate here in case clarification is needed. The HiSSE analyses used a “species-level tree” (in the sense that all sampled species have been included as terminals) but each member of a clade was assigned the same clade-level state. We have clarified this in the method section, as maybe our previous explanation was not sufficiently clear. While performing data analyses, we had initially discussed an approach including species-level data. However, we reached the conclusion that it is not feasible to assign states on the species level as too many taxa are missing information about their biology, and a thus coded data set will further not be representative of the true trait diversity within the group. The generalized approach taken in our study is unfortunately the only one that is feasible at this point in time. We have already discussed this topic in the supplemental discussion (p. 46-47, first paragraph on under “Methodological considerations for macroevolutionary analyses”).

Apologies if some of those points are indeed mentioned somewhere, although I searched both the manuscript and supplementary material for it and couldn't find it, which may suggest that it has to be stated somewhere more obvious or had to be referred to more explicitly.

We have tried to clarify as much as possible and hope you will find all points answered sufficiently.

A few of those main points are echoed in my specific line comments below again too.

On the side, I would like to point out that there are multi-state SSE models, such as MuSSE (in diversitree for regular state-dependent diversification) or MuHiSSE (Nakov et al. 2018 for a hidden-states version) - or any custom variations of such one could devise of in RevBayes -, which might be preferable over multiple binary-trait analyses of what is really a multi-state trait (e.g. parasitoidism vs. carnivory vs. secondary herbivory etc.). While I wouldn't want to insist on the use of these for the paper at hand, I feel the authors may want to consider them, even if for future work (and maybe still address somewhere in this manuscript why they decided against them).

Similarly, the authors may want to consider work by Braga et al 2020 and 2021 on a new Bayesian approach to analyse ancestral host-parasite interactions. While the available data might not quite allow for this yet, this might be an avenue to consider in the future. As they mention themselves e.g. on line 305, the connection of parasitoidism and diversification hinges on the idea of the parasitoids tracking separate particular host lineages, which could be tested/established using such an approach.

We did indeed consider some of these options (MuSSE and MuHiSSE) to jointly analyze multiple traits (rather than states). However, we decided against this approach as we specifically wanted to evaluate support for each trait as a key innovation independently. In addition, the MuSSE model suffers from criticism to the SSE groups models brought forth a while ago, so we

were also reluctant to use it. We clarified this reasoning in the method section of the manuscript (lines 1119-1122).

The method used in the suggested Braga et al. papers seems interesting, although not directly applicable to our data as we would need much more detailed host data. We agree that incorporating some multi-state models in the future could present interesting avenues and thank the reviewer for all these suggestions.

Overall, I very much enjoyed this manuscript, though I strongly feel that my points above should be addressed in a satisfactory manner before publishing. This may include some of my minor comments below too. I hope the authors perceive my comments as constructive and potentially enhancing this exciting work further, and that they forgive any criticism that was merely the result of me misunderstanding their writing. I am looking forward to seeing an improved version of this work published in the near future!

Thank you!

Specific comments by line number: [Some of these may be shifted by down by a few lines due to some edits I made, though I hope this won't be a problem in finding what they refer to]

59: calling them 'singular' struck me as odd. I would have called them 'unique' or something the like, but this may just be me, it is definitely clear what is meant.

Changed to "unique".

59ff: It is unclear to me what those reasons are - the fact that a very diverse clade is containing a lot of parasitoids? The hypothesis seems to be that they diversify because they adapt to different hosts, maybe this needs to be stated more explicitly and elaborated a bit?

We expanded the statement to read "Parasitoidism and its associated features may be key drivers that explain diversification in Hymenoptera, as the adaptation to different host species and therefore niche subdivision may have resulted in increased speciation rates in parasitoids." (lines 126-128).

64: not sure why 'the stinger' is in quotation marks?

We removed the quotation marks.

66ff: Also here, is the suspected connection between diversity and stingers that the defense possibilities of it decrease extinction risk? Or something else? This should be clarified and referred to when discussing the results.

We clarified this as requested in lines 200-202.

68ff: I appreciate you not calling them 'basal', though I believe technically 'early branching' may also not be entirely correct. If you can find a way to reword that along the lines of "sister to the rest of the clade" or something the like, that might be good.

Rephrased (lines 130-131).

71: I've not encountered the term groundplan before, only bauplan or body plan, though that might just be ignorance on my side.

The term groundplan is often used for insects (equivalent to bauplan) and we have therefore kept it.

86ff: Maybe I'm nitpicking here, but I don't think they can be considered catalytic if that hasn't been formally tested yet. They 'could' be seen as such, or be suspected or hypothesised as such, of course.

Changed wording to "could".

88: Maybe nitpicking too, but you are technically analysing them separately (albeit in the same study), and not jointly (i.e. in the same analysis).

Very true, thanks. We removed the word "jointly."

141ff: I suppose the last sentence here is a foreshadowing on how results using topology C-1 and A-0 gave qualitatively similar results?

Correct. We have tried to clarify the wording (line 297).

150: You don't seem to be using the acronym ACR anywhere else in the manuscript, so since it afaiK isn't standard usage in the field, I'd suggest dropping it for simplicity.

Removed as requested.

160ff: As stated as a major point in the methods section, it should be discussed whether the location and frequency of these shifts could be affected by how the traits were coded by family. Please see our response to this comment above under major point 5). This concern is now addressed in the supplemental discussion.

162: Maybe write out 'node' for clarity.

We spelled out "node".

242ff: along those lines, it should also be looked at and discussed, how the reconstructions and transition rate differences compare to those resulting from the ancestral trait reconstructions. Some differences can be expected - also because SSE models were initially thought up primarily as a way to account for diversification when reconstructing traits, rather than as a way to test for trait dependent diversification - but nevertheless inconsistencies should be addressed at least briefly.

Thank you for this suggestion. We have summarized transition and diversification rates estimated from best-fitting HiSSE models in Supplementary Data 11 (see also our reply to major comment #4). However, these cannot be compared directly with the transition rate differences in the ancestral trait reconstructions, since our main ancestral trait analysis (reconstructing life histories parasitoidism, phytophagy, predation and secondary phytophagy) was performed as a multistate reconstruction, while the HiSSE models reconstructed binary ancestral states.

250: I believe it should say Vespina

Corrected, thanks.

272: Oxford comma after 'regime', if you subscribe to that

Added.

289: Might be worth spelling out that the evolution of secondary phytophagy does not necessarily have to coincide with the onset of angiosperm diversification in order to be caused/affected by it.

We elaborated as suggested, please see lines 595-597.

305ff: Again, I think the reasons/mechanisms why the long history and tracking host lineages would have led to higher diversity (though it's more intuitively clear for the latter).

We elaborated as suggested, please see lines 629-630.

318ff: Inspecting the rate estimates more closely could help elucidate that idea.

This is reiterated from major point 4), please see our response to this comment above.

321ff: Might be worth exploring whether any rate increase there might have been interpreted as a rate decrease in the sister-taxa of Apocritans (or generally members of the grade before)?

We discuss the implications of rate decreases recovered in non-apocritan lineages; please see the expanded discussion in lines 684-689.

324ff: Maybe it could be discussed as a preadaptation/background trait (sensu Bouchenak-Khelladi et al. 2015 and therein)? Especially since it seems that having that kind of waist is critical to being a parasitoid (unless I have misunderstood that)? However, it should also be noted that SSE models are likely experiencing issues if a trait is too rare, which may be the case for wasp waist too.

Thank you for the suggestion. However, R1 pointed out that because parasitoidism is (most likely) ancestral to Orussoidea and Apocrita (and not exclusive to Apocrita) and stinging has only evolved once in Apocrita, a direct link between these traits and the wasp waist cannot be tested and should not be suggested. We therefore refrained from suggesting this trait as a preadaptation or precursor.

325ff: I very much like the careful wording here!

Thank you.

329: the colours of the BAMM plots should be interpreted with care. The initial decrease could also be the result of high turnover, etc.

Thank you for this observation. We assessed the clade rates at nodes immediately prior to the non-chrysidoid aculeate node, but were not able to pinpoint a decrease in net diversification within those data. It is possible that the rate decrease visible in the BAMM plot may actually occur post the non-chrysidoid aculeate node. In any case, the rate change appears very small and therefore we have removed the mention of this “initial decrease” from the discussion. We also appreciate the point about not relying solely on interpreting the colors of the BAMM plots to judge rate increases/decreases. We have focused now more on interpreting the differences in rates between focal clades vs background rates (Table 3) and made minor adjustments in this regard to the results section (lines 362ff).

341ff: I like the idea that the stinger only had an impact once other modifications were in place too (making the stinger a preadaptation/background trait again, maybe). However, one could argue that a HiSSE model should potentially capture that, by finding elevated rates for a combination of the stinger and a hidden state nested within this clade. Support for character independence also must not mean that the stinger had no effect, but e.g. if there's elevated rates in clades outside the stinger-bearing hymenopterans (or nested within, etc.), the rates within them would conflict with the impact of the stinger - and after all, you are looking at other candidate traits which could cause this, so potentially other traits affecting diversification might cancel out the signal of this one. This is purely speculative of course, but again looking at rate estimates (and possibly the location of the hidden states on the tree) could help interpret this. We appreciate the comment and agree with the reviewer that support for a trait-independent model may not necessarily mean that the stinger had no effect on diversification in Hymenoptera at all. We calculated net diversification rate estimates for Aculeata in BAMM analyses, which were also elevated compared to the background rate, though not as much as in Aculeata minus Chryridoidea. However, the support for the trait-independent model for the stinger over the next best trait-dependent model was substantial (32-44 difference in AIC score). We chose not to attempt to explore this “stinger as preadaptation” hypothesis further with our data as veering away from the statistical tests while digging deeper into the raw results becomes increasingly subjective. Instead, we have inserted a paragraph in the discussion acknowledging the potential caveat of having interfering rate variation across the remaining tree (lines 721-726).

347ff: A few things I pointed out for traits above apply here too.

We are unsure what the specific issues to address here are. We think that this comment may be directed to our interpretation of BAMM plot colors in this paragraph. However, the net diversification rates we extracted from HiSSE (Supplementary Data 11) based on the reviewer's suggestion also point to a slight decrease in rate for parasitoidism/Vespina, therefore we have not made further modifications to this paragraph.

352ff: This is a very interesting result/interpretation, especially since coding by carnivory only adds a few species (23 I think) to the set, when compared to parasitoidism. I would encourage to investigate this more (also looking at inferred rates and location of hidden states, I may add - with the hazard of sounding like a broken record). It's technically possible that this only reveals that the positive result for parasitoids was just driven by some odd bias, of course, which one might discover that way as well.

We agree with the reviewer that the comparison of model fit for parasitoidism and carnivory shouldn't be overinterpreted as the adjustments to the trait matrix between analyses are quite small. We have moderated this interpretation (lines 735-736). The suggestion to investigate this further is rather general and we do not see how the inferred rates or hidden states would be helpful in this case. Therefore we have not addressed this further.

361ff: I find this point about the scenario for delayed effect of parasitoidism very interesting and

indeed in line with the initial theory of adaptive radiations sensu Simpson, i.e. requiring both the key innovation as well as entering the adaptive landscape - which would here be the presence of an abundance of suitable hosts (this is sort of what I meant above regarding angiosperms too).

We have elaborated on this scenario in the discussion – please see lines 760-763.

414ff: Again, the scenario needs to be fleshed out more to justify why increased larval provisioning would accelerate diversification (i.e. whether and how this leads to more speciation and/or less extinction)

We have fleshed this scenario out more as suggested; see lines 821-823 and 830-834.

417ff: The idea of a 'pre-innovation' would probably connect to related concepts in Donoghue et al. 2015 or Bouchenak-Khelladi et al. 2015?

Thanks, we agree that these ideas are connected, and now cite Donoghue and Sanderson (2015) in this context.

440ff: I like the idea of the modification of 'key innovations' as an actual driver of diversification (like a modifier sensu Bouchenak-Khelladi et al. 2015), and would be looking forward to seeing this addressed more in future work! Same for the analysis of the subcomponents mentioned at 447!

Thank you, we hope as well that this can be explored further in the future, by us or others.

459ff: I think it would be good to explicitly state what coverage these taxa represent at different taxonomic levels (I'm aware this information can be found in table S13, but maybe a coverage-range might be helpful for readers to get a rough idea of the sampling heterogeneity).

We have added this information in line 960.

467ff: The apparent availability of the raw data is great!

Thank you.

477: "50%, 60%, and 70%"

Corrected.

490/491: Oxford commas before the 'and' in both cases here (and throughout), if you are partial to that

Added.

508: Not sure, but think 'Additional' should instead either say 'Additionally', or better 'In addition'?

"In addition" sounds best, thanks.

508/515/516: It's not immediately clear to me here how those maximum and minimum bounds are made to be soft. I believe this is usually meant to say that the prior distribution has some kind of tail where the probability decreases gradually. Is the truncated Cauchy distribution referenced later on meant to describe these soft bounds (and all of them)? It would seem a truncated distribution would have a hard bound rather (unless the truncated end is on the side of the present time), so maybe this should be clarified a bit. Also, if possible add a short comment on why the default settings are justified here.

The full description of the prior distribution (per PAML manual) would be "a heavy-tailed density based on a truncated Cauchy distribution". The Cauchy distribution as implemented is indeed truncated at the calibration point, but made soft by adding a tail ($a=2.5\%$) left of the calibration point. We clarified this in lines 1025-1027.

We used the default parameters (offset $p=0.1$, a scale parameter $c=1$ and a left tail probability of $a=0.025$), as these represent a relatively flat, uninformative distribution which we deemed most appropriate for the almost exclusively stem-group fossil calibrations that were available to us. We realize that there are caveats going along with using uninformative priors (for example, the posterior densities estimates are fairly wide), but think this is preferable to using priors that are informative, but potentially misleading. We added a comment about why the default settings were used in the supplementary discussion (p.7).

511ff: Maybe I missed this, but what is the rationale for using these two data sets specifically here?

We used the best-supported and the most frequently recovered topology across our phylogenetic analyses to perform the divergence dating and macroevolutionary analyses. The reasoning for this was already presented in the results section (lines 292-296) but we have added a clarification to the methods section as well (lines 1018-1020).

519: For the effective sample size, what was your cutoff to signify sufficient convergence (and did all parameters reach/exceed it)? I presume the ESS were post-burnin too?

We used a threshold of 200 for ESS (post-burnin). Most parameters well exceeded this value in individual runs, but with a phylogeny of >750 nodes, some parameter estimates only reached the threshold after combining runs. We have elaborated on this in the methods (lines 1030-1034).

520: I got a bit confused with the post burnin samples here. You said you ran 2mio samples, subsampling every 10th, but then each run has more than 2mio states even after discarding burnin? I'm sure I'm just overlooking something that's maybe implied here, but that might suggest this needs to be clarified a bit better.

Thank you for this observation, indeed these specifications were summarized in a confusing way. The analyses were set up for 2 million samples with sampling frequency of 10, which equals 10x2 million states or chain length. However, we stopped the analyses (after 8 months on a computing cluster) once convergence was reached. We have clarified this, clearly distinguishing between how many states were run and how many samples were summarized. Please see lines 1030-1035. We also realized that this passage had not been adequately revised from a previous version of the methods; we have now thoroughly checked and corrected the specifications.

528ff: I believe assigning missing sampling by clade - while being an important correction - has the potential to bias the analysis towards finding shifts at the base of the clades to which the sampling fractions were assigned. I am not aware of a formal way to test for that (or whether the taxonomic level at which you chose to assign missing sampling has an impact on shift location), but maybe this can be addressed somewhere as a possible caveat?

The reviewer already pointed this out under major comment #5. We have acknowledged this caveat in the supplementary discussion (p. 47) as outlined above.

Also, in table S13, some taxa inexplicably have a slash instead of the sampling fraction, including one of the clades that have a shift, it seems unclear whether this is an error or whether it means something.

The taxa with a slash filled in instead of the sampling fractions were lumped into larger clades for the macroevolutionary analyses. This concerns, for example, all of chalcidoid families. We have now clarified in supplementary data 13 (now suppl. data 14) what the "/" represents.

543: You do mention the low frequency of the MAP shift configuration is, and I agree that the cumulative shift probability is a good way to deal with this problem. However, I would have liked to see a plot of at least the few most common configurations and their frequencies, e.g. somewhere in the suppmat, to get a better intuition for just how representative the MAP configuration is.

We have added a new supplementary figure (S21; other suppl. figures have been renumbered) that depicts the 9 most common plots from the credible shift set for both topologies.

546ff: As you point out correctly here, Chang et al. suggest that the taxonomic method is preferred when sampling fraction is below 1%, which seems to be the case for your data. However, when comparing the total number of taxa in the whole group vs represented in your tree between what you describe on line 398 (7882+144809 = 152691 total) and line 99 (765 taxa sampled) vs. the numbers shown in table S13 (181358 total, 930 sampled), it might strike some as confusing why those numbers differ. The overall sampling fraction of both number pairs comes out to be pretty much the same (0.0051 vs 0.005), so I'm sure the difference relates

to something reasonable that doesn't reveal itself to me here, and probably doesn't affect the result, but you might note somewhere why the numbers differ, to clarify that.

The species numbers quoted in the manuscript are correct. The species numbers in supplementary data 13 (now 14) cannot be directly added since they are listed both on a family basis and for the larger clades that lump some non-monophyletic families for analyses, such as Chalcidoidea and Diapriiidea. Thus, for some taxa, the species numbers are represented twice if one simply adds all the numbers in the table. We have clarified this by changing the font to grey for those families that are lumped for the purpose of macroevolutionary analyses and adding an explanation to the table caption.

581: I am very pleased to see the thoroughness in testing alternative models here!

Thank you.

590ff: It technically becomes apparent from the plots and supplementary tables (and esp the caption of figure 4), but I think it would make things clearer if you would state somewhere around here which topology you used for these analyses, and that it was on the full tree and not the one collapsed to family level again.

We clarified this as requested (lines 1131-1132).

Figure captions:

- I thought citing references in figure captions was unusual, but I'll trust the journal has guidelines for that.

We think the references in the figure captions are useful and allowed per journal referencing guidelines, so we kept them. But we gladly follow any additional editorial recommendations in this regard.

- I overall find the figures very clear and aesthetically pleasing!

Thanks, we appreciate the positive feedback on our figures.

- Figure 3: the word darkcyan should probably be separated into two; the signs for the shift probability should probably be switched to 'smaller or equal to 0.99'

Both corrected, thank you.

- Figure 4: dito for darkcyan; I like the colour scheme overall, but was wondering whether the choice of grey for absence of the trait makes it blend in a bit too much with the outer colour (though HiSSE plots are notoriously difficult to make clear)

We corrected "darkcyan" to "dark cyan". As for changing the colors for the states in HiSSE plots: we feel we have already experimented with a lot of different color schemes for states (and rates) and these work best in our opinion, so we kept them as is.

Reviewer #3 (Remarks to the Author):

Authors are presenting a strong, very impressive study on the evolution of Hymenoptera. This study is outstanding by the quality and quality of the dataset (sampling of taxa and development of molecular dataset) but also by the comprehensive analyses of this dataset. The evolutionary questions are very interesting and they will interest a wide community of biologists. I have a very few comments to improve the manuscript.

Abstract: The description of the goal is a bit confusing as you insist on the impact of innovation. So we are expecting line 32 to have information on the impact of parasitoid regime on the diversification. SO, I would write line 27 "... to describe the evolution of the Hymenoptera, the timing of apparition of their key innovation: ...; and their impact on diversification.

We have re-phrased relevant sentences in the abstract (lines 27-28, 31-32).

Introduction: I would add at least a sentence on the abundance of Hymenoptera. Their

diversification is an important point, but the function they have in the ecosystem is also linked to their high abundance (bees for pollination, ants for biomass in tropics, ...).

We have included a reference to abundance of Hymenoptera (line 102).

Line 58: the term generalist is a bit ambiguous here while you describe after the wasp as predators. So, in this 30%, I would say, phytophages like bees, predators like wasp and generalists like ants.

We had re-phrased this statement already based on comments from other reviewers (line 124).

Line 62: the characters are here as key innovation. So, I would not present this as an hypothesis and not as a common fact.

Changed to read “putative key innovations.”

Line 65-66: bees and ants are ecosystem engineer because they are diverse and because they are abundance (partly due to their social behavior).

We are not sure how this comment relates to our description of the stinger as a potential key innovation in Aculeata. No changes were made.

Results: very well summarized, and it was not an easy task with such a huge dataset set and important number of analyses.

Thank you.

Discussion: Line 286, I would not cite the study Genise as they don't describe body fossils but trace fossil. Moreover, this study postulates the sweat bee as 100MA old. The present study itself does not support this age as all bee clades are younger than 75MA (including the Halictidae, sweat bees). Following references are more accurate:

Cardinal, S., & Danforth, B. N. (2013). Bees diversified in the age of eudicots. *Proceedings of the Royal Society B: Biological Sciences*. <https://doi.org/10.1098/rspb.2012.2686>

Michez D., M. Vanderplanck & M. S. Engel 2011. Chapter 5. Fossil bees and their plant associates. Pp 103-164, in *Evolution of Plant-Pollinator Relationships* (Ed. Patiny S.). Cambridge University press, ISBN-13: 9780521198929.

Thank you for the suggestion, we have replaced Genise et al. (2020) with Cardinal & Danforth (2013) and Michez et al. (2011).

Line 326: like to say in the introduction, the sting could be one of the trait supporting the apparition of sociality, which is a key evolution, more in term of abundance than species diversity probably.

We agree that the evolution of sociality probably also was a major innovation in aculeates, but our results do not allow us to infer a link with the evolution of the stinger. This would require a finer scale analysis within Aculeata with increased taxon sampling. We have therefore refrained from alluding to a possible connection of the two traits.

Line 375: generalism is quite rare in Hymenoptera or in Insect. So generalism (wide ecological niche), more than polylectism (many host plants) could be the key here. Moreover it's a generalism on an expending clade (Angiosperm). The secondary phytophagous shifted on an diversifying clade, not on Gymnosperm or Pteridophyta. There are key element showing co-diversification in these clades.

Masaridae are also secondary phytophagous and do nest, and they did not diversify well. Same for Melittidae (Michez et al. 2009: Phylogeny of the bee family Melittidae (Hymenoptera: Anthophila) based on combined molecular and morphological data. *Systematic Entomology*, 34: 574-597).

So, I'm not fully convinced that the combo "phytophagy" + "nest" is the key to explain diversification .

We have added a clarification that the switch of non-melittid bees to polylecty essentially resulted in a broader ecological niche (line 774). We had already discussed the potential co-diversification of the secondary phytophagous clade with angiosperms (lines 593-595) and refrain from reiterating this point.

Our hypothesis for the diversification in the secondary phytophagous clades was that the ancestral parasitoid "provisioning" behavior + the switch to "secondary phytophagy" influenced diversification (not "nesting" + "phytophagy"). But the reviewer is correct to point out that this hypothesis does not necessarily apply to Masarinae (subfamily with Vespidae) and melittid bees. We have added a comment pointing out that our hypothesis does not hold up for these two clades (lines 835-838).

Methodology: I don't understand why authors selected only 12 fossils and why they did not select fossils from more derived groups with well characterized taxonomic association (e.g. bees).

We limited our analysis to 12 well-supported fossils as these covered all major lineages for which confident fossil information was available. We selected the 12 fossils to calibrate deep and medium-level divergences within Hymenoptera, as there is evidence that this increases the accuracy of divergence estimation (see Mello & Schrago 2014). Adding fossil information for shallower nodes or more derived groups may increase the chance for disagreements between the assigned ages of calibrations and unwanted artifacts in the analyses. We have clarified this rationale in the main text (line 1009-1013) and supplementary methods (p.6).

Overall, congratulations again for this amazing, inspiring article. I fully support its publication. Thank you!

Reviewers' Comments:

Reviewer #2:

Remarks to the Author:

I am generally very pleased by the effort the authors have made to address the concerns of myself and the other reviewers, and I believe those efforts have significantly improved the paper. I have a main last point to make (unfortunately a rather wordy one, my apologies), with regards to the correct interpretation and nomenclature of the parameters of the HiSSE models, and also on how I feel they could/should be leveraged much more in order to explain the data and more strongly support the points the authors are already making in the discussion. I do also have very few specific corrections that caught my eye, and which I have listed at the end of my review (line numbers refer to the Word document without tracked changes). But pending these points being addressed, I would regard this manuscript to be ready for publication and I congratulate the authors for this exciting piece of research.

My main last comment on the current version of this manuscript is based in now having the rate estimates of the HiSSE runs available. I am glad you agreed that it is useful to consult the inferred rates, and that it helped cross-check the results for errors. I do think it greatly improved the manuscript. However, I think you should go a step further with it (this may be reiterating some of my comments in the first round regarding investigating some of these findings a bit further).

When looking at the states and their associated rates (in S11), some patterns come to light that should help explain the results some more. You have largely recognised and discussed these patterns, but I think you maybe misconstruing the meaning of the hidden trait (or maybe just have phrased it in a way that makes it sound like that). I believe that fixing some of these wordings will particularly also allow to compare/integrate the HiSSE results more with the BAMM/MEDUSA results.

E.g. for wasp waist, you rightly point out that wasp waist is associated to higher net diversification rates in both states of the hidden trait (i.e. $0A < 1A$, $0B < 1B$). However, I'm not sure why you word it as '[...] an increase in the observed [...] and the associated hidden trait' - all four states combine a state of the observed and the hidden trait each, and the hidden trait states don't necessarily imply presence vs absence, but just two different categories (whether they imply presence/absence, or something else is open to interpretation case by case), so you probably shouldn't refer to them as such. Really, the point of the hidden states is that there's another thing involved, or that there's something special/different about SOME of the clades with that observed trait. Thus it also sounds a bit odd when you say that a trait affected diversification 'via' a hidden state (lines 265/366/407) as opposed to maybe 'in combination with' (if particular combinations have the highest rates), or something the like.

So looking at the rates for wasp waist, they imply that presence of wasp waist has consistently higher rates than absence of wasp waist, but that within the taxa with wasp waist there are (at least) two different sub-categories, one of which has a net div rate that is yet an order of magnitude higher than the other.[*] When comparing fig 4A to fig 3, where the clades with increased rates are shown, it would feel quite tempting to suggest that those clades (all of which have a wasp waist) are responsible for the highest rates (i.e. are state 1B), whereas the remaining lineages with wasp waist may correspond to the intermediate rate (i.e. are state 1A).

Similarly, when looking at parasitoidism, it looks even more interesting, as the highest net div rates are 0A and 1B (and you do report how the trend is opposite for parasitoids in hidden state A vs B - except you only refer to B as the hidden state, which is not technically correct and may be confusing, as discussed above). Comparing Figure 4C with figure 3 shows intriguingly how shifts 1 and 6 are associated to parasitoid lineages, whereas shifts 4 and 5 are not parasitoid (with 2 and 3 being mixed), which again would strongly imply that the respective clades are perhaps responsible for the high rates in 0A and 1B respectively.

A very similar pattern (albeit less clear) can be seen for secondary phytophagy, with 1B diversifying

fastest, and 0B coming in second. Intriguingly, the patterns are reversed here compared to parasitoidism, with shifts 4 and 5 not parasitoid but secondary phytophagous, shift 1 being the reverse, and shift 6 being both.

As for the stinger I believe we had already discussed the notion that the CID model is favoured over HiSSE because there are too many clades with rate shifts outside of Aculeata than HiSSE could accommodate with two hidden states only.

Naturally, one should be careful with such ad hoc interpretations, not to jump to convenient conclusions, but I would argue the possibility should at least be brought up. Indeed, you do essentially discuss the results to that effect, but by focusing on shifts coinciding with the origin of those traits, and not incorporating the inferred shifts from BAMM & MEDUSA in order to help explain the meaning of the hidden states, I feel you are not using the very intriguing results that you got to their full extent. But in fact, as HiSSE also allows to infer tip rates (and thus states), the extent to which this interpretation is true could potentially be tested too. It would essentially allow you to show which tips on the tree HiSSE infers as A or B (in addition to the known states of 0 and 1), and these results could be compared to the BAMM/MEDUSA shifts. I would personally consider this low hanging fruit in comparison to the gained depth of insights into the results, although your opinion on this may differ.

Along the same lines, when I read the discussion, it sometimes reads as if you were interpreting the results of the analysis of one trait in isolation of the remaining results. I find it quite striking how e.g. the results of parasitoidism and secondary phytophagy might actually describe the same diversification patterns, but from different perspectives - assuming the respective other trait is partially explained by the respective hidden traits. While all these could be confirmed by tip rates/states, I would be quite intrigued to see what a MuSSE/MuHiSSE combining Parasitoidism and secondary phytophagy would yield. I have read your reasons against using MuSSE, and so you may of course reasonably elect not to analyse this still at this stage, but I feel the aforementioned notions of how the HiSSE results could be interpreted in the light of the BAMM/MEDUSA results should at least in some form be incorporated in the discussion (unless of course you strongly disagree with my interpretations here - but since you already discuss the fact that additional factors or modifications within those candidate traits may be responsible for diversification in this group, I would assume including the perspective of connecting hidden states and diversification rate shifts would only serve to strengthen your arguments).

As an addition to the results on wasp waist: On one hand, these results would imply to me that wasp waist could be an important pre-adaptation for a lot of those radiations, as you discussed. On the other hand, one could also argue that any derived trait that encompasses enough of the clades with rate shifts could be inferred as best fitting a HiSSE model, so a trait like wasp-waist which is present in all but the Symphyta grade may just as well not have any actual bearing on diversification. I personally find your arguments as to why it may represent an important preadaptation very convincing, and also the notion of a poorly-detected slowdown in the non-apocritan clades, but would argue it would be better to at least mention that from a strictly methodological perspective, we can't confidently say so based on these results alone, as any trait that is shared by most of these taxa and originates early on would give the same result. This caveat (together with the known problems of these models, which you already address), relativises your claim at line 341 about how a strong correlation needs to highlight the relevance of that trait.

Additionally:

- Your statement that the full irreversible HiSSE model was only marginally better supported than the CID model only seems to be the case for the A0 tree, and not for the C1 tree which you focus on in the figures (unless I'm misreading table S10).
- Your description in S11 about the state-coding of the CID models is not correct. E.g. 0A vs 1A does

not imply state A being absent vs present, but instead, just like for the full hisse model, implies the lineage is in states 0 and A vs 1 and A. The hidden states are A-D, and one of those four is always present in any lineage. 0 and 1 still refers to the observed trait state, so it is equally included in CID-4 as in HiSSE, ensuring the same number of parameters in both models. But in CID-4, while a lineage can still transition from 0A to 1A, both of those states are constrained to have the same rate (which is apparent in the respective tables in S11, so they could even be reduced in size).

- Regarding the bias of clade level trait assignments (second part of point 5 in the initial review), I do understand the process much better now, thank you for the clarifications and references to relevant supplementary materials! However, regarding your claim that only MuSSE analyses would have biased the finding of shifts at the base of those clades, I do not entirely agree. Since HiSSE is relying on trait states, this kind of assignment could potentially bias it the same way, and if I understood it correctly, you used clade specific sampling fractions for BMM, and if those were applied to the same clades (which I presume), they could bias BMM the same way. I agree that these procedures are probably the best approach you could take to ensure missing sampling is incorporated (and not doing so would bias the analyses in worse ways, arguably), but I think the potential of such a concerted bias should be acknowledged somewhere, even if just briefly.

Minor comments:

- there are a couple of oxford commas missing throughout, if you subscribe to them (I refrained from listing them all here, as e.g. the grammar check of a text editor should highlight them for you)
- 67: Shouldn't it be "a myriad of host niches" or "myriads of host niches"?
- 486: may have accelerated
- 624: put computeBayesFactors in "", just like setBMMpriors before?
- 198ff I wonder whether just referring to it as MEDUSA instead of 'stepwise AIC' throughout might make clearer which analysis is referenced to (just like you call it BMM throughout)?

S10: delta AIC for last one missing

S11: row names on phytophagy A0 are shifted one up

* [I personally have a hard time wrapping my head around the rates in a matrix. But they can be visualized by drawing the same kind of bubble-and-arrows diagram as in Fig1 of the original HiSSE paper, or even more so Fig 1 in this tutorial (<https://revbayes.github.io/tutorials/sse/hisse.html>), and e.g. draw the size or thickness of the arrows in proportion to the rates. You could make such plots for the manuscript or not, but at least as quick doodles for myself they helped me recognise the patterns. The transition rates might be a bit more challenging to interpret.]

Reviewer #4:

Remarks to the Author:

This paper provides an exceptionally thorough and balanced analysis of Hymenoptera. It also tethers the analysis of modern material firmly with fossil calibration points chosen to represent a large part of the ingroup. The study arrives at conclusions that are of interest to anyone who works on Hymenoptera, or entomology in general. The analytical work appears robust, and the authors include global experts on many of the taxa involved, providing strong support for their statements about taxonomic choices and the ecology of the groups involved.

I was asked to concentrate on the comments of Reviewer 3 and how they were addressed by the authors. I can see where some of the criticisms or suggestions made by Reviewer 3 came from, but I can also appreciate how they have been largely addressed in the revised draft of the manuscript, and

in the rebuttal letter. I would consider most of these issues essentially dealt with in the revised work.

However, Reviewer 3's comment about adding additional fossil calibration points should probably be given a bit more thought, and likely warrants a small addition to the discussion or conclusions.

I understand the authors' refusal to add unnecessary fossil calibration points after the Cretaceous, particularly if they are going to introduce uncertainty in the results without any benefit (the divergence points for most clades other than bees predate this time interval). That said, recent works like the revision of Rosa and Melo (2021: Apoid wasps (Hymenoptera: Apoidea) from mid-Cretaceous amber of northern Myanmar, *Cretaceous Research*), have described Cretaceous Crabronidae and Pemphredonidae at 99 Ma, which would push nodes like #42 back in time by another ~20 Ma. Although it is not necessary here, adding research developments like this one to the current analyses would influence where some of the divergence points occur, altering diversity estimates across time. The authors have done a good job of explaining how their analyses differ from previous works in terms of taxon choices and fossil calibration points, and the calibration points that they have chosen are sufficient. They just need to include some statement about how new fossil discoveries could alter the patterns observed.

The current limitations of the fossil record may have some influence on the overall interpretation of the findings and this needs to be captured in the discussion somewhere. Only three of the twelve fossil calibration points predate the Cretaceous, and these compression fossils can be more difficult to interpret than amber specimens (the taxonomic justifications based on venation given in the revised SD13 table are sufficient). In general, the timing of key innovations seems to match well when fossil sampling is strong (Late Cretaceous onward), but there are problems explaining innovations the further back in time you go, and the more reliant on compression fossils you become. Reduced fossil availability may partly explain the apparent mismatch between parasitoidism, the wasp waist, and stinger, with diversification events. It is worth mentioning that finding older fossils of Hymenoptera, or pre-Cretaceous exemplars of some clades may alter the timing between the origin of innovations and diversification. In some cases, this would extend the gap inferred in this work, but in many cases, it would drive the nodes deeper in time and push the diversification closer to the innovation. Dealing with this could be as simple as adding to the end of the sentence on line 516: "Future research should focus on dissecting these traits into their functional subcomponents to relate them with biological implications, and searching for fossils that could alter the timing of divergences." A more significant discussion would be appreciated though.

In terms of minor edits or suggested changes:

- Ln.486: "may accelerated" to "may have accelerated"
- fig.2: stars for 72 and 73 are positioned in such a way that it looks like the fossils are much younger than they really are (both need to be moved to the left significantly)
- SD 13, Row 1: "Hymneoptera" to "Hymenoptera"

Congratulations on an excellent paper, and I look forward to seeing it published.

Reviewer #5:

Remarks to the Author:

This manuscript expands on previous work on Hymenoptera phylogeny, which has also used phylogenomics data and performed divergence times. More uniquely, it uses this as a framework to understand key innovations in Hymenoptera and how they impact diversification. As one of the most diverse lineages on the planet (~8% of all spp.), it is a relevant one to understand what promotes diversity more broadly. The paper is thus impactful and it also involves a commendable amount of data collection and analysis.

I did not take issue with the methods used and agree with literature use and interpretation for the most part. Most of my recommendations are to improve how the study is framed so that the message is more clear.

Major recommendations:

1. Missing taxa: One of the points needing more clarity is with regard to how the study deals with the taxa they do not represent in diversification statistics and thus whether diversification metrics can be adequately assessed. I was only clear to me how non-included taxa were dealt with in the response to previous reviewers. It would be great if the methods and results elaborated on how the non-represented taxa are treated so that we know that the known species diversity of the respective clades are included in the analysis ("family-level diversity" gives the impression that only families are tested as single lineages and not the species diversity they contain). Also there are missing families. Please justify in the main text the degree to which this may impact the results given how they are treated in the analysis. Overall, I think the respective diversity is represented well here with the applied methods, but that is not clear as written.
2. Fossils: My first thought was that 12 fossils are not a great representation of the available fossils of Hymenoptera. It took reading details of the supplement to see that these were representatives that exemplified clade dates and were the best fossils to adequately age the clades. I think this is a good idea but the rationale needs to be described in the main text. Furthermore, it would be nice to see emphasis on the ways this compares to prior dating analysis in approach and accuracy.
3. Phylogenetic relationships: As stated, there are several Hymenoptera phylogenies. Refining this is not a focus of the paper, but it is also not that clear how the paper improves this or does things differently. A few sentences in the discussion also could highlight how it is different from and improves on prior work.
4. Figures: Figure 4 – I spent considerable time trying to see the outline vs. the internal branch patterns to take home the key message here. I recommend finding a better way to show these patterns. Also the small inset of the net diversification for each figure needs more description in the legend as I did not understand how to interpret this. Figure 1 – The choice of taxa to show vs. make a gray triangle is odd. Why the focus on pteromalids? I don't see any focus on this in the text to justify this.
5. Discussion: The discussion was quite long and it was hard to take away key points, so it could use some streamlining (see below). Paragraphs on innovations (starting in 366 and ending in 503) - Some of this reads like results (restating model results) and could stand to be more generalized. At this point the text gets quite detailed. Consider ways to simplify the message. Given that phytophagy is a more significant result it should be featured most.

Minor Points:

1. Figure 1. Some of the basal nodes for sawflies are in the Apocrita box.
2. Figure 3. Nice graphic. It could be improved by having a legend in the figure as to what the numbers are or putting the names of those taxa on the figure, otherwise you have to read the figure legend to see what the diverse lineages are.
3. 70% missing data is still a lot of missing data and missing data can have a big effect on results. Why not use 100%? How does your missing data representation compare to prior studies?
4. Lines 313 and 315 (in pdf without tracked changes). Please state the dates for these more specifically so we can put this in context. Same for Lines 189 – 192.
5. Line 324. Can you comment more about the reliability of your dates and whether other studies match your dates on this?
6. Lines 334 – 356. These paragraphs are oddly conceptual and detract from the key message of the paper, thus I recommend these points should be simplified or integrated into other parts of the paper or into the supplement. My feeling is that these are understood key features in Hymenoptera and that your metric is one metric that can test whether it is an innovation, so I don't think it needs this much rationale. Also, it seems to me you are testing whether diversification suggests this is an innovation, so looking at how the innovation impacts diversification seems cyclical (Line 334).

7. Lines 359 is too vague. I think these limitations are better addressed elsewhere.
8. Line 360 is overly subjective thus I recommend removing it.
9. Line 361 – 364. I recommend starting the next paragraph with this and not mention that you will write about these in following paragraphs.
10. I do not agree with “reasonably certain” in line 505. You saw diversification shifts in several of these but there were reasons to not be convinced, so I don’t see this as a reasonably certain conclusion. I would state that these traits showed less reliable suggestion as to their direct role of diversification and emphasize earlier on in this paragraph the more clear role of phytophagy.

NCOMMS-22-11260A - Response to reviewers

We thank all reviewers for taking the time to review our manuscript and provide valuable comments which significantly improved our manuscript.

In our point-by-point response below the original reviewer comments are copied in black font and our responses are inserted below in blue font. Line numbers refer to the word document of the manuscript with track changes highlighted (not the pdf). Changes in the supplementary text have been indicated with blue font.

In addition to the changes outlined below, we have made minor corrections to the text to improve clarity and conciseness.

REVIEWER COMMENTS

Reviewer #2 (Remarks to the Author):

I am generally very pleased by the effort the authors have made to address the concerns of myself and the other reviewers, and I believe those efforts have significantly improved the paper. I have a main last point to make (unfortunately a rather wordy one, my apologies), with regards to the correct interpretation and nomenclature of the parameters of the HiSSE models, and also on how I feel they could/should be leveraged much more in order to explain the data and more strongly support the points the authors are already making in the discussion. I do also have very few specific corrections that caught my eye, and which I have listed at the end of my review (line numbers refer to the Word document without tracked changes). But pending these points being addressed, I would regard this manuscript to be ready for publication and I congratulate the authors for this exciting piece of research.

My main last comment on the current version of this manuscript is based in now having the rate estimates of the HiSSE runs available. I am glad you agreed that it is useful to consult the inferred rates, and that it helped cross-check the results for errors. I do think it greatly improved the manuscript. However, I think you should go a step further with it (this may be reiterating some of my comments in the first round regarding investigating some of these findings a bit further).

When looking at the states and their associated rates (in S11), some patterns come to light that should help explain the results some more. You have largely recognised and discussed these patterns, but I think you maybe misconstruing the meaning of the hidden trait (or maybe just have phrased it in a way that makes it sound like that). I believe that fixing some of these wordings will particularly also allow to compare/integrate the HiSSE results more with the BAMM/MEDUSA results.

E.g. for wasp waist, you rightly point out that wasp waist is associated to higher net diversification rates in both states of the hidden trait (i.e. $0A < 1A$, $0B < 1B$). However, I'm not sure why you word it as '[...] an increase in the observed [...] and the associated hidden trait' - all four states combine a state of the observed and the hidden trait each, and the hidden trait states don't necessarily imply presence vs absence, but just two different categories (whether they imply presence/absence, or something else is open to interpretation case by case), so you probably shouldn't refer to them as such. Really, the point of the hidden states is that there's another thing involved, or that there's something special/different about SOME of the clades with that observed trait. Thus it also sounds a bit odd when you say that a trait affected diversification 'via' a hidden state (lines 265/366/407) as opposed to maybe 'in combination with' (if particular combinations have the highest rates), or something the like.

>We appreciate the clarification and have amended the wording throughout the text appropriately (e.g., lines 274, 290, 300, 507, etc.).

So looking at the rates for wasp waist, they imply that presence of wasp waist has consistently higher rates than absence of wasp waist, but that within the taxa with wasp waist there are (at least) two different sub-categories, one of which has a net div rate that is yet an order of magnitude higher than the other.[*] When comparing fig 4A to fig 3, where the clades with increased rates are shown, it would feel quite tempting to suggest that those clades (all of which have a wasp waist) are responsible for the highest rates (i.e. are state 1B), whereas the remaining lineages with wasp waist may correspond to the intermediate rate (i.e. are state 1A).

>We have extracted tip states from the HiSSE results and present these now for the wasp waist, parasitoidism and secondary phytophagy in a new Supplementary Data Table 12. The results are not quite as straightforward. None of the clades with higher net diversification rates estimated for BAMM has state 1B estimated for wasp waist as most probable.

However, the high-rate clades correspond overall with higher probabilities of having state 1B (up to 0.58), meaning for those clades, the probabilities tend to be split among 1A and 1B, but with a considerable proportion in state 1B. So, this seems to indicate that state 1B could be largely responsible for the highest rates. We have incorporated these results in line 280–284 and line 512–515.

Similarly, when looking at parasitoidism, it looks even more interesting, as the highest net div rates are 0A and 1B (and you do report how the trend is opposite for parasitoids in hidden state A vs B - except you only refer to B as the hidden state, which is not technically correct and may be confusing, as discussed above). Comparing Figure 4C with figure 3 shows intriguingly how shifts 1 and 6 are associated to parasitoid lineages, whereas shifts 4 and 5 are not parasitoid (with 2 and 3 being mixed), which again would strongly imply that the respective clades are perhaps responsible for the high rates in 0A and 1B respectively.

>Again, clades with higher net diversification rates for parasitoidism correspond with higher probabilities of having state 1B (up to 0.71), but most taxa in these clades do not have 1B estimated as most probable state. Clades corresponding with shifts 1 and 6 are among those with significant probabilities in state 1B. For shifts 4 and 5 (which are secondary phytophagous clades), the associated clades have highest probabilities for state 0B, but compared to other non-parasitoid clades, they have significantly higher proportional probabilities for state 0A as well (up to 0.38). We outline these results in lines 291–297 and line 632–635.

A very similar pattern (albeit less clear) can be seen for secondary phytophagy, with 1B diversifying fastest, and 0B coming in second. Intriguingly, the patterns are reversed here compared to parasitoidism, with shifts 4 and 5 not parasitoid but secondary phytophagous, shift 1 being the reverse, and shift 6 being both.

>For secondary phytophagy, the pattern is indeed reversed, with most secondarily phytophagous taxa estimated with state 1A as the most probable but having proportionally high probabilities for state 1B (up to 0.58; Supplementary Table 12). These patterns are largely congruent with the rate shifts 4,5, and iii (Table 4) recovered by BAMM and MEDUSA. Conversely, for non-secondarily phytophagous taxa state 0A is the most probable, but 0B has the faster net diversification rate. We describe these results in lines 301–305.

As for the stinger I believe we had already discussed the notion that the CID model is favoured over HiSSE because there are too many clades with rate shifts outside of Aculeata than HiSSE could accommodate with two hidden states only.

>Correct, and we had added a statement to this regard in last version of the manuscript (lines 588–590).

Naturally, one should be careful with such ad hoc interpretations, not to jump to convenient conclusions, but I would argue the possibility should at least be brought up. Indeed, you do essentially discuss the results to that effect, but by focusing on shifts coinciding with the origin of those traits, and not incorporating the inferred shifts from BAMM & MEDUSA in order to help explain the meaning of the hidden states, I feel you are not using the very intriguing results that you got to their full extent. But in fact, as HiSSE also allows to infer tip

rates (and thus states), the extent to which this interpretation is true could potentially be tested too. It would essentially allow you to show which tips on the tree HiSSE infers as A or B (in addition to the known states of 0 and 1), and these results could be compared to the BAMM/MEDUSA shifts. I would personally consider this low hanging fruit in comparison to the gained depth of insights into the results, although your opinion on this may differ.

>We have presented the inferred tip states in Supplementary Data 12 and incorporated the results in the main text results and discussion, making connections between HiSSE, and BAMM and MEDUSA results wherever appropriate. Additionally, we have inferred tip rates and states across the phylogeny using the MiSSE model and present these results in a new Supplementary Data Table S13 and Supplementary Fig. S28. Results have also been incorporated into the main text (lines 265–267, 333–340, line 681–682, 998–1003), and we have added the additional code to an updated version of the Dryad repository. Although models with six hidden states are preferred, the tip rates and states point to one hidden state being responsible for the highest diversification rates. This state has highest probabilities in clades that already had rate shifts estimated by BAMM/MEDUSA, and is prevalent in the three secondarily phytophagous clades. We interpret this result as additional support that secondary phytophagy (or some trait connected with the former) plays an important role in hymenopteran diversification.

Along the same lines, when I read the discussion, it sometimes reads as if you were interpreting the results of the analysis of one trait in isolation of the remaining results. I find it quite striking how e.g. the results of parasitoidism and secondary phytophagy might actually describe the same diversification patterns, but from different perspectives - assuming the respective other trait is partially explained by the respective hidden traits. While all these could be confirmed by tip rates/states, I would be quite intrigued to see what a MuSSE/MuHiSSE combining Parasitoidism and secondary phytophagy would yield.

>We tried a MuSSE/MuHiSSE analyses for parasitoidism and secondary phytophagy, using the 8 character-independent (MuCID) models available (Nakov et al. 2019). However, for both topologies a MuCID model with either 7 or 8 hidden states was best-scoring, indicating there either is no joint association of the two traits with diversification or that this cannot be captured by the models. We have not included these results in the manuscript, as we think this would increase the complexity of the results and discussion even more (the manuscript already being on the long side) without much benefit.

I have read your reasons against using MuSSE, and so you may of course reasonably elect not to analyse this still at this stage, but I feel the aforementioned notions of how the HiSSE results could be interpreted in the light of the BAMM/MEDUSA results should at least in some form be incorporated in the discussion (unless of course you strongly disagree with my interpretations here - but since you already discuss the fact that additional factors or modifications within those candidate traits may be responsible for diversification in this group, I would assume including the perspective of connecting hidden states and diversification rate shifts would only serve to strengthen your arguments).

>As outlined above, we have incorporated hidden state rates and state estimates into the results/discussion, connecting them as best as possible with the BAMM/MEDUSA results. We have tried to keep these additions to the manuscript brief, however, given the overall constraints in terms of manuscript length.

As an addition to the results on wasp waist: On one hand, these results would imply to me that wasp waist could be an important pre-adaptation for a lot of those radiations, as you discussed. On the other hand, one could also argue that any derived trait that encompasses enough of the clades with rate shifts could be inferred as best fitting a HiSSE model, so a trait like wasp-waist which is present in all but the Symphyta grade may just as well not have any actual bearing on diversification. I personally find your arguments as to why it may represent an important preadaptation very convincing, and also the notion of a poorly-detected slowdown in the non-apocritan clades, but would argue it would be better to at least mention that from a strictly methodological perspective, we can't confidently say so based on these results alone, as any trait that is shared by most of these taxa and originates early on

would give the same result. This caveat (together with the known problems of these models, which you already address), relativises your claim at line 341 about how a strong correlation needs to highlight the relevance of that trait.

>We incorporated mention of this additional caveat briefly in line 735. We opted not to expand this paragraph in the discussion much more, based on R5's comment.

Additionally:

- Your statement that the full irreversible HiSSE model was only marginally better supported than the CID model only seems to be the case for the A0 tree, and not for the C1 tree which you focus on in the figures (unless I'm misreading table S10).

>Thank you for pointing this out. We have removed this statement.

- Your description in S11 about the state-coding of the CID models is not correct. E.g. 0A vs 1A does not imply state A being absent vs present, but instead, just like for the full hisse model, implies the lineage is in states 0 and A vs 1 and A. The hidden states are A-D, and one of those four is always present in any lineage. 0 and 1 still refers to the observed trait state, so it is equally included in CID-4 as in HiSSE, ensuring the same number of parameters in both models. But in CID-4, while a lineage can still transition from 0A to 1A, both of those states are constrained to have the same rate (which is apparent in the respective tables in S11, so they could even be reduced in size).

>We modified the caption in Supplementary Table S11 to read "*For CID-models, 0A-D denotes the combination of the absent observed trait state each with four different hidden states, whereas 1A-D denotes the combination of the present observed trait state with the four hidden states.*"

- Regarding the bias of clade level trait assignments (second part of point 5 in the initial review), I do understand the process much better now, thank you for the clarifications and references to relevant supplementary materials! However, regarding your claim that only MuSSE analyses would have biased the finding of shifts at the base of those clades, I do not entirely agree. Since HiSSE is relying on trait states, this kind of assignment could potentially bias it the same way, and if I understood it correctly, you used clade specific sampling fractions for BAMM, and if those were applied to the same clades (which I presume), they could bias BAMM the same way. I agree that these procedures are probably the best approach you could take to ensure missing sampling is incorporated (and not doing so would bias the analyses in worse ways, arguably), but I think the potential of such a concerted bias should be acknowledged somewhere, even if just briefly.

>We had already inserted a paragraph in the supplemental discussion acknowledging this bias, and have modified and expanded this further to address this concern (p.50): "*It is further possible that our clade-level approach to character coding could have resulted in a similar bias of shifts occurring prior to or at the base of clades for BAMM, HiSSE and MEDUSA analyses. This is more likely for the Medusa estimates as these were performed on a clade-level tree only and necessarily limited to shifts prior to the base of these clades. HiSSE recovered a rate increase in Eurytomidae, a clade that was sunk for the purpose of the analyses within the larger Chalcidoidea clade. This result that was also confirmed by BAMM analyses performed on a species-level tree while excluding all trait information. BAMM analyses also recovered a shift within Ichneumonidae, giving further notion to believe that the placement of shifts in these analyses was not influenced by our clade-level character coding. Nonetheless, the possibility of such a bias exists.*" We have not mentioned this issue in the main text as we could not think of a way to do this space-efficiently.

Minor comments:

- there are a couple of oxford commas missing throughout, if you subscribe to them (I refrained from listing them all here, as e.g. the grammar check of a text editor should highlight them for you)

>We thoroughly checked the text and added the missing commas.

- 67: Shouldn't it be "a myriad of host niches" or "myriads of host niches"?

>Changed to “a myriad of host niches”.

- 486: may have accelerated

>Corrected.

- 624: put computeBayesFactors in "", just like setBAMMpriors before?

>Corrected.

- 198ff I wonder whether just referring to it as MEDUSA instead of 'stepwise AIC' throughout might make clearer which analysis is referenced to (just like you call it BAMM throughout)?

>Agreed. We changed all mentions of “stepwise AIC” to read “MEDUSA.”

S10: delta AIC for last one missing

>Thank you, added.

S11: row names on phytophagy A0 are shifted one up

>Corrected.

* [I personally have a hard time wrapping my head around the rates in a matrix. But they can be visualized by drawing the same kind of bubble-and-arrows diagram as in Fig1 of the original HiSSE paper, or even more so Fig 1 in this tutorial

(<https://revbayes.github.io/tutorials/sse/hisse.html>), and e.g. draw the size or thickness of the arrows in proportion to the rates. You could make such plots for the manuscript or not, but at least as quick doodles for myself they helped me recognise the patterns. The transition rates might be a bit more challenging to interpret.]

>Thanks for the helpful suggestion. We created such plots for wasp waist, parasitoidism and secondary phytophagy and inserted them in Supplementary Table S11 to visualize the differences in rates.

Reviewer #4 (Remarks to the Author):

This paper provides an exceptionally thorough and balanced analysis of Hymenoptera. It also tethers the analysis of modern material firmly with fossil calibration points chosen to represent a large part of the ingroup. The study arrives at conclusions that are of interest to anyone who works on Hymenoptera, or entomology in general. The analytical work appears robust, and the authors include global experts on many of the taxa involved, providing strong support for their statements about taxonomic choices and the ecology of the groups involved.

I was asked to concentrate on the comments of Reviewer 3 and how they were addressed by the authors. I can see where some of the criticisms or suggestions made by Reviewer 3 came from, but I can also appreciate how they have been largely addressed in the revised draft of the manuscript, and in the rebuttal letter. I would consider most of these issues essentially dealt with in the revised work.

However, Reviewer 3's comment about adding additional fossil calibration points should probably be given a bit more thought, and likely warrants a small addition to the discussion or conclusions.

I understand the authors' refusal to add unnecessary fossil calibration points after the Cretaceous, particularly if they are going to introduce uncertainty in the results without any benefit (the divergence points for most clades other than bees predate this time interval). That said, recent works like the revision of Rosa and Melo (2021: Apoidea wasps (Hymenoptera: Apoidea) from mid-Cretaceous amber of northern Myanmar, Cretaceous Research), have described Cretaceous Crabronidae and Pemphredonidae at 99 Ma, which would push nodes like #42 back in time by another ~20 Ma. Although it is not necessary here, adding research developments like this one to the current analyses would influence where some of the divergence points occur, altering diversity estimates across time. The authors have done a good job of explaining how their analyses differ from previous works in

terms of taxon choices and fossil calibration points, and the calibration points that they have chosen are sufficient.

They just need to include some statement about how new fossil discoveries could alter the patterns observed.

>We agree that there is always a chance that new fossil discoveries will substantially change our understanding of the evolution of a group (the Rosa & Melo (2021) paper was published after our dating analyses were completed). We have inserted a statement in the discussion (lines 362–363).

The current limitations of the fossil record may have some influence on the overall interpretation of the findings and this needs to be captured in the discussion somewhere. Only three of the twelve fossil calibration points predate the Cretaceous, and these compression fossils can be more difficult to interpret than amber specimens (the taxonomic justifications based on venation given in the revised SD13 table are sufficient). In general, the timing of key innovations seems to match well when fossil sampling is strong (Late Cretaceous onward), but there are problems explaining innovations the further back in time you go, and the more reliant on compression fossils you become. Reduced fossil availability may partly explain the apparent mismatch between parasitoidism, the wasp waist, and stinger, with diversification events. It is worth mentioning that finding older fossils of Hymenoptera, or pre-Cretaceous exemplars of some clades may alter the timing between the origin of innovations and diversification. In some cases, this would extend the gap inferred in this work, but in many cases, it would drive the nodes deeper in time and push the diversification closer to the innovation. Dealing with this could be as simple as adding to the end of the sentence on line 516: "Future research should focus on dissecting these traits into their functional subcomponents to relate them with biological implications, and searching for fossils that could alter the timing of divergences." A more significant discussion would be appreciated though.

>We agree with this point and have added a statement to that effect to the discussion (lines 788–790). Given the length of the manuscript, we have refrained from adding a more substantial discussion.

In terms of minor edits or suggested changes:

- In.486: "may accelerated" to "may have accelerated"

>Corrected.

- fig.2: stars for 72 and 73 are positioned in such a way that it looks like the fossils are much younger than they really are (both need to be moved to the left significantly)

>We moved both stars somewhat to the left, but since #72 represents a stem-group fossil of a subfamily within Figitidae s.l., it would not be correct to move this all the way to the crown-group node of that clade.

- SD 13, Row 1: "Hymneoptera" to "Hymenoptera"

>Corrected.

Congratulations on an excellent paper, and I look forward to seeing it published.

>Thank you!

Reviewer #5 (Remarks to the Author):

This manuscript expands on previous work on Hymenoptera phylogeny, which has also used phylogenomics data and performed divergence times. More uniquely, it uses this as a framework to understand key innovations in Hymenoptera and how they impact diversification. As one of the most diverse lineages on the planet (~8% of all spp.), it is a relevant one to understand what promotes diversity more broadly. The paper is thus impactful and it also involves a commendable amount of data collection and analysis.

I did not take issue with the methods used and agree with literature use and interpretation for the most part. Most of my recommendations are to improve how the study is framed so that

the message is more clear.

Major recommendations:

1. Missing taxa: One of the points needing more clarity is with regard to how the study deals with the taxa they do not represent in diversification statistics and thus whether diversification metrics can be adequately assessed. I was only clear to me how non-included taxa were dealt with in the response to previous reviewers. It would be great if the methods and results elaborated on how the non-represented taxa are treated so that we know that the known species diversity of the respective clades are included in the analysis (“family-level diversity” gives the impression that only families are tested as single lineages and not the species diversity they contain). Also there are missing families. Please justify in the main text the degree to which this may impact the results given how they are treated in the analysis. Overall, I think the respective diversity is represented well here with the applied methods, but that is not clear as written.

>We have tried to clarify as much as possible in the methods of the main text how the information of missing taxa was incorporated for BAMM, MEDUSA and HiSSE analyses (see lines 899–912, 938–941). This necessitated moving a substantial amount of text from the supplementary material to the main text, but we hope this clarifies the procedures. We further inserted some text in the discussion about our thoughts to what degree missing taxa may have influenced the analyses (lines 744–752).

2. Fossils: My first thought was that 12 fossils are not a great representation of the available fossils of Hymenoptera. It took reading details of the supplement to see that these were representatives that exemplified clade dates and were the best fossils to adequately age the clades. I think this is a good idea but the rationale needs to be described in the main text. Furthermore, it would be nice to see emphasis on the ways this compares to prior dating analysis in approach and accuracy.

>We already had a short statement on the rationale of the calibration scheme in the main text, which we have now expanded (lines 857–862). Concerning placing an emphasis on the comparison to prior dating analyses, we had already included a section into the supplementary discussion on this topic (Comparison of divergence results, p.46–47) comparing the methodology used in the three main recent analyses, an abbreviated version of which is included in the main text (lines 358–369). We expanded the supplementary text slightly to include a statement comparing the number of fossil calibrations, which were not yet explicitly described. The comparison of divergence dating with other studies was not one of our objectives (one could focus an entire separate article on this topic) and we simply do not have the space in the main text to go more in depth on this topic. Moreover, as already stated in the supplemental discussion, it is impossible to compare the accuracy between these analyses as many variables other than the calibrations were different between them (e.g., taxon sampling, type of sequence data, type of divergence dating algorithm).

3. Phylogenetic relationships: As stated, there are several Hymenoptera phylogenies. Refining this is not a focus of the paper, but it is also not that clear how the paper improves this or does things differently. A few sentences in the discussion also could highlight how it is different from and improves on prior work.

>We appreciate this well-intended comment, but have to emphasize again that the discussion of phylogenetic relationships was not an objective for the present paper. We include a paragraph in the results describing the main phylogenetic results of our study (lines 140–161), which is expanded in the supplementary results (starting p.11). This is complemented by an extensive discussion + figure in the supplementary material (p. 42–47) that compares our study to previous analyses on Hymenoptera, to which we refer the reader in the main text. The format of the journal does not allow us to include this discussion in the main text. We feel that adding just a few sentences on phylogenetic relationships to the discussion will not be sufficient to summarize the wealth of information and will appear out of

context. We therefore politely decline to follow this recommendation, referring to the information that is already provided in the supplementary material.

4. Figures: Figure 4 – I spent considerable time trying to see the outline vs. the internal branch patterns to take home the key message here. I recommend finding a better way to show these patterns. Also the small inset of the net diversification for each figure needs more description in the legend as I did not understand how to interpret this.

>The HiSSE plots shown in Fig. 4 are the standard way to show the results of these kind of analyses. We realize that with such a large phylogeny, the plots get crowded and harder to read. But we think it will be easier to read the figure in the final high-resolution version rather than the manuscript pdf, as one can zoom in further into the figure. The histograms in the lower right of each panel of Fig. 4 represent the distribution of net diversification rates associated with the observed states. We have clarified this in the caption.

Figure 1 – The choice of taxa to show vs. make a gray triangle is odd. Why the focus on pteromalids? I don't see any focus on this in the text to justify this.

>We initially wanted to display the wildly polyphyletic nature of pteromalids in this figure and thus did not collapse many branches for that family. But we see the point; as the manuscript is currently presented, this level of detail is not appropriate. We have therefore modified the figure, collapsing all monophyletic clades within Pteromalidae.

5. Discussion: The discussion was quite long and it was hard to take away key points, so it could use some streamlining (see below). Paragraphs on innovations (starting in 366 and ending in 503) - Some of this reads like results (restating model results) and could stand to be more generalized. At this point the text gets quite detailed. Consider ways to simplify the message. Given that phytophagy is a more significant result it should be featured most.

>We have followed most of the specific recommendations of the reviewer as outlined below. We have also tried to simplify the discussion on innovations and cut redundancies with the results section. However, the underlying diversification analyses are not trivial and we feel that a brief summary of joint support from the three different analyses is warranted for each innovation before launching into an interpretation. Since secondary phytophagy is already featured to a much greater extent in the discussion than other innovations (2 pages vs 2,5 pages for the remaining three characters), we refrained from trying to highlight this innovation even more in the discussion.

Minor Points:

1. Figure 1. Some of the basal nodes for sawflies are in the Apocrita box.

>Corrected in new version of the figure.

2. Figure 3. Nice graphic. It could be improved by having a legend in the figure as to what the numbers are or putting the names of those taxa on the figure, otherwise you have to read the figure legend to see what the diverse lineages are.

>Thank you for the suggestion. We experimented with providing the names of the diverse lineages in the figure, but concluded that the figure then looks much more cluttered. We think it should not be too much effort for the reader to refer to the caption for this information.

3. 70% missing data is still a lot of missing data and missing data can have a big effect on results. Why not use 100%? How does your missing data representation compare to prior studies?

>The matrix does not have 70% missing data (we agree, that would be a lot), but was filtered for 70% taxon completeness in the included loci. This means that only loci that were captured in at least 70% of the taxa were included in the final matrix. Consequently, a 70% matrix is more complete than, e.g., a 50% matrix. This level of completeness filtering and missing data is similar to other phylogenomic studies. The filtering procedures are described in more detail in the supplementary methods, but we tried to clarify this in the main text as well (see lines 823–824).

4. Lines 313 and 315 (in pdf without tracked changes). Please state the dates for these more specifically so we can put this in context. Same for Lines 189 – 192.

>We have added references to the dates (lines 192, 196, 379, 390).

5. Line 324. Can you comment more about the reliability of your dates and whether other studies match your dates on this?

>The second paragraph of the discussion (lines 355–369) compares our age estimate for the evolution of parasitoidism (i.e. the age of *Vespina*) with the three main other contemporary studies and comments on the methodological aspects of these and our studies and their impact on the reliability of divergence estimates. The supplementary discussion expands on these issues (p.46–47). We therefore choose not to reiterate this information further below in the discussion.

6. Lines 334 – 356. These paragraphs are oddly conceptual and detract from the key message of the paper, thus I recommend these points should be simplified or integrated into other parts of the paper or into the supplement. My feeling is that these are understood key features in Hymenoptera and that your metric is one metric that can test whether it is an innovation, so I don't think it needs this much rationale. Also, it seems to me you are testing whether diversification suggests this is an innovation, so looking at how the innovation impacts diversification seems cyclical (Line 334).

>We have paired this paragraph down and simplified as much as possible, but Reviewer 2 had strong opinions about addressing some of these points in the discussion of the main text. Therefore, we cannot move this discussion in the supplements. We moved the entire paragraph further down towards the end of the discussion, which hopefully works better for the flow of the discussion.

7. Lines 359 is too vague. I think these limitations are better addressed elsewhere.

>We removed this specific statement as recommended, but based on major comment #1 the taxon sampling limitation was addressed in this context in more detail.

8. Line 360 is overly subjective thus I recommend removing it.

>Removed as recommended.

9. Line 361 – 364. I recommend starting the next paragraph with this and not mention that you will write about these in following paragraphs.

>Modified (now lines 409–410).

10. I do not agree with “reasonably certain” in line 505. You saw diversification shifts in several of these but there were reasons to not be convinced, so I don't see this as a reasonably certain conclusion. I would state that these traits showed less reliable suggestion as to their direct role of diversification and emphasize earlier on in this paragraph the more clear role of phytophagy.

>Agreed. We have modified these statements accordingly (lines 754–755).

Reviewers' Comments:

Reviewer #2:

Remarks to the Author:

Once again I am happy about the implementation of my comments (and those of the other reviewers, as far as I can tell), and I commend the authors for this great study, which I now again support to be published. The improved explanations of methods and rationale make things very clear now, and the interpretations of the results paint a clearer picture. I have elaborated a few comments on the newly presented findings, some of which come with suggestions. I believe all my suggestions can be resolved by some minor edits, rendering this manuscript version practically final.

In brief the main suggested changes are:

1. clarify wording on the association of inferred hidden states and inferred rate shifts and avoid cherry-picking those results
2. briefly mention/clarify the result on wasp waist perhaps being solely driven by it encompassing all rate shifts
3. same for parasitoidism (plus a minor wording suggestion)
4. minor typo (along with some comments)
5. very minor formatting error in supp mat
6. very minor point on MiSSE results
7. same as point 2. actually, up to the authors which part of the manuscript the point would be addressed better
(also: a shortening suggestion of two partially redundant sentences)
8. adding clearer acknowledgement of the limits of what diversification rate shifts secondary phytophagy can explain
9. adding short caveat about hypothesis that parental care reduces extinction risk, given the HiSSE results

In detail:

1. It was very interesting to see the inferred HiSSE tip states, and I agree that the result isn't as straightforward as one could have hoped. It is actually a bit puzzling that the two hidden states for wasp waist overlap so poorly with the inferred rate shifts (because in theory the SSE states should match the inferred rate shifts to a good extent). It is also striking that for some models, some trait states aren't represented in the tips at all (e.g. 0A for wasp waist). However, looking at the transition rates (thanks for adding those figures in Suppdata 11, they make things a lot clearer), one can see that there are high transitions from 0A to 0B and inversely from 1B to 1A. This might suggest that the model inferred 0A and 1B to mainly be at internal nodes, perhaps explaining some kind of rate dynamics over time there. Plotting all the reconstructed nodes could show that, though there's probably a good amount of uncertainty for deeper nodes, and that would more serve to interpret what the model did than reveal exciting biology, so doing this is probably not necessary here.

However, I would urge you to word the section where you point out the overlap of shift clades and taxa with higher likelihood of state 1B a bit more carefully (lines 273-277 in the pdf/word doc with tracked changes). I initially read it and looked at the inferred state likelihoods and thought since merely 12 species with wasp waist have state 1B inferred as more than 50% likely, I probably wouldn't want to argue that their overlap with the clades who had rate shifts inferred would explain much. Only after inspecting the likelihood for Parasitoidism I realised that you meant taxa where the likelihood for 1B is on the higher side within 1B, albeit 1A may still be inferred as more likely in many of them.

Perhaps this could be clarified by wording it similarly to this (merely a suggestion, as I found this the

easiest way of communicating my intentions here):

"Most tip states are estimated with higher probabilities in state 1A (Supplementary Data 12), and only very few are inferred as having 1B as the most likely state. However, state 1B has a higher net diversification rate than state 1A and taxa whose marginal probabilities for state 1B are on the higher side (up to 0.58) overlap to some extent with clades that showed elevated rates in BMM and MEDUSA analyses (Supplementary Data 11)."

Seeing it like that, I indeed find that intriguing, but since also a bunch of other taxa have slightly higher likelihoods of 1B, you should be careful not to cherry-pick these cases, so perhaps the presence of 1B taxa outside the radiations should be noted briefly.

2. It is actually also intriguing that the main state inferred for the Symphyta grade (0B) has relatively low net diversification rates but is actually inferred to have high turnover (i.e. both high speciation and extinction). Possible that this is related to the inferred slowdown in BMM, though whether you want to elaborate that I shall leave up to you. In any case, the idea that wasp waist might have laid the background to enable the other key innovations which eventually triggered the radiations (e.g. sensu Donoghue or Bouchenak-Khelladi et al.) seems reasonable, as long as you emphasize that there is likely no direct effect (which you do). But it is also likely that the rate increase associated with the presence of a wasp waist is simply driven by the fact that the lineages with that trait contain all the inferred shift clades, and its absence particularly includes the low diversity Symphyta grade, so maybe this should also be mentioned briefly.

3. The same (as in point 1. and 2.) would seem to be the case for Parasitoidism. After all, it is essentially the same as wasp waist, minus BMM shifts 4 and 5 (and a few smaller clades), which explains why there's not a clear association of its presence with higher rates (to put it oversimply, the low rates of the Symphyta grade and the shifts 4 and 5 cancel each other out). Though given the conflicting trend depending on the hidden state, I'd rather say the association of this trait with diversification was 'suggested' rather than 'supported' on line 281. Also here I agree that the overlap of slightly elevated likelihoods for state 1B and 0A with some of the shifts is intriguing, but the same caveat for cherry picking and overinterpretation applies.

4. The results for Secondary phytophagy are probably the most intriguing, especially since the increased likelihoods for the respective hidden states seem to match the BMM shifts even better, especially the cumulative shift probability (i.e. which includes more bees in shift 4, and which differentiates two separate sub-shifts in the Ichneumonidae, which seem to be found by HiSSE too, maybe?). Though also here it is a pity that those hidden states aren't recovered more clearly and that other taxa outside of those shifts also show higher likelihoods for the respective hidden states. Yet I still think your basis to argue that Secondary Phytophagy may have had an impact on diversification rates seems fairly strong.

I think the parenthesis on line 299 should close after "MEDUSA" and not after "Table 4"?

5. Personally, I'd now want to dig deeper into this to figure out whether these results make sense or are artefactual, i.e. whether perhaps another hidden state is needed, or whether there were convergence issues, etc., but it is clear that this would go beyond the scope of this article at this point.

BTW: some of the bold highlighting of the highest probabilities in SuppData 12 are wrong (e.g. some Eurytomidae have 1B as highest probability for phytophagy, but 1A is bold), though I would consider this a very minor issue.

6. The use of MiSSE as additional confirmation was certainly a good idea and I agree that these results back up the other findings (although the shifts in Cynipidae and the Bees seem to include a few more taxa, and there's also some higher rates in Proctotrupidae and Braconidae, but I agree the association to the shifts is meaningful). As a minor point, I think the results table suggests that for topC-1 there were only 5 hidden states, and only topA-0 had 6?

7. I think the diversification parts of the discussion are also much improved now. The lack of direct effect of wasp waist is made clear enough and the suggestion that hidden states play a major role is highlighted appropriately. Perhaps the aforementioned point that all the radiated clades being within the clade with wasp waist would simply raise its overall diversification rate could be made explicit here, although line 432 already points in that direction.

I think the first two sentences on the wasp waist discussion (lines 420-427) largely seem to say the same thing (with the second one being more explicit), so I think those could be merged into one to shorten the text.

8. The remainder of the discussion combines the results intriguingly with other knowledge on the group and is very exciting to me. However, regarding secondary phytophagy, I am still missing an acknowledgement that it is unclear why only Eurytominae and not the whole Chalcidoidea have elevated diversification rates, beyond the taxon sampling. Also the lack of increased rates for Vespidae and Formicidae (although you acknowledge them at the end of this section). The hidden states (whatever they may represent) may explain the former but might not explain the latter two as well.

9. Also, while the hypothesis that parasitoid-derived parental care may have reduced extinction risk in those secondary phytophagous lineages, it might be necessary to point out that HiSSE results only suggest lower extinction for state 1A, while the faster diversifying 1B lineages actually have higher extinction. Of course, with extinction being hard to estimate well (as you point out on line 578 as well), this result does not per se prove this hypothesis wrong, but does relativise its credibility a bit.

I am not sure if I entirely follow the MuCID results you describe, but I agree that at this point they would unnecessarily bloat the manuscript if they do not yield anything substantial.

Again, I congratulate the authors on this great study and thank them for their diligent addressing of all my suggestions.

Reviewer #4:

Remarks to the Author:

Dear Authors and Editor,

Thank you for making the suggested changes to the manuscript. I am satisfied that the Authors have responded appropriately to the comments and suggestions that I have made, as well as those made by the other reviewers. I recommend the work for publication with no further suggestions.

Reviewer #5:

Remarks to the Author:

I find the manuscript improved and my prior comments adequately addressed. I recommend just a few more additional minor edits of the newly edited portions:

- 1) Line 432/433 (pdf with track changes). Instead of estimated, do you mean inferred? Do you mean "in Apocrita" or at the origin of it? In general I find the new text from Line 430 - 435 confusing as written.
- 2) Line 493-495. I don't understand the logic here as written. What is "this hypothesis"? The point being made with these clades is unclear.

NCOMMS-22-11260B - Response to reviewers

We thank all reviewers and the editor once again for their time and effort to review our manuscript and provide comments to improve the paper.

Please see below our point-by-point response with our replies inserted in blue font. Line numbers refer to the word document of the manuscript with track changes highlighted.

In addition to the changes outlined below, we have made last minor corrections to the text, tables and figure captions.

REVIEWER COMMENTS

Reviewer #2 (Remarks to the Author):

Once again I am happy about the implementation of my comments (and those of the other reviewers, as far as I can tell), and I commend the authors for this great study, which I now again support to be published. The improved explanations of methods and rationale make things very clear now, and the interpretations of the results paint a clearer picture. I have elaborated a few comments on the newly presented findings, some of which come with suggestions. I believe all my suggestions can be resolved by some minor edits, rendering this manuscript version practically final.

In brief the main suggested changes are:

1. clarify wording on the association of inferred hidden states and inferred rate shifts and avoid cherry-picking those results
2. briefly mention/clarify the result on wasp waist perhaps being solely driven by it encompassing all rate shifts
3. same for parasitoidism (plus a minor wording suggestion)
4. minor typo (along with some comments)
5. very minor formatting error in supp mat
6. very minor point on MiSSE results
7. same as point 2. actually, up to the authors which part of the manuscript the point would be addressed better

(also: a shortening suggestion of two partially redundant sentences)

8. adding clearer acknowledgement of the limits of what diversification rate shifts secondary phytophagy can explain
9. adding short caveat about hypothesis that parental care reduces extinction risk, given the HiSSE results

>We really appreciate these last comments and corrections from reviewer 2. Please see our replies below after the detailed version of each point.

In detail:

1. It was very interesting to see the inferred HiSSE tip states, and I agree that the result isn't as straightforward as one could have hoped. It is actually a bit puzzling that the two hidden states for wasp waist overlap so poorly with the inferred rate shifts (because in theory the SSE states should match the inferred rate shifts to a good extent). It is also striking that for some models, some trait states aren't represented in the tips at all (e.g. OA for wasp waist). However, looking at the transition rates (thanks for adding those figures in Suppdata 11, they make things a lot

clearer), one can see that there are high transitions from 0A to 0B and inversely from 1B to 1A. This might suggest that the model inferred 0A and 1B to mainly be at internal nodes, perhaps explaining some kind of rate dynamics over time there. Plotting all the reconstructed nodes could show that, though there's probably a good amount of uncertainty for deeper nodes, and that would more serve to interpret what the model did than reveal exciting biology, so doing this is probably not necessary here.

> Thank you for the insights. The suggestion that states 0A and 1B may be mainly present at internal nodes is interesting. A deeper investigation of the model behavior is outside the scope of our study, as you note, but could be worthwhile for a different paper.

However, I would urge you to word the section where you point out the overlap of shift clades and taxa with higher likelihood of state 1B a bit more carefully (lines 273-277 in the pdf/word doc with tracked changes). I initially read it and looked at the inferred state likelihoods and thought since merely 12 species with wasp waist have state 1B inferred as more than 50% likely, I probably wouldn't want to argue that their overlap with the clades who had rate shifts inferred would explain much. Only after inspecting the likelihood for Parasitoidism I realised that you meant taxa where the likelihood for 1B is on the higher side within 1B, albeit 1A may still be inferred as more likely in many of them.

Perhaps this could be clarified by wording it similarly to this (merely a suggestion, as I found this the easiest way of communicating my intentions here):

"Most tip states are estimated with higher probabilities in state 1A (Supplementary Data 12), and only very few are inferred as having 1B as the most likely state. However, state 1B has a higher net diversification rate than state 1A and taxa whose marginal probabilities for state 1B are on the higher side (up to 0.58) overlap to some extent with clades that showed elevated rates in BAMM and MEDUSA analyses (Supplementary Data 11)."

Seeing it like that, I indeed find that intriguing, but since also a bunch of other taxa have slightly higher likelihoods of 1B, you should be careful not to cherry-pick these cases, so perhaps the presence of 1B taxa outside the radiations should be noted briefly.

>We agree that the referenced statement could be misleading and have incorporated the suggested re-phrasing and a note on the presence of higher probabilities for 1B also outside the clades in which rate shifts were detected (lines 326-329).

2. It is actually also intriguing that the main state inferred for the Symphyta grade (0B) has relatively low net diversification rates but is actually inferred to have high turnover (i.e. both high speciation and extinction). Possible that this is related to the inferred slowdown in BAMM, though whether you want to elaborate that I shall leave up to you. In any case, the idea that wasp waist might have laid the background to enable the other key innovations which eventually triggered the radiations (e.g. sensu Donoghue or Bouchenak-Khelladi et al.) seems reasonable, as long as you emphasize that there is likely no direct effect (which you do). But it is also likely that the rate increase associated with the presence of a wasp waist is simply driven by the fact that the lineages with that trait contain all the inferred shift clades, and its absence particularly includes the low diversity Symphyta grade, so maybe this should also be mentioned briefly.

>Agreed, we now mention this caveat in the discussion (lines 487-489): "It is possible that the rate increase associated with hidden states in the presence of a wasp waist is driven by the fact that lineages with this trait contain all the inferred rate shifts, while absence of the wasp waist is restricted to lineages with low present-day diversity". We further modified the paragraph slightly to adapt the flow of the text.

3. The same (as in point 1. and 2.) would seem to be the case for Parasitoidism. After all, it is essentially the same as wasp waist, minus BAMM shifts 4 and 5 (and a few smaller clades), which explains why there's not a clear association of its presence with higher rates (to put it

oversimply, the low rates of the Symphyta grade and the shifts 4 and 5 cancel each other out). Though given the conflicting trend depending on the hidden state, I'd rather say the association of this trait with diversification was 'suggested' rather than 'supported' on line 281.

>We changed the wording accordingly (line 342 and 518).

Also here I agree that the overlap of slightly elevated likelihoods for state 1B and 0A with some of the shifts is intriguing, but the same caveat for cherry picking and overinterpretation applies.

>We adjusted the wording of this section to a more cautious statement (lines 349-353) and have removed our interpretation of the elevated likelihoods for state 1B and 0A from the discussion (previous lines 540ff).

4. The results for Secondary phytophagy are probably the most intriguing, especially since the increased likelihoods for the respective hidden states seem to match the BMM shifts even better, especially the cumulative shift probability (i.e. which includes more bees in shift 4, and which differentiates two separate sub-shifts in the Ichneumonidae, which seem to be found by HiSSE too, maybe?). Though also here it is a pity that those hidden states aren't recovered more clearly and that other taxa outside of those shifts also show higher likelihoods for the respective hidden states. Yet I still think your basis to argue that Secondary Phytophagy may have had an impact on diversification rates seems fairly strong.

I think the parenthesis on line 299 should close after "MEDUSA" and not after "Table 4"?

>Corrected, thanks (line 359).

5. Personally, I'd now want to dig deeper into this to figure out whether these results make sense or are artefactual, i.e. whether perhaps another hidden state is needed, or whether there were convergence issues, etc., but it is clear that this would go beyond the scope of this article at this point.

BTW: some of the bold highlighting of the highest probabilities in SuppData 12 are wrong (e.g. some Eurytomidae have 1B as highest probability for phytophagy, but 1A is bold), though I would consider this a very minor issue.

>We thoroughly checked and corrected all bold highlighting in Supplementary Table 12 (and 13).

6. The use of MiSSE as additional confirmation was certainly a good idea and I agree that these results back up the other findings (although the shifts in Cynipidae and the Bees seem to include a few more taxa, and there's also some higher rates in Proctotrupidae and Braconidae, but I agree the association to the shifts is meaningful). As a minor point, I think the results table suggests that for topC-1 there were only 5 hidden states, and only topA-0 had 6?

>Indeed. We corrected the text accordingly (line 387), thank you for catching this mistake.

7. I think the diversification parts of the discussion are also much improved now. The lack of direct effect of wasp waist is made clear enough and the suggestion that hidden states play a major role is highlighted appropriately. Perhaps the aforementioned point that all the radiated clades being within the clade with wasp waist would simply raise its overall diversification rate could be made explicit here, although line 432 already points in that direction.

>As outlined above, we now mention this caveat in the discussion (lines 487-489).

I think the first two sentences on the wasp waist discussion (lines 420-427) largely seem to say the same thing (with the second one being more explicit), so I think those could be merged into one to shorten the text.

>We have merged these two sentences (lines 464-468).

8. The remainder of the discussion combines the results intriguingly with other knowledge on

the group and is very exciting to me. However, regarding secondary phytophagy, I am still missing an acknowledgement that it is unclear why only Eurytominae and not the whole Chalcidoidea have elevated diversification rates, beyond the taxon sampling. Also the lack of increased rates for Vespidae and Formicidae (although you acknowledge them at the end of this section). The hidden states (whatever they may represent) may explain the former but might not explain the latter two as well.

>We acknowledge now that the support of the eurytomine rate shift remains unclear (line 570), and added a note regarding absence of rate shifts in Formicidae and Vespidae (line 530-531).

9. Also, while the hypothesis that parasitoid-derived parental care may have reduced extinction risk in those secondary phytophagous lineages, it might be necessary to point out that HiSSE results only suggest lower extinction for state 1A, while the faster diversifying 1B lineages actually have higher extinction. Of course, with extinction being hard to estimate well (as you point out on line 578 as well), this result does not per se prove this hypothesis wrong, but does relativise its credibility a bit.

>We have added a note pointing out that our results only partially support this hypothesis (line 609-610).

I am not sure if I entirely follow the MuCID results you describe, but I agree that at this point they would unnecessarily bloat the manuscript if they do not yield anything substantial.

Again, I congratulate the authors on this great study and thank them for their diligent addressing of all my suggestions.

Reviewer #4 (Remarks to the Author):

Dear Authors and Editor,

Thank you for making the suggested changes to the manuscript. I am satisfied that the Authors have responded appropriately to the comments and suggestions that I have made, as well as those made by the other reviewers. I recommend the work for publication with no further suggestions.

>Thank you.

Reviewer #5 (Remarks to the Author):

I find the manuscript improved and my prior comments adequately addressed. I recommend just a few more additional minor edits of the newly edited portions:

>Thank you for these final corrections.

1) Line 432/433 (pdf with track changes).

Instead of estimated, do you mean inferred?

>Yes, we meant estimated in the sense of "inferred". This sentence was deleted, however, while re-writing this section based on reviewer #2's comments.

Do you mean "in Apocrita" or at the origin of it?

>We meant "at the origin of Apocrita" and have made this modification (line 482).

In general I find the new text from Line 430 - 435 confusing as written.

>Based also on reviewer #2's comments, we have revised this section significantly and hope it is more easily understandable now (lines 464ff).

2) Line 493-495. I don't understand the logic here as written. What is "this hypothesis"? The point being made with these clades is unclear.

>We were trying to highlight the presence of a faster diversifying hidden state in some of the parasitoid clades. Since this was criticized as a potential over-interpretation by reviewer #2 this sentence has been removed (line 540ff).